# An immunobiliary single-cell atlas resolves crosstalk between type 2 conventional dendritic cells and γδ T cells in cholangitis

Stefan Thomann [1], Helene Hemmer [1], Ankit Agrawal[1], Sukanya Basu [1], Judith Schaf [1], Nadine Vornberger[1], Tobias Krammer [2], Sagar [3], Fabian Imdahl[2], Tanja Poth [4], Marcell Tóth[5], Christina E. Zielinski[6,7,8], Tobias Poch[9,14,15], Jenny Krause[9,16], Andreas Rosenwald[10], Katja Breitkopf-Heinlein [11], Nuh Rahbari[12] & Dominic Grün [1,13] ✉

The liver biliary niche serves as a reservoir of tissue-resident immune cells and supports tissue fibrosis upon damage, yet the role of peribiliary immune cells during cholangitis remains poorly understood. Here, we induce cholestatic liver injury mirroring human biliary diseases with bile acid retention in mice to establish a spatial and multimodal single-cell RNA-sequencing atlas of the liver and liver-draining lymph nodes (LN). We characterized a hepatic disease state trajectory from dendritic cell precursors (preDCs) to a mature subset of pro-inflammatory Mgl2+ type 2 conventional dendritic cells (cDC2B) and observed dynamic crosstalk with γδ T cells inducing an Il17 response (γδ T17). Dissection of the cDC2B-γδ T cell communication node identified the Icosl-Icos pair as an important cell contact-dependent interaction, which was validated in vitro. In vivo, cDC2B depletion attenuated γδ T17 responses in cholestatic liver injury, and liver fibrosis was reduced in a model of inducible γδ T cell depletion and in an Il17-deficient background. Our work demonstrates dynamic turnover of cDC2 within the biliary niche during cholestasis, and a profibrogenic function of γδ T cells contingent on the induction by peribiliary cDC2B, highlighting relevant disease determinants within the immunobiliary and liver-draining LN niche.

Cholestasis and associated biliary immune responses may contribute to liver disease development, since associated inflammation may amplify tissue injury responses independent of its main etiologic factor. For instance, primary sclerosing cholangitis (PSC) is a multi-factorial autoimmune-related disease with rising occurrence in the US and EU[1,2], and the disease course may lead to cirrhosis and the requirement of liver transplantation. While there are different sub-types of PSC[3], and PSC is frequently linked to other chronic inflammatory diseases, the common histological features include periductal fibrosis, proliferation of reactive bile ducts known as ductular reaction (DR), bile duct stenosis, dilatation and rarefication leading to biliary cirrhosis, cholestasis and altered myeloid responses[4,5]. Clinical presentation of PSC and its chronic disease activity may include intermittent episodes of acute cholangitis and cholestasis[6] and involve dysfunctional cDC2 and T cell responses contributing to disease severity[7,8], yet the role of unconventional T cells (UTC), their potential crosstalk with cDC2 and their role in disease amplification remains to be elucidated.

In the hepatic cDC2 niche, continuous reconstitution by circulating cells and the DC life-cycle have been characterized in steady-state and disease[9,10], and follow the steps of liver sinusoid attachment, transmigration to the space of Dissé and migration towards the portal field stroma[11]. The portal microenvironment may furthermore instruct DCs towards a more anti-inflammatory phenotype[12], however, a

---

significant fraction of the liver cDC2 pool has been characterized as proinflammatory cDC2B[13] and the behavior of recently recruited DCs versus niche-adapted cells remains challenging to resolve in inflammation and resolution.

In this study, we explore the tissue niche, differentiation dynamics, and functional roles of cDC2 during cholangitis in the 0.1% 3,5-Diethoxycarbonyl-1,4-Dihydrocollidine Diet (DDC)-diet mouse model[14]. By administering DDC for durations of 3–25 days and following inflammatory resolution for 7 days, we established a high-resolution temporal and multimodal single-cell RNA-sequencing (scRNA-seq) atlas covering the cellular dynamics in cholestasis that shape the immunobiliary niche and the liver draining lymph nodes (LN). Combined with spatial transcriptomics, this resource provides novel insights into differentiation dynamics of cDC2 and their heterocellular interactions and complements available spatiotemporal DDC/PSC data of more abundant cell types[5,15]. Our data reveal crosstalk of cDC2 with γδ T cells governing the inflammatory response, which we confirm in other acute models of liver injury and functionally validate in vivo with the help of diphtheria toxin receptor (DTR)-based depletion of cDC2B using the CD301b-DTR (Mgl2-DTR) and Tcrd-GDL models[16,17]. Together, our work highlights new site-specific and -independent avenues of cDC2B–γδ T cell interactions in cholangitis.

## Results
### Cell type composition of the immunobiliary niche in human and mouse

To determine spatial immune cell type localization in human steady-state liver, we first explored publicly available spatial transcriptomics (ST) data ("Methods") and found enrichment of *CD1C* and *CD3E* transcripts near large vessels and biliary tracts (Fig. 1A). For cell type annotation of this high-resolution ST reference, we generated a human non-cholestatic liver disease control scRNA-seq reference dataset with the aim to enrich for immune cell types. This dataset comprised ~33k cells from 6 patients (clinical characteristics summarized in Fig. S1A) purified by combining enrichment of biliary epithelial cells (BECs) and immune cell populations with unbiasedly sampled cells ("Methods", Figs. 1B and S1B–D). We performed spatial cell type annotation by running RCTD[18], which defined DCs, continuous endothelial cells (CECs), BECs and T cells to be part of the immunobiliary niche (Fig. 1C). Topological niches were predicted by Seurat's *BuildNicheAssay* and confirmed the existence of a portal immunobiliary ("1") and lobular niche ("2"; Fig. 1C), providing evidence for the immunobiliary niche as a T cell- and DC-containing microenvironment. These findings were confirmed using non-cholestatic disease control human liver tissues derived from a characterized tissue microarray (TMA)[19,20] by immunohistochemical detection and correlation analysis of CK7+ BECs, CD34+ CECs, PDPN+ lymphatics (clone D2-40), CD207+ cDC2, CD3+ T cells and TRDC+ cells among other markers enriched at the peribiliary niche (Fig. 1D, E). The immunobiliary niche was conserved in mouse, where multicolor immunofluorescence (IF) revealed co-localization of Epcam+ BECs with CD146+ CECs and CD45+ immune cells (Figs. 1F and S2A). CD11c+ cells were detected in the biliary niche of steady-state mice and co-expressed CD172a (encoded by *Sirpa*). Furthermore, CD301b+ (encoded by *Mgl2*) cells were detected, likely representing cDC2 (Fig. 1G, H).

The xenobiotic DDC model has been described to induce an inflammatory response of the biliary tree, partly mediated by cholangiocyte-driven reactive changes that also affect the peribiliary niche[14,15]. To locally perturb the immunobiliary niche, we made use of the DDC diet, that is known to induce cholangitis and ductular reaction (Fig. 1I)[14]. qPCR detected DDC-induced upregulation of biliary and myeloid marker genes suggested expansion of these populations (Fig. S2B). Physical interaction of CD45+ immune cells with Epcam+ bile ducts was validated by IF image co-localization, indicating a dynamic

immunobiliary crosstalk upon ductular reaction (Fig. S2C–E). To unbiasedly investigate dynamics of gene expression across cell types residing in the immunobiliary niche, we performed scRNA-seq on liver tissue isolated at five different timepoints of DDC diet (day (D) 3, D5, D9, D19, and D25) and from control liver (D0) (Fig. 1J). To enrich all cell types at equal ratios, two independent rounds of cell isolation and sequencing were performed at all timepoints (Methods) and we were able to capture abundant cells of the liver immunobiliary niche, as well as rare liver immune cells such as cDCs (Figs. 1K and S2F–H, Supplementary Data S1, Methods).

### DDC diet induces small duct inflammatory disease

Based on the mechanism of action of DDC in causing cholangitis, we first analyzed the BEC scRNA-seq subset. To dissect the BEC subset of our scRNA-seq atlas, we performed a separate clustering analysis of the two main cholangiocyte populations (clusters 9,22; 5,137 cells, Fig. 1K) resulting in 16 more granular clusters (Figs. 2A and S3A). Our data recapitulates two major subsets: *Dmbt1*+ *Muc4*+ *Sox17*+ BEC (cluster 2,6,11) and *Sox9*+ BEC (remaining clusters). Among *Sox9*+ BEC, clusters 4, 8 and 16 showed an increased expression of chemotaxis- and adhesion-regulating genes (*Ccl2, Ccl20, Vcam1, Mif, Csf1, Tnf, Il23a*) (Fig. 2B, C), which are known to regulate cDC2 and T cells among other immune cell populations. In particular, Il23 is a key cytokine for the induction of an Il17-response which is a known driver of liver inflammation and fibrosis[21]. Increased transcript abundance of *Tnf* and *Il23a* was confirmed within DDC tissue at D10 and D19 using qPCR (Fig. S3B).

To assess the general nature of this pro-inflammatory early liver damage response, we revisited published data of a bile duct ligation (BDL) model and a chronic model of repetitive CCl4-induced liver damage[22]. Acute biliary injury (BDL) showed highly similar gene expression patterns within *Sox9*+ BECs, while chronic CCl4-induced damage did not lead to upregulation of *Tnf, Il23a, Cxcl16* and *Nfkb2* expression (Fig. S2C, D). Intersecting upregulated genes within our *Sox9*+ BEC DDC data (clusters 4,8 vs. 3,14) and *Sox9*+ BEC BDL data (clusters 3,4,7 vs. 1,2,9) revealed a common set of 128 genes (Fig. S2E), which were enriched for gene ontology annotations related to cell communication and inflammatory response (Fig. S2F). These findings highlight small duct BECs as important immune cell niche regulators in acute liver damage, including cDC2 and T cells, independent of the noxious background (DDC/BDL). In this niche, the Nfkb pathway inducer Tnf may trigger a proinflammatory change within the cDC containing immunobiliary niche. To functionally test Nfkb-driven alterations in biliary niche-residing cDCs triggered by BEC-secreted Tnf, we treated cDCs isolated from control livers with Tnf in vitro[23] and observed induction of the maturation markers *Cd80, Ccl17, Ccl22* in cDCs, indicating increased maturation dynamics (Fig. 2D).

To test the existence of an analogous Tnf-inducible Nfkb pathway activation node in the human liver immunobiliary niche within DCs, we made use of our human ST liver dataset and Niche Covariation (NiCo) analysis within a representative portal field data subset[24]. Cell type interaction prediction by NiCo recapitulated the immunobiliary niche and indicated a cellular localization of DCs within a defined cell type neighborhood hosting BECs and CECs (Figs. 2E, F and S3G, H). Since in the steady state we expected an inactivity of this communication node, we screened for covariation of BEC and DC gene programs represented by latent factors (Fa), and detected a negative covariation of BEC Fa2 and DC Fa1 (Fig. 2F, G). Correlating ligand-receptor pairs contained an interaction of BEC-derived *GRN* with DC-derived *TNFRSF1A/B*, an interaction mediating either pro- or anti-inflammatory functions dependent on the proteolytic cleavage of the gene product[25].

In conclusion, these results highlight small duct BEC-derived conserved and model-independent inflammatory gene expression patterns that may modify the surrounding myeloid niche.

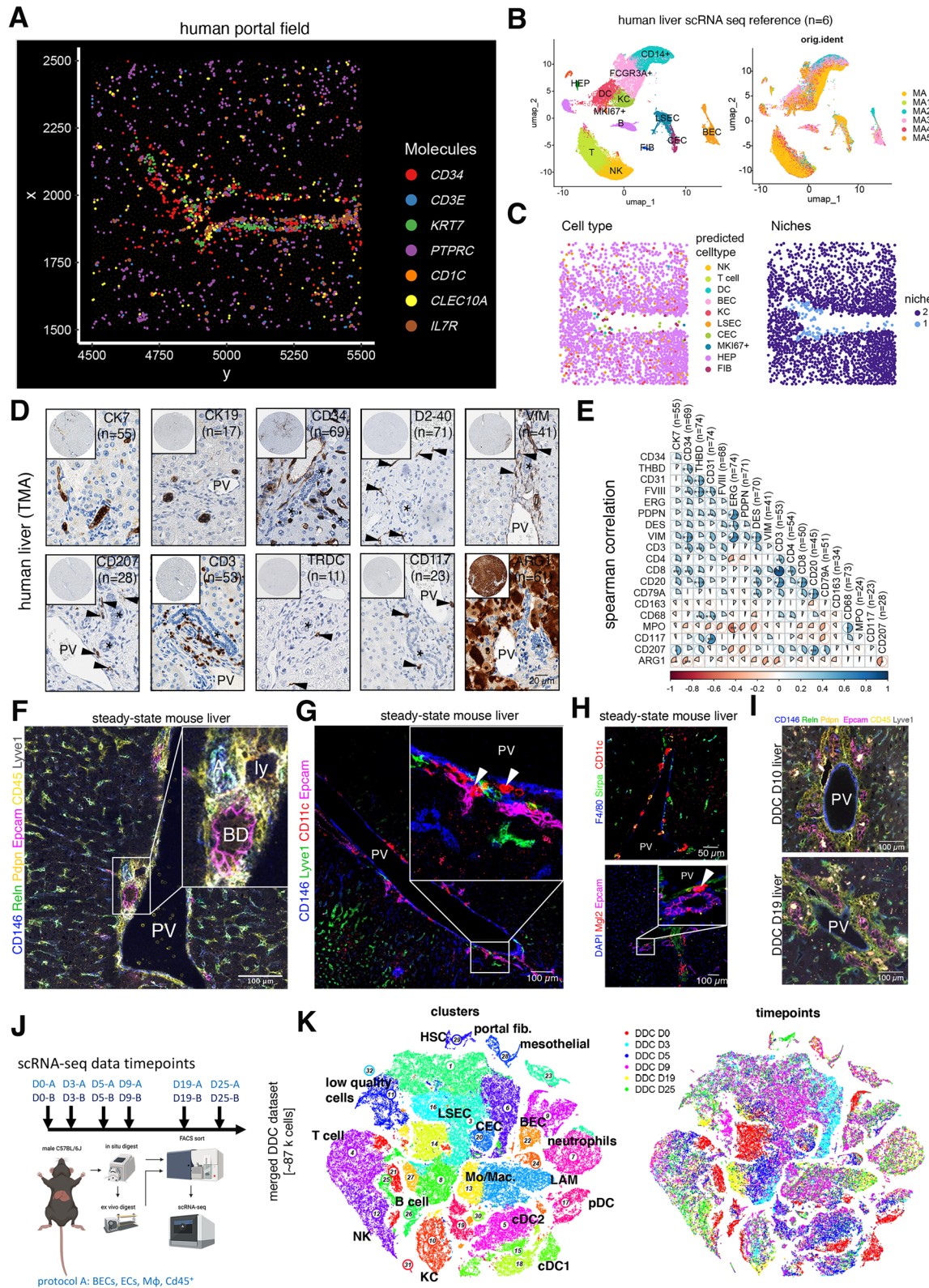

## Disease stage-specific cell state transition in cDC2 reflects immature niche restoration

Based on their spatial proximity to the BEC compartment and their capacity to activate lymphocytes, we explored the temporal disease dynamics of DC in our DDC disease atlas. While DC subtype frequencies were approximately equally distributed in steady-state, a frequency shift towards cDC2 predominance was observed starting from D3

(Fig. 3A). To confirm this shift by FACS analysis, we made use of a previously published gating strategy to discern cDC1 and cDC2[26] (Fig. S4A). Increased cDC2 proportions at D5/6 and D10 after DDC diet were observed, extending previous observations on frequency changes within the cDC1/2 compartment in short-term DDC-fed mice[7] (Fig. 3B). Cell state dynamics of cDC1/2 were comparably affected in a model of acute hepatotoxic injury and ductular reaction (CDE diet; Fig. S4B).

**Fig. 1 | Immunobiliary microenvironment in human and mouse. A** Spatial transcriptomics data subset of a human portal field, displaying expression of 7 RNA molecules of interest in the immunobiliary niche. **B** UMAP representation of a human control liver scRNA-seq reference (32,901 cells from 6 patients) highlighting major cell types of the portal niche (left) and individual samples (right). **C** Spatial map highlighting cell type composition predicted by label transfer (left) and inferred niche domains (right). **D** Immunohistochemical stainings of a human TMA containing non-cholestatic liver disease control tissues including CK7+, CK19+, CD34+, PDPN+ (clone D2-40), VIM+, CD207+, CD3+, TRDC+, CD117+, ARG1+ cells (arrowheads). Asterisks demarcate bile ducts adjacent to portal veins (PV). Small magnification displays TMA core, and large magnification portal region containing cell types of interest. **E** Pairwise correlation of marker-positive cells within non-cholestatic liver disease control cores (*n* indicates number of cores). Spearman correlation coefficient displayed as pie chart and color code. Asterisks indicate level of statistical significance (*$p < 0.05$, **$p < 0.01$, ***$p < 0.001$). Two-sided Spearman's rank correlation test. **F** IF of major cell type markers of immunobiliary niche on steady-state mouse liver tissue. High magnification displays the presence of CD45+ immune cells close to PV, hepatic artery (A), bile duct (BD) and lymphatic vessels (ly). One representative portal field (*n* = 3 mouse livers, two independent experiments). **G** IF of steady-state mouse liver portal fields visualizes close spatial proximity of Epcam+ BECs with CD11c+ cells (*n* = 3 mouse livers, two independent experiments). **H** IF showing CD11c+ Sirpa+ cells (top) and close proximity of Mgl2+ cDC2B and Epcam+ BECs (bottom) in steady-state liver tissue (*n* = 3 mouse livers, two independent experiments). **I** IF of major cell type markers on DDC D10 and D19 mouse liver tissue visualizes expansion of the immunobiliary niche (DDC D10: *n* = 3 mouse livers, DDC D19 *n* = 4 mouse livers; two independent experiments). **J** Scheme of DDC scRNA-seq disease atlas containing a control (D0) and 5 disease timepoints (3–25 days). Two isolation protocols with different enrichment of cell types were used per timepoint. **K** tSNE representation of the complete DDC atlas data ( - 87k cells) highlighting cell type annotation (left) and individual timepoints (right). A hepatic artery, BEC bilary epithelial cell, BD bile duct, cDC conventional dendritic cell, CEC continuous endothelial cell, D day, DC dendritic cell, DDC 3,5-Diethoxycarbonyl-1,4-Dihydrocollidine, FIB fibroblast, HEP hepatocyte, HSC hepatic stellate cell, KC, Kupffer cell, LAM, lipid-associated macrophage, LSEC liver sinusoidal endothelial cell, ly lymphatic vessel, Mac Macrophage, Mo Monocyte, pDC plasmacytoid dendritic cell PV portal vein.

This common dynamical switch towards cDC2 predominance prompted us to analyze cell state dynamics of cDC2 within our DDC atlas (cluster 5 and 15 of the main atlas, 7563 cells), on which we performed separate clustering to obtain a more granular annotation (Fig. 3C). Clusters 3, 12, 15 were excluded from further analysis based on remnant expression of cDC1 genes (*Xcr1, Cadm1, Gcsam*). Anti-inflammatory cDC2A and pro-inflammatory cDC2B can be distinguished based on their gene expression profiles[13]. The expression of *Clec10a* and *Mgl2*, hallmark genes for a cDC2B identity, indicated a predominant pro-inflammatory phenotype of the cDC2 subset, while maturation markers *Ccr7* (cluster 11) or *Ly6c2* (cluster 4) reflected temporal and differentiation-related changes along the x-axis in the UMAP representation (Fig. 3D). Prompted by these observations, we focused on trajectory analysis within the cDC2B compartment. Disease onset was associated with the expression of "maturation-on" (MAT-ON) genes (cluster 11), which have been described in the context of DC maturation[27] and included, e.g., *Ccl17, Ccl22*, and *Arpin* (Fig. S4C). Cells derived from later timepoints (from D5 onwards) included Mgl2neg DC precursors (preDCs) expressing *Cd7, Ly6c2, Tcf4, Runx2, Siglech*, and *Csf1r* (Fig. S4C). The majority of cDC2 derived from D9 onwards were *Mgl2*-negative, suggesting that *Mgl2* expression may be a proxy for the cDC2B tissue residency and maturation state. However, Mgl2 expression in cDC2B is downregulated as part of the MAT-OFF switch, as seen in tissue egressing Ccr7+ DCs (cluster 11).

We applied VarID2[28] to infer transition probabilities between clusters, which further supported the concept of a cDC2B trajectory linking disease-specific cell states, where *Mgl2*+ clusters were interconnected as part of the early DC maturation trajectory (clusters 1 – 7 – 9; Fig. S4D). To systematically investigate gene expression dynamics during maturation, we calculated *self-organizing maps* (SOMs) of pseudotemporal gene expression profiles along this trajectory (Fig. S4E)[29]. The genes in SOM modules 5, 13, and 20, which were upregulated along the trajectory (Fig. S4F) encoded soluble factors, transcription factors or proteins interacting with Nfkb1/2 (Relb, Nfkbia, Nfkbid, Nfkbiz; Fig. S4F). Since cDC2B have been linked to the induction of Il17-responses[30], we next examined the expression of genes within the gene ontology pathway "positive regulation of T helper 17 type immune responses" in the *Mgl2*+ clusters. Indeed, increased expression of pathway genes, such as *Jak2, Nfkbiz, Nfkbid, Nlrp3* was observed along the *Mgl2*+ cDC2B disease trajectory (clusters 1 – 7 – 9, Fig. S4G). Il17-response induction at the biliary niche was supported by DDC-induced *Il23a* production in BECs (Fig. 2B, C).

For the validation of increased recruitment of preDCs in prolonged DDC diet, Mgl1 and Mgl2 were co-detected using immunohistochemistry, which marks immature cDCs and macrophages[31]. As such, combined Mgl1 and Mgl2 detection does not enable quantification of mature Mgl2+ cDC2B but detects macrophages and cDC2 independent of their maturation state. Mgl1/2+ cells were specifically enriched in inflamed biliary foci and total numbers were increased at DDC D10 and D19 compared to controls (Figs. 3E and S4H). This specific cellular localization was also evident in a model of lobular liver injury and steatohepatitis (Fig. S4H, I). To assess the maturation state of hepatic cDC2, we further investigated whether the loss of *Mgl2* mRNA within cDC2 in prolonged disease can be equally observed on the protein level. Indeed, the relative proportion of Mgl2+ cDC2B was significantly reduced from 34.4% ± 0.2% (D0) to 19.8% ± 4.9% at D10 (Figs. 3D, F and S4A). Finally, we tested the plasticity of Mgl2neg cDC2 precursors by comparing cDC2 derived from long-term DDC diet (D18/19) to cDC2 that underwent an episode of disease recovery (DDC D18/19 + 7 days of recovery, "STOP"; Fig. 3G). Analyzing cDC2A/B gene signatures revealed induction of the restorative cDC2A gene signature during disease resolution (cluster 2 and 6), while proinflammatory cDC2B genes remained partially expressed (Fig. 3H).

In conclusion, the cDC2 scRNA-seq data reveal DC maturation and temporal exhaustion of liver-resident Mgl2+ cDC2B at disease progression, along with recruitment of Mgl2neg preDCs that may differentiate towards a proinflammatory or restorative phenotype depending on the niche signal input.

## Heterogeneous γδ T cells produce profibrogenic Il17a in a niche-dependent manner

To better delineate T cell subsets with key functions during the inflammatory response and tissue fibrosis, we determined *Il17a* expressing T cell populations in our scRNA-seq atlas[32]. The main *Il17a*-expressing population was γδ T cells (cluster 15) co-expressing *Trdc*, *Sox13* and *Blk*[33] (Fig. S5A, B), which segregated into *Il17a*high cells positive for *Scart1* (encoded by *Cd163l1*), *Trdv4* and *Trgv6*, and *Il17a*low cells expressing *Scart2* (encoded by *5830411N06Rik*) (Fig. S4C). Since γδ T cells were sparsely represented in our atlas (282 cells), we multiplexed enriched γδ T cells from a hepatic control (D0), DDC D5- and D9-fed mice for scRNA-seq (Fig. 4A, B). Gene signatures defining Vγ6+ and Vγ4+ γδT cells indicated that Scart1+ cells were Vγ6+, while Scart2+cells were Vγ4+[34]. Both populations expressed *Nr4a1* and *Cd5*, indicating T cell receptor (TCR) activation, and *Scart1*+ γδ T17 cells (clusters 0, 3) expressed *ll17a* at higher levels (Figs. 4B and S5D). γδ T17 cells expressed different receptors that would enable their interaction with cDC2 and induce Il17 production, including, e.g., the Ccl2-Ccr2, Ccl20-Ccr6, Il1b-Il1r1, Il23a-Il23r, Ccl17/Ccl22-Ccr4 axes (Fig. 4C). These observations suggest a multifaceted crosstalk of γδ T cells with BEC and cDC2, expressing *Ccl17, Ccl22* and *Il1b* (Fig. S4C). Close proximity

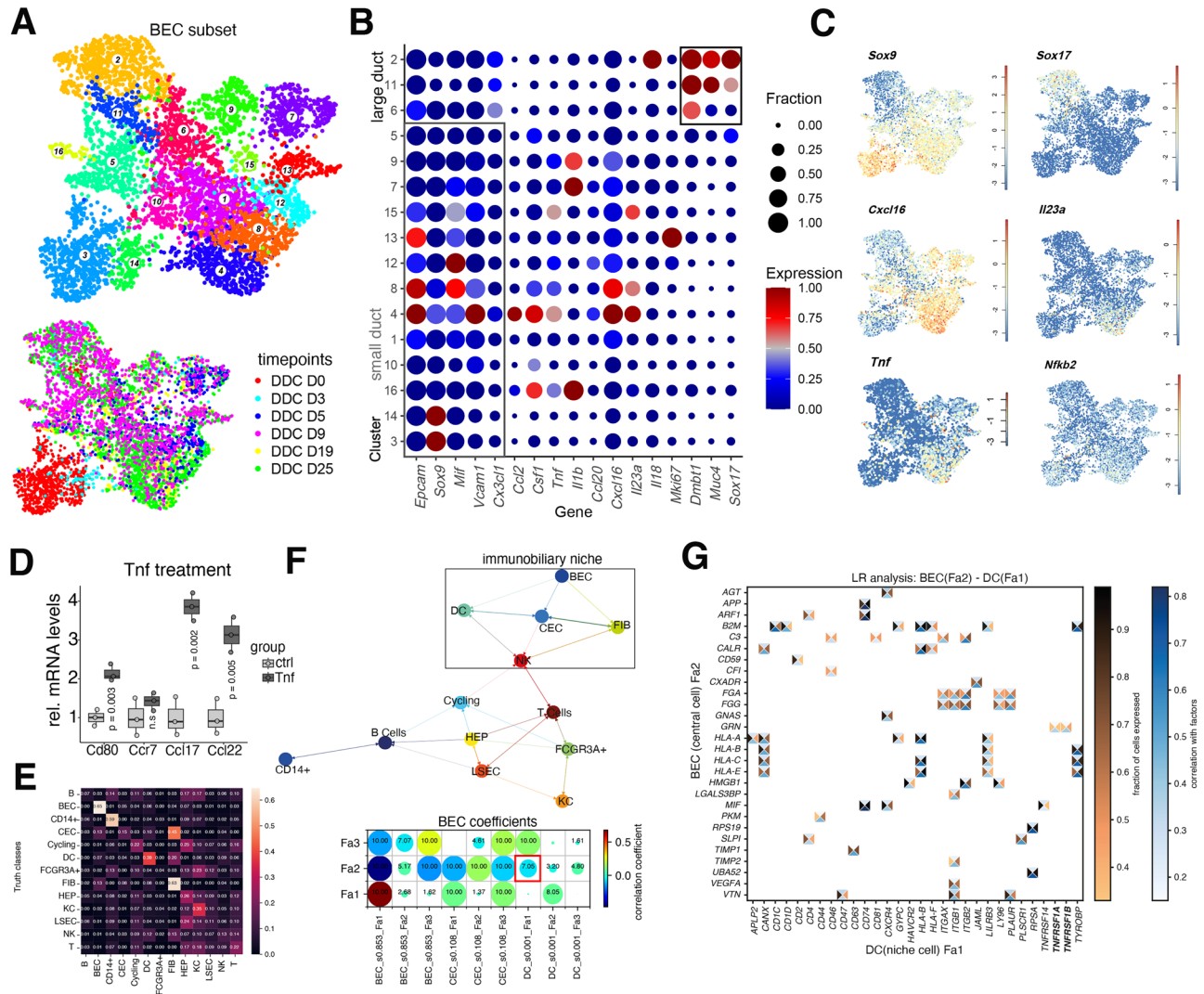

**Fig. 2 | DDC induces ductular reaction and small duct inflammatory disease.**
**A** UMAP representation highlighting BEC sub-types (top) and individual timepoints (bottom). **B** Dotplot displaying BEC cluster-specific gene expression. *Z* score of the mean expression is color-coded and fraction of cells expressing the gene is encoded by dot size. **C** UMAP of log-normalized expression of BEC and inflammatory markers. **D** Boxplot displaying relative tissue mRNA levels of *Cd80, Ccr7, Ccl17* and *Ccl22* in cultivated liver DCs for untreated control and after Tnf treatment (*n* = 3 mice per group). Two-sided *t*-test. **E** Confusion matrix displaying NiCo-based central cell identity prediction based on surrounding niche cell type frequencies within a human portal field (same as Fig. 1A). **F** Top: Spatial cell type interaction map derived by NiCo. Normalized logistic regression coefficient cutoff *c* = 0.07. Bottom:

Regression coefficients between latent BEC factors (*y*-axis) and colocalized niche cell types (*x*-axis). Circle size encodes *p* value (−log₁₀ scale, derived from two-sided *t*-test), circle color encodes ridge regression coefficients. **G** Ligand-receptor pairs correlated with covarying BEC Fa2 and DC Fa1 (cc, central cell; nc, niche cell). The rectangle's north and south faces represent ligand and receptor correlation to the factors, while west and east faces represent the proportion of cells expressing the respective ligands or receptors. Gene correlation threshold 0.15 and gene expression threshold of 0.33 were applied. Ligands, *y*-axis; receptors, *x*-axis. BEC biliary epithelial cell, CEC continuous endothelial cell, D day, DC dendritic cell, DDC 3,5-Diethoxycarbonyl-1,4-Dihydrocollidine, FIB fibroblast, HEP hepatocyte, KC Kupffer cell, LSEC liver sinusoidal endothelial cell, LR ligand receptor.

---

relationships depicted by IF using γδ T cell reporter mice (Tcrd-GDL model[17]) further substantiated such a hypothesis (Fig. S5E, F).

To extend this observation, we conducted a ligand-receptor analysis of a merged liver cDC2 and γδ T cell dataset using CellChat[35] (Fig. S6A–C). This analysis further identified contact-dependent and soluble interactions, including Icosl-Icos, Pvr-Cd226 or Cxcl16-Cxcr6, Lgasl9-Cd44 (Fig. S6B, C).

To support the hypothesis of differential signaling patterns in *Scart1⁺/Scart2⁺* γδ T cells giving rise to distinct transcriptional regulation, we ran transcription factor (TF) regulon predictions using SCENIC[36]. TF regulons exhibited differential activities within and between *Scart1⁺* and *Scart2⁺* γδ T17 subsets (Fig. 4D). For example, *Scart1⁺* γδ T17 cell regulons included Rora, Maf, Irf4, and Xbp1, whereas

Klf2 and Klf4 were more active within *Scart2⁺* γδ T17 cells, in accordance with published data[37,38]. We then used regulon specificity scores (rss) to assess TF regulon cell type specificity. Rora, Maf, Irf4 TF regulon activity in *Scart1⁺* γδ T17 cells (cluster 0 and 3) displayed high rss values, while Klf2 and Jund were specific for *Scart2⁺* clusters 1 and 5 (Figs. 4E and S7A). Motif analysis for Rora, Irf4, and Xbp1 predicted target gene expression in *Scart1⁺* γδ T cells, including *Il17a* (Fig. 4F).

Flow cytometric analysis of Il17a on αβ T cells (βTCR⁺) and γδ T cells confirmed increased relative γδ T17 cell numbers in cholangitis at D5/6 and D10 with a comparable but delayed trend in the CDE model (Fig. S7B).

To interrogate whether γδ T and αβ T cells compete for signal cues within the peribiliary niche, Tcrd-knockout (KO) mice were

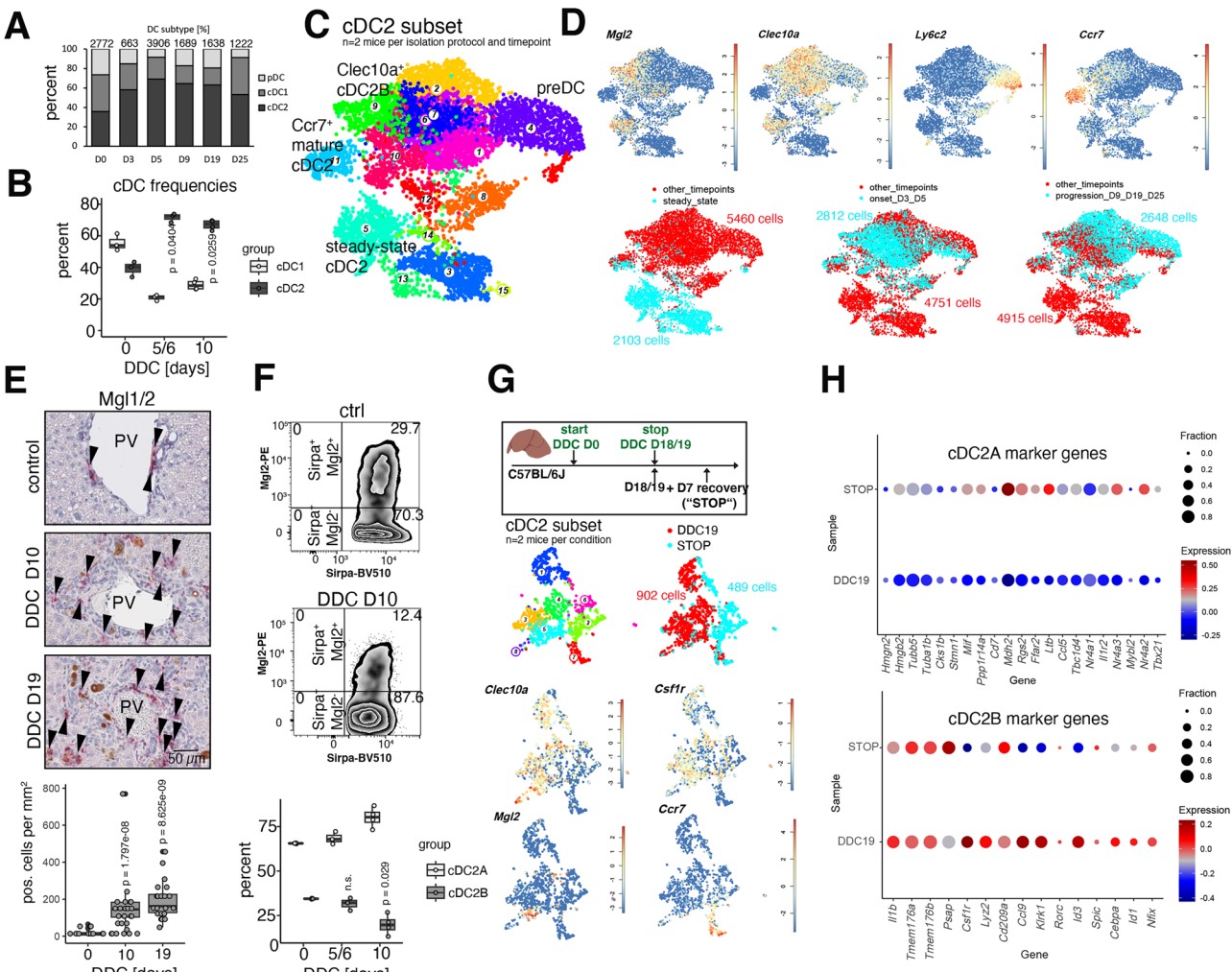

**Fig. 3 | Disease stage-specific cell state transition in cDC2 reflects immature niche restoration. A** Barplot displaying DC-subtype frequencies per timepoint in the DDC atlas data. Captured cell numbers are indicated at the top of each column. **B** Boxplot displaying cDC1/cDC2 FACS-based quantification at D0, DDC D5/6, and D10. One-sided Wilcoxon test (D0, D5/6: $n = 3$ mice, D10: $n = 4$ mice per group, two independent experiments). **C** UMAP representation of cDC2 subset (Cl.5,15 from Fig. 1K). **D** UMAP representation highlighting log-normalized expression of cDC2B marker genes (top) and timepoints of different disease stages (bottom). Number of cells per condition indicated. **E** Top: Immunohistochemical staining of Mgl1/2 in control and DDC D10/D19 liver tissues. Arrowheads highlight Mgl1/2+ cells. Bottom: Quantification of Mgl1/2+ cells across conditions. Every dot represents a tissue section that has been quantified. Two-sided Wilcoxon rank sum test (D0: $n = 4$ mice;

D10, D19: $n = 3$ mice/group). **F** FACS plots displaying hepatic LC− Cd64− MHC+ CD11c+ Sirpa+ cells (top) and boxplot depicting relative fraction of Mgl2+ cDC2B (bottom). One-sided Wilcoxon test (two independent experiments, D0, D5/6: $n = 3$ mice, DDC D10: 4 mice/group). **G** Top: experimental design comprising a DDC D18/19 and a recovery (DDC18/19 + D7 recovery; "STOP") timepoint. Bottom: UMAP representation of cDC2 subset, samples, number of cells, and log-normalized expression of marker genes. **H** Dotplot displaying sample-specific cDC2A/B gene expression in DDC D18/19 and STOP conditions. Log-normalized gene expression is shown, and the fraction of cells expressing the genes is encoded by dot size. cDC conventional dendritic cell, D day, DDC 3, 5-Diethoxycarbonyl- 1,4-Dihydrocollidine, PV portal vein.

treated with DDC. Indeed, compensation of the γδ T17 function was reflected by a higher percentage of Il17a+ αβ T cells in Tcrd-KO versus control mice at DDC D5, suggesting potential competition for γδ T17/Th17 polarization cues in liver cholestasis within a shared micro-anatomical niche (Fig. S7C).

In vitro treatment of γδ T cells with DDC D5 bile did not induce Il17a expression (Fig. S7D). Conditioned medium of DDC D5 derived DCs showed a trend towards inducing increased Il17a+ γδ T cell frequencies ($p = 0.06$), however, compared to direct co-culture, the fraction of Il17a+ cells was low (Fig. S7E). Direct co-culture of DDC D5- or control liver-derived DCs with UTCs from control spleen led to increased Il17a production in γδ T cells but only weak induction in αβT cells, indicating a DC-derived contribution to the induction of a γδ T17 cell state (Figs. 4G and S7F). Based on this observation, we used TCR stimulation in vitro to further elucidate the impact of contact-

based/soluble mediators in inducing a γδ T17 cell state. Specific stimulation with recombinant Il18, Cxcl16, Lgals9 and activating Icosl-antibodies revealed a moderate γδ T17 cell state induction for Icos activation, while soluble mediators did not induce significant differences (Figs. 4H and S7G). This prompted us to test a disease-contributing role of γδ T cells in DDC-induced damage response and cholestasis using a constitutive (Tcrd-KO) and conditional depletion model of γδ T cells (Tcrd-GDL[17]). Conditional γδ T cell depletion was associated with reduced circulating total bilirubin and alkaline phosphatase levels indicating reduced cholestasis and disease activity (Fig. 4I). Furthermore, tissue-derived transcript levels of *Pdgfrb, Col1a1* were reduced in Tcrd-KO and/or Tcrd-GDL liver tissues implying a functional contingency of liver fibrosis on γδ T cell-derived Il17a (Fig. 4J). To directly test the effect of Il17a on fibrosis, we treated wildtype (wt) and Il17a/f KO mice with a long term DDC diet (16 days)

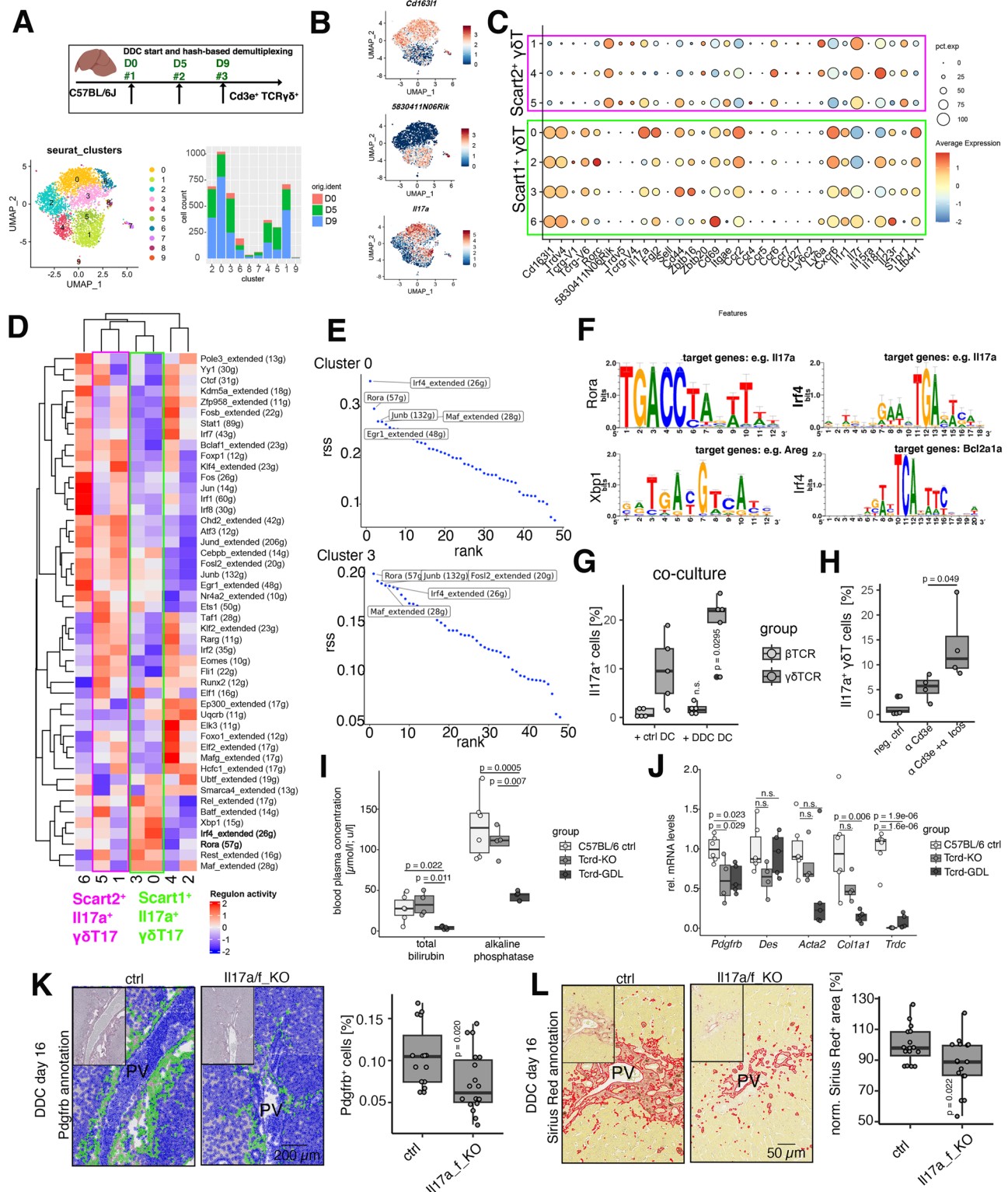

and quantified the number of Pdgfrb⁺ myofibroblasts and collagen (Sirius Red) in tissue sections. The number Pdgfrb⁺ cells and collagen area was significantly reduced in Il17a/f KO versus control mice, mirroring a profibrogenic role of Rorc-induced Il17-responses[39] (Fig. 4K, L). Similarly, tissue-derived transcript levels for *Acta2, Des, Col1a1* were reduced in DDC-treated Il17a/f KO vs. controls (Fig. S7H).

Together, these results suggest Il17a-producing γδ T17 cells as an early-onset fibrosis-supporting communication node in murine cholangitis.

## Spatial niche deconvolution resolves cDC2B and γδ T cell-containing cholangitis-associated tissue domains

For undeterred inspection of tissue dynamics in the cDC2B-γδ T cell niche, we generated imaging-based spatial transcriptomics data of DDC D5 liver. For high confidence cell type identification of γδ T cells and cDC2B, we used a 379 gene atlassing panel adjunct with 100 add-on genes to further improve cell type resolution of cDC2 and γδ T cells (Methods, Table S1). QC metrics (233 median transcripts/cell and >52 million high-quality decoded transcripts) indicated a high quality ST

**Fig. 4 | *Scart1*⁺ and *Scart2*⁺ γδT cells produce profibrogenic Il17a in a niche-dependent manner. A** Top: experimental design of the multiplexed liver γ δ T dataset. Bottom left: UMAP representation of the γδ T17 subset. Barplot displaying γδ T17 cluster-specific absolute cell numbers of multiplexed D0, D5, and D9.
**B** UMAP representation of log-normalized expression of γδ T cell markers *Scart1* (Cd163l1), *Scart2* (5830411N06Rik), *Il17a*. **C** Dotplot displaying cluster-specific gene expression. Log-normalized gene expression is color-coded, and the fraction of cells expressing the gene is encoded by dot size. Boxes highlight *Scart1*⁺ (green) and *Scart2*⁺ (magenta) clusters. **D** Heatmap displaying SCENIC-inferred TF regulon activity in *Scart1*⁺/*Scart2*⁺ γδ T17 clusters (boxes highlight Il17a⁺ clusters). Predicted TF regulon activity displayed as a color code. **E** Rank visualization of *Scart1*⁺ γδ T17 cluster-specific regulon specificity scores (rss). Top 5 regulons indicated.
**F** Predicted Rora-, Irf4- and Xbp1-specific binding motifs and target genes.
**G** Boxplots displaying frequency of Il17a⁺ cells within Cd3⁺ βTCR⁺ (light gray)/Cd3⁺ γδ TCR⁺ (dark gray) compartments after hepatic DC coculture derived from control or DDC D5-treated mice. One-sided *t*-test (five independent experiments).
**H** Boxplots displaying frequency of Il17a⁺ cells in CD3⁺ γδ TCR⁺ cells after anti-CD3e

or anti-CD3e + anti-Icos stimulation. One-sided *t*-test (four independent experiments). **I** Boxplot displaying blood plasma concentrations of total bilirubin and alkaline phosphatase in DDC D5-treated control (ctrl), Tcrd-KO and Tcrd-GDL mice. One-sided ANOVA followed by Tukey's test (C57BL/6: *n* = 6 mice, Tcrd-KO: *n* = 4 mice, TCRD-GDL: *n* = 5 mice). **J** Boxplot displaying relative *Pdgfrb*, *Des*, *Acta2*, *Col1a1* and *Trdc* mRNA levels (qPCR) in DDC D5-treated ctrl, Tcrd-KO and Tcrd-GDL liver tissues normalized to control. One-sided ANOVA followed by Tukey's test (C57BL/6: *n* = 6 mice, Tcrd-KO: *n* = 4 mice, TCRD-GDL: *n* = 5 mice). **K** Left: Pdgfrb immuno-histochemistry and cell detection (green overlay) of control (ctrl) and Il17a/f KO mice after 16 days of DDC. Right: Boxplot displaying percentage of Pdgfrb⁺ cells. One dot represents one quantified tissue section. Two-sided *t*-test (*n* = 4 mice/group). **L** Left: Sirius Red histochemistry and area detection (red overlay) of control (ctrl) and Il17a/f KO mice after 16 days of DDC. Right: Boxplot displaying percentage of stained area. One dot represents one quantified tissue section. Two-sided *t*-test (*n* = 4 mice/group). D day, DDC 3,5-Diethoxycarbonyl-1,4-Dihydrocollidine, rss regulon specificity score, PV portal vein.

dataset that was further analyzed using NiCo[24]. For NiCo analysis, we generated a merged and cleaned DDC atlas reference of the main atlas (Fig. 1K) and liver γδ T cells (Fig. 4A) comprising ~81 k cells (Methods, Fig. S8A). DDC D5 ST data confirmed enrichment of cDC2 and γδ T17 cells at the periportal and biliary site (Fig. 5A, B). Label transfer of annotated scRNA-seq cell labels resulted into classification of 25 spatial cell types including small and large duct BEC, Il17a⁺ γδ T cells and DC subsets (Fig. 5C, D). NiCo predictions of cell-cell interactions identified an intertwined network containing specialized immune (e.g. naïve T vs. Il17a⁺ γδ T cells) and functional niche environments (e.g. proliferating niche). Immunobiliary cell types including small duct (sd) BEC, cDC2 and Il17a⁺ γδ T cells exhibited mutual colocalization within a niche (Fig. 5E, F). To demonstrate colocalization of γδ T17 and cDC2 within the immunobiliary niche, we used the add-on pool of our ST dataset, which contained multiple DC and γδ T cell-specific genes (*Mgl2*, *Clec10a*, *Trdv4*, *Cd163l1*, *5830411N06Rik*, *Il23r*) and enabled the assessment of T17 effector state (e.g., *Rorc*, *Irf4*, Fig. 5G). Niche covariation analysis for small duct (sd) BEC and cDC2 suggested commonly regulated gene programs within identified latent factors (Fa1) that were captured by NiCo's factor analysis (Fig. S8B)[24]. Spearman correlation analysis of highly correlating genes for Fa1 in sd BEC identified genes representing cellular stress responses (*S100a11*, *Hscp90ab1*) and changes in cell mobility/adhesion (*Anxa2*, *Cd9*). Highly correlating genes for Fa1 of cDC2 included genes related to DC maturation and antigen presentation (*H2-Ab1*, *H2-Eb1*, *H2-Aa*). These results implied coordinated cellular responses in this close proximity neighborhood. We further investigated if such covarying gene programs may be connected to cell-cell communication between respective cell types. NiCo's inbuilt cell-cell communication analysis predicted sd BEC-derived signals that may modify cDC2 cell states (Fig. S8C). Once again, the ligand receptor pair Tnf-Tnfrsf1a was detected, similar to the human steady-state communication analysis (compare Fig. 2). Based on the identification of sd BEC, cDC2, Il17a⁺ γδ T cell neighborhoods in DDC D5 mouse liver, we further investigated, if similar cell-cell proximity relationships are detectable in human liver diseases such as PSC. While PSC pathophysiology is more complex and not identical with the acute injury-driven immune response in a xenobiotic model of cholangitis, we could still detect TRDC⁺ and CD207⁺ cells, which showed a localization-dependent concentration towards inflamed biliary foci compared to a lobular localization (average TRDC⁺ cell concentration in portobiliary and lobular regions was 66.33 ± 54.07/mm² and 19.87 ± 25.38/mm²; Fig. 5H). Furthermore, close proximity niche relationships were detected by CD207/TRDC and CLEC10A/TRDC in situ hybridization (Fig. 5H).

In conclusion, DDC D5 ST data in early cholestasis recapitulated in situ distribution of rare immune cell populations such as cDC2/ γδ

T cells and their niche neighborhoods at initiation of an early fibrotic response.

## Early cDC2B depletion and replacement by epigenetically restricted cDC2 results in reduced γδ T17 effector differentiation

To uncover the role of Mgl2⁺ cDC2B in regulating γδ T cell recruitment and function in the context of DDC-induced biliary injury, we depleted cDC2B using diphtheria toxin (DT) in Mgl2-DTR mice[16] ("DDC D5-DT", Figs. 6A and S9A). The cDC2 subset of these data confirmed a strong abrogation of *Clec10a*⁺ *Mgl2*⁺ cDC2Bs and maturation-associated gene expression was reduced, while preDC-associated genes were up-regulated and cDC1/pDC were only marginally affected (Figs. 6B and S9B, C). Pathway enrichment among differentially expressed genes between cDC2 from the depleted versus control animals reflected a dysfunctional, preDC-associated cellular state according to gene signatures that affect cell migration (GPCR, chemokines), and outside-in signaling (TNFR1 signaling, Fig. S9D). Upon cDC2B depletion, BECs exhibited reduced expression of pro-inflammatory mediators *Tnf*, *Il23a*, *Ccl2* and *Ccl20* (Fig. S9E). Top differentially expressed genes within the γδ T cell subset included *Il17a*, which was significantly reduced in cDC2-depleted animals (Fig. 6C, D, Methods).

Flow cytometric analysis of Mgl2-depleted versus control animals revealed a significant reduction of Il17a⁺ γδ T cell frequencies, with a similar trend in βTCR⁺ αβ T cells (*p* = 0.07, Fig. 6E). To capture liver γδ T cells at higher resolution, we enriched γδ T cells from Mgl2-depleted mice for scRNA-seq and interrogated the frequency distribution of γδ T17 cells by comparing DDC D5 and DDC D5-DT derived γδ T cells (Fig. 6F). Reduced total *Scart1/2*⁺ γδ T17 cell numbers were observed in the DT condition, with a more prominent reduction in the *Scart2*⁺ population (44% vs. 85% reduction, Fig. 6G, H). Further subtyping of *Scart1/2*⁺ γδ T17 cells revealed individual clusters with circulatory and resident gene signatures[40,41], respectively (Figs. 6G and S9F, G). A reduced proportion of circulatory cells in the D5-DT versus the D5 condition indicated altered γδ T cell recruitment and tissue half-life in cDC2B-depleted mouse livers (Fig. 6G, H). qPCR analysis of tissue-derived transcript levels of D5/D10 with DT-mediated cDC2-depletion resulted in decreased abundance of profibrotic transcripts (*Pdgfrb*, *Des*) indicative of reduced fibrosis initiation (Fig. S9H).

To address how cDC2B depletion and replacement of immature cDC2 may affect γδ T17 polarization across molecular layers, we performed a combined single-cell transcriptome RNA-seq and Assay for Transposase-Accessible Chromatin using sequencing (ATAC-seq) analysis (10x Multiome) on a control and a disease time-point, where all subclusters including preDCs can be observed (control (D0) and DDC D5, see "Methods"; Figs. 6I and S10A−C). In the cDC2 subset, label transfer of RNA to ATAC clusters revealed similar manifold architecture and transitioning cell states in both data modalities proposing

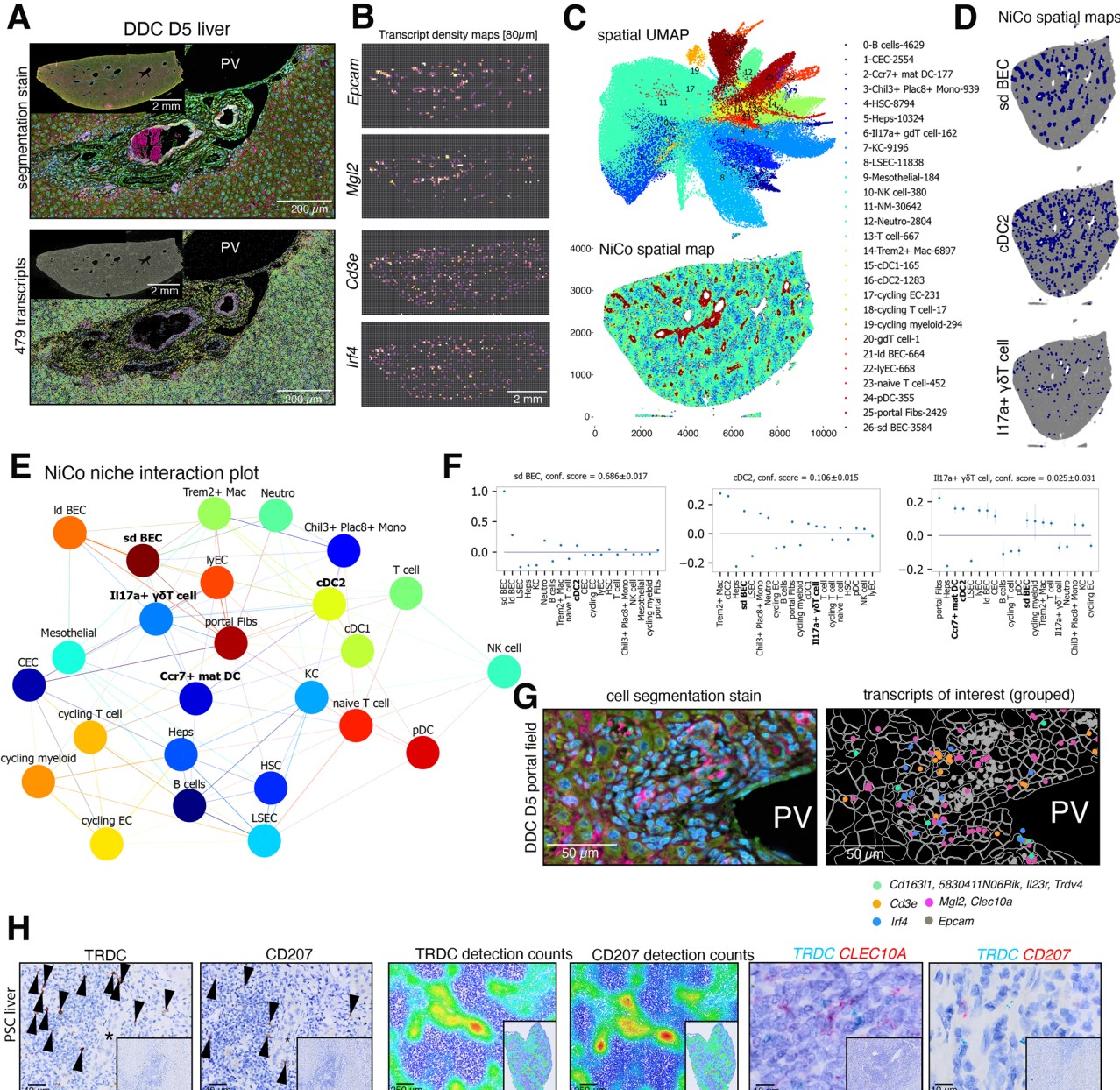

**Fig. 5 | Spatial niche deconvolution resolves cDC2B and γδ T cell-containing cholangitis-associated tissue domains. A** Portal field overview of cell segmentation stain (top) and decoded transcripts (bottom) of a mouse DDC D5 liver tissue (*n* = 1). Image overviews of whole sample are shown on the top left. Scale bar, 2 mm and 200 μm. **B** Density maps displaying DDC D5 liver tissue overview with binned expression of *Epcam, Mgl2, Cd3e* and *Irf4*. Bin size, 80 μm. **C** Spatial UMAP highlighting Leiden clusters (top) and spatial location map (bottom) highlighting NiCo cell type annotations of DDC D5 liver tissue. Right column indicates cluster number, annotated cell types and predicted cell numbers in spatial section. NM indicates not matched cells[24]. **D** Spatial maps of NiCo-predicted BEC, cDC2 and Il17a⁺ γδ T cells. **E** Top: Spatial cell type interaction map derived by NiCo using a radius of 20. Normalized logistic regression coefficient cutoff *c* = 0.05. **F** Spatial neighborhoods (normalized logistic regression coefficients) for small duct (sd) BEC, cDC2, Il17a⁺ γδ T cells. Error bars indicate standard deviation of the coefficient estimates derived from five-fold cross-validation. **G** High-resolution spatial map containing segmentation stain information (left) or grouped transcript abundance (right) of γδ T cell- (*Cd163l1, 5830411N06Rik, Il23r, Trdv4*), T cell- (*Cd3e*), T17 polarization (*Irf4*),

cDC2B- (*Mgl2, Clec10a*) and BEC-related (*Epcam*) transcripts in a DDC D5 liver portal field of interest reflecting niche predictions in (**E**). Data derived from (**A**). **H** Left: TRDC and CD207 immunohistochemical staining on human PSC liver tissue (*n* = 18 PSC liver tissues). Peribiliary immune infiltrates are enriched for TRDC⁺ and CD207⁺ cells. Asterisk indicates bile duct. Scale bar, 40 μm. Middle: High- and low-resolution visualization of color-coded TRDC⁺ and CD207⁺ detection counts in human PSC tissue. Analysis indicates varying cell concentration within different tissue compartments. Scale bar, 250 μm. Right: Duplex TRDC/CLEC10A, TRDC/CD207 in-situ hybridization of human PSC liver tissue (*n* = 5 PSC liver tissues). Cellular colocalization and/or close proximity relationships between cells can be detected. Scale bar 10 μm. CEC continuous endothelial cell, cDC conventional dendritic cell, D day, DDC 3,5-Diethoxycarbonyl-1,4-Dihydrocollidine, Heps hepatocytes, HSC hepatic stellate cell, KC Kupffer cell, ld BEC large duct biliary epithelial cell, LSEC liver sinusoidal endothelial cell, lyEC lymphatic endothelial cell, Mac macrophage, mat. DC mature dendritic cell, Mono monocyte, Neutro neutrophil, NM not matched cell, pDC plasmacytoid dendritic cell, portal Fibs portal fibroblasts, PSC primary sclerosing cholangitis, sd BEC small duct biliary epithelial cell.

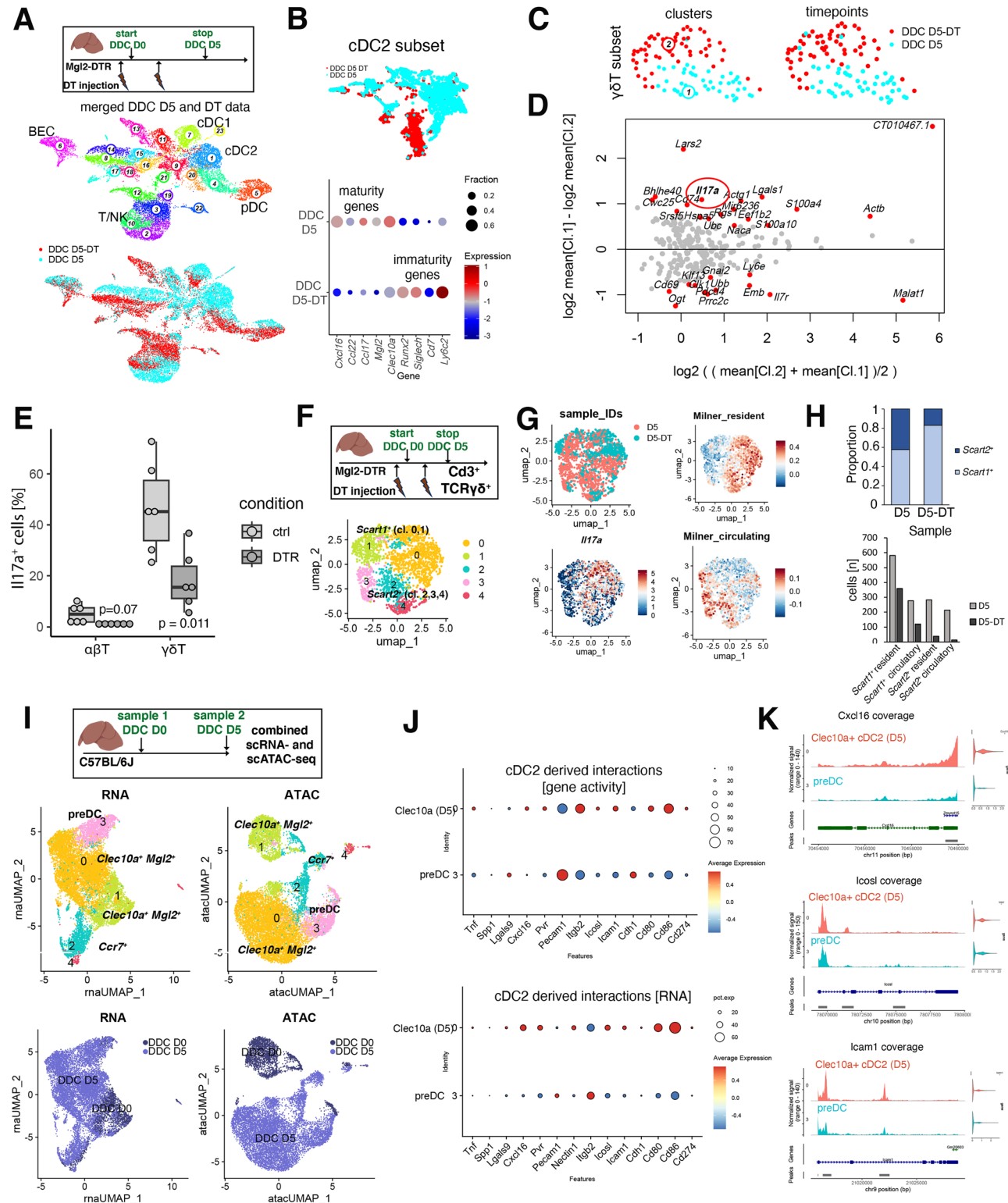

concordant cDC2 cell state regulation (Fig. 6I). While the Mgl2 locus did not exhibit obvious differential chromatin accessibility across cell states, the Ccr7 locus showed an increase in accessibility along the cDC2 differentiation trajectory (Fig. S10D).

This prompted us to test the genomic accessibility of identified DC-γδ T cell ligand-receptor pairs (Fig. S6), to screen for interactions that are epigenetically regulated and may explain reduced cDC2-γδ T cell interactions after mature cDC2B depletion. Multiple predicted interactions as part of the cDC2-γδ T cell communication node showed

concordant reduced genomic accessibility and RNA expression in preDCs compared to *Clec10a*+ cDC2 (Figs. 6J and S10E). Individual coverage plots of key cDC2-derived signal cues exemplified reduced genomic accessibility in preDC compared to *Clec10a*+ cDC2 (Figs. 6K and S10F).

Taken together, these data propose perturbed circulatory and residency properties of *Scart1*+ and *Scart2*+ γδ T17 cells and reduced γδ T17 effector differentiation upon cDC2B depletion, partially due to reduced genomic accessibility of γδ T cell ligands within preDCs.

**Fig. 6 | Early cDC2B depletion results in reduced γδ T17 effector differentiation.** **A** Top: experimental design and structure of the merged DDC D5 and DDC D5-DT data. Bottom: UMAP representation of isolated cell types and samples. **B** Top: UMAP representation of cDC2 subset from data in (**A**) (top). Bottom: Dotplot displaying expression of immaturity and maturity markers in cDC2. Log-normalized gene expression is color coded and fraction of cells expressing the gene is encoded by dot size (bottom). **C** UMAP representation of γδ T cell subset clusters (left) and samples (right) of merged DDC D5 and DDC D5-DT data. **D** Differential gene expression analysis of DDC D5 and DDC D5-DT derived γδ T cells. Red circle highlights *Il17a*. Significantly differentially expressed genes (adjusted *p* < 0.05, Methods) are highlighted in red. **E** Boxplot displaying frequency of Il17a⁺ αβ and γδ T cells in DDC D5 and DDC D5-DT measured by FACS. Two-sided *t*-test (three independent experiments, *n* = 6 mice/group). **F** Top: Experimental design of multiplexed liver γδ T cell experiment (DDC D5 and DDC D5-DT). Bottom: UMAP representation of γδ T17 clusters. **G** UMAP representation of multiplexed γδ T17 samples and log-normalized expression of *Il17a* and resident and circulatory gene signatures from ref. 41. **H** Top: stacked barplot displaying sample-specific proportions of *Scart1*⁺ and *Scart2*⁺ γδ T cell subsets in DDC D5/DDC D5-DT data. Bottom: absolute numbers of *Scart1*⁺ and *Scart2*⁺ resident and circulatory populations in DDC D5/DDC D5-DT data. **I** Experimental design (top). UMAP of merged control and DDC D5 derived scRNA-seq and scATAC-seq data modalities highlighting clusters derived for the data modality (middle), and sample identity (bottom). **J** Dotplot displaying cDC2-derived γδ T cell interacting gene activities (top) and RNA expression (bottom) across clusters 3 (preDC) and 0 (*Clec10a*⁺ cells). Log-normalized mean expression is color-coded and fraction of cells expressing the gene is encoded by dot size. **K** Coverage plots displaying gene-specific pseudobulk accessibility tracks in preDC (cluster 3) and *Clec10a*⁺ cells (cluster 0). Genomic regions of interest and coordinates are displayed in the bottom row. Violin plots on the right display RNA expression of the corresponding gene. ATAC assay for transposase-accessible chromatin, BEC biliary epithelial cell, cDC conventional dendritic cell, D day, DDC 3,5-Diethoxycarbonyl-1,4-Dihydrocollidine, DT(R) diphtheria toxin (receptor), pDC plasmacytoid dendritic cell.

## γδ T17 cell state induction by Mgl2⁺ cDC2B in liver draining lymph nodes

After the identification of a γδ T17 instructive role of cDC2B, we asked whether mature *Mgl2*⁺ *Ccr7*⁺ cDC2B induce Il17 production in liver-draining *lymph nodes* (LN), thereby promoting Th17/γδ T17 responses in an organ-site independent manner[42] (Fig. 7A). To enable the identification of *Mgl2*⁺ *Ccr7*⁺ cDC2 undeterred by the maturation-associated mRNA expression loss of typical DC markers[27], we performed CITE-seq on liver-draining LN at steady-state (D0), DDC D5 and D19 (19,338 cells; Fig. 7A, B). This permitted a subclassification of cDC1/2 in LN on the protein level since proteins have a much longer half-life than RNA. Indeed, the *Ccr7^high* DC cluster in the liver-draining LN dataset exhibits a clear separation into Cd11b⁺/Mgl2⁺ and Xcr1⁺ cells on the protein level, while the cDC2 subset displayed the loss of RNA marker genes (Figs. 7C and S11A). Raw numbers of CD301b⁺ cDC2Bs in the process of DDC-associated inflammation continuously increased in the LN scRNA-seq data mirroring liver egress (Fig. 7B–D). In liver draining LN, Mgl2⁺ cDC2 were in physical contact with γδ T cells, as seen in Tcrd-GDL mice (Fig. 7E). Contact and ligand-receptor-mediated interaction analysis confirmed the previously observed DC - γδ T cell signaling module (Fig. 7F, G). Again, multiple liver interactions including Icosl-Icos, Cxcl16-Cxcr6, Icam1-Itgal/Spn were also predicted in LN data promoting the idea of site-independent cDC2 - γδ T cell interactions in cholestatic liver disease. Next, we analyzed a scRNA-seq dataset containing γδ T17 cells derived from liver draining LN of D0 and D5 and detected *Ccr7*⁺ *Sell*⁺ *Lck*⁺ (cluster 3), *Scart1*⁺ (cluster 0) and *Scart2*⁺ (clusters 1,2) γδ T17 cells. We performed SCENIC analysis to identify TF networks uniquely associated with these LN γδ T17 cells (Fig. 7H–J). A unique cassette of TF-regulons with predicted activity restricted to *Ccr7*⁺ *Sell*⁺ γδ T17 cells ("recruited", cluster 3) was identified, with the highest rss for the TFs Myb, Lef1, Bptf and Ikzf2, in accordance with the RNA expression levels (Figs. 7H and S11B). Altered γδ T17 frequencies in LN prompted us to analyze γδ T in the circulation. Indeed, FACS analysis of circulating γδ T cells from ctrl (D0) and DDC D5 timepoints revealed altered γδ T dynamics, with a decrease of naïve Cd62l⁺ Cd44⁻ and Cd62l⁺ Cd44⁺ circulatory γδ T cells, while Cd44⁺ Cd62l⁻ effector γδ T cells increased (Figs. 7K and S11C). Last, we compared γδ T17 cells derived from draining LN at DDC D5 and upon loss of *Mgl2*⁺ cDC2B at DDC D5-DT. γδ T17 populations derived from the D5-DT were almost completely abrogated in our scRNA-seq data, implying a connection of LN γδ T17 effector differentiation and cDC2 homing (Fig. S11D–F).

Altogether, we here characterized a communication network between cDC2 and γδ T cells (Fig. 7L), where altered cDC2B dynamics impact on γδ T17 effector differentiation in liver and draining LN, exerting a profibrogenic function in liver cholangitis.

## Discussion

While a number of immunological susceptibility genes have been identified in PSC[43], the disease-aggravating and self-perpetuating biliary niche- and injury-specific immunological circuits that maintain and drive disease pathology remain incompletely understood. The DDC diet model used in this study, may mimic certain aspects of cholestasis and biliary injury in human PSC and how these local cues may amplify local tissue injury responses.

Il17 has been identified as a key driver of liver disease and fibrosis[44], yet γδ T cell-derived Il17a as an important cellular source has been widely neglected. While our data hints towards a profibrogenic role of γδ T cell-derived Il17a, γδ T cell-specific knockout models will be required to fully elucidate the role of these cells in liver scarring and exclude potential DT-mediated off-target effects affecting our analysis.

To understand the heterotypic interplay of the biliary niche including cDC2 interacting with γδ T cells, we here generated a DDC liver and LN-specific spatial and single-cell reference. Local and distant γδ T17 responses were mediated by *Scart1*⁺ and *Scart2*⁺ γδ T cell sub-populations, which have been described to exhibit both resident and circulating properties. Although antigen processing and presentation by MHCs are thought to be dispensable for γδ T cell activation[45,46], our results show that antigen-presenting cDC2B within the immunobiliary niche may contribute to a γδ T17 response, e.g. through contact-dependent signals such as Icosl-Icos interaction. While these results need to be confirmed with independent and possibly more cell-specific depletion models, our data imply that Icosl and additional ligands derived from cDC2 may promote γδ T17 cell states, while expression of these ligands in cDC2 was partially regulated by genomic accessibility.

The here inferred TF regulon activity extends known concepts of divergent γδ T17 populations and their transcriptional regulation in disease[47]. While γδ T cell landscapes in multi-organ settings have revealed site-dependent states[48], our study highlights functional γδ T responses in primary organ and the tissue-draining LN. For example, the chemokine receptor Ccr6 known to be essential for αβ T and γδ T cell positioning in liver fibrosis[49,50] was expressed in *Scart2*⁺ but not *Scart1*⁺ γδ T cells. *Scart2*⁺ cells also express S1pr1 and S1pr2 for which opposing functions for regulating tissue residency and egression have been described in skin γδ T cells[51,52]. Our data imply a role for *Scart2*⁺ γδ T cell recruitment in cholangitis, while resident *Scart1*⁺ γδ T cells were the main Il17a producers. Despite these phenotypic differences, γδ T17 cells showed a dependence on cDC2B across subtypes.

The presence of naïve *Sell*⁺ *Ccr7*⁺ γδ T cells in LN at steady-state and early disease (D5) may imply, that γδ T cell niche replenishment from the circulation depends on restorative signal cues, as these cells contribute to tissue regeneration and scarring[53].

Context-dependent signal input from neighboring cells as part of cellular crosstalk may affect cDC2B tissue retention and γδ T17 differentiation in liver and LN. The biliary tree contains a unique

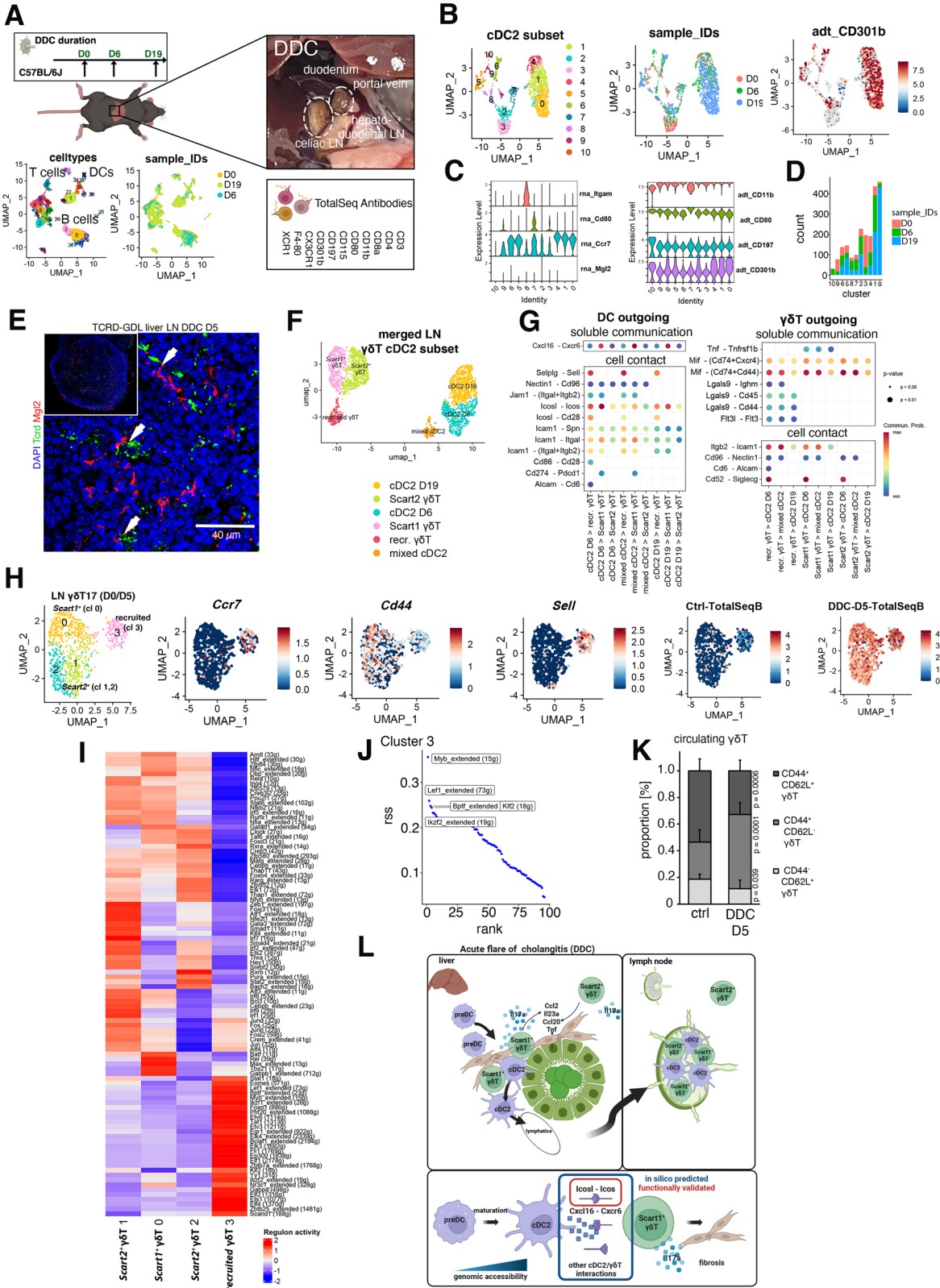

immunological niche[54], and our DDC and PSC data corroborate such multi-dimensional molecular crosstalk through spatial and in silico validation of immunobiliary myeloid (e.g., via Csf1, Ccl2, Mif, Lgals9) and γδ T cell (e.g., via Il18, Cxcl16, Icosl) interactions. However, UTC composition and heterogeneity may not be fully conserved across species[55], yet according to the functional unit-concept of UTC, partial

transferability of the cDC2-γ T cell interactions from mouse to human within UTC subtypes will be interesting to further resolve[56].

Moreover, altered γδ T cell dynamics were detectable in different mouse models of cholestatic liver injury (DDC, CDE, BDL), indicating a more general disease response. More profound analysis of all γδ T cell states, spatial and multimodal analyses are required to get a more

**Fig. 7 | γδT17 cells and Mgl2⁺ cDC2B in liver draining lymph nodes. A** Experimental design of the merged liver draining LN control, DDC D6 and DDC D19 CITE-seq dataset. Macroscopic view of the hepatoduodenal and celiac trunk lymph nodes in situ[42]. **B** UMAP representation of the LN-derived cDC2 clusters (left), timepoints (right) and centered log-ratio transformed CD301b (encoded by *Mgl2*) protein expression. **C** Violin plots displaying log-normalized RNA expression for *Itgam*, *Cd80*, *Ccr7* and *Mgl2* (left) and centered log-ratio transformed protein expression for the corresponding proteins (right). **D** Barplot displaying timepoint-specific absolute cell numbers of cDC2 in LN. **E** IF displaying DAPI, Mgl2 co-stained liver-draining LN of DDC D5-treated Tcrd-GDL mice. Scale bar, 40 μm (*n* = 3 mice). **F** UMAP representation of merged LN γδ T cell and cDC2 subset displaying clusters used for cell communication inference. **G** Dotplot displaying cDC2 and γδ T cell outgoing molecular contact-based and soluble communication pairs. Communication probability encoded as color code and *p* values as dot size. Statistics

derived from permutation test. **H** UMAP representations displaying clusters of LN γδ T17 cells derived from DDC D0/D5 timepoints, log-normalized *Ccr7*, *Sell*, *Cd44* expression and centered log-ratio transformed data of sample hashtags. **I** Heatmap displaying SCENIC-inferred TF regulon activity in *Scart1*⁺ (cluster 0), *Scart2*⁺ (clusters 1,2) and recruited (cluster 3) γδ T17 clusters shown in (**H**). Predicted TF regulon activity is displayed as a color code. **J** Rank visualization of recruited γδ T17 cluster-specific regulon specificity scores (rss). Top 5 regulons are indicated. **K** Stacked barplot displaying relative proportions of CD44⁺ CD62L⁺ (central memory-like), CD44⁺ CD62L⁻ (effector) and CD44⁻ CD62L⁺ (naïve) circulating γδ T cell populations. Two-sided *t*-test (3 independent experiments, ctrl: *n* = 8 mice, DDC D5: *n* = 7 mice). Data are presented as mean values and error bars indicate standard deviation. **L** Graphical Abstract summarizing key findings of the study. adt antibody-derived tag, cDC conventional dendritic cell, D day, DDC 3,5-Diethoxycarbonyl-1,4-Dihydrocollidine, LN lymph node, rss regulon specificity score.

holistic understanding about potential disease-specific mechanisms. In that regard, our DDC atlas also captures naïve γδ T, γδ T1 cells, and cellular dynamics of other disease promoting myeloid cells, which should be further explored[10,57,58].

Altogether, our site- and time-resolved cholangitis atlas provides novel insights into heterocellular interactions of under-explored immune cells at unprecedented resolution and captures the dynamics of hepatic cDC2 and γδ T cells within the immunobiliary niche.

## Methods

### Mouse models
All animal experiments were authorized by the German Regional Council of Baden-Württemberg and Bavaria (Regierungspräsidium Freiburg and Regierung Unterfranken, reference numbers G20/164, 2-1554). Mice were bred in a 12-h light/dark cycle with free access to water and food and kept under defined pathogen free conditions with littermates receiving control chow being kept and housed in separate cages in the same facility room (Tables S2 and S3). Ambient temperature ranged between 20–24 °C and the humidity range was 30–70%. C57BL/6J were bought from Janvier (Saint-Berthevin Cedex, France). Il17a/f-KO[59] and Tcrd-KO (Tcrdtm1, JAX:002120) mice[60] were obtained from Wolfgang Kastenmüller. Tcrd-GDL mice (JAX:038040)[17] were obtained from Martin Väth. Mgl2-DTR (JAX:023822)[16] were obtained from Kristin Hogquist with permission from Akiko Iwasaki. Chow supplemented with 0.1% 3,5-Diethoxycarbonyl-1,4-Dihydrocollidine (DDC) or chow deficient in choline was purchased at ssniff (Soest, Germany)[14]. For CDE diet experiments, mice received a chow deficient in choline and 0.05% ethionine-supplemented drinking water. Prior to organ collection, mice were killed by cervical dislocation or $CO_2$.

For scRNA-seq experiments, male mice were used. For FACS analysis, male mice were used with the exception of the cDC1/2 and cDC2A/B quantifications, where female mice were used.

### Isolation of cells, FACS, and GEM generation
Primary liver cells were isolated from 6–12-week-old C57BL/6J mice by a Collagenase D digest[61]. Prior to FACS sorting, cells of interest were magnetically enriched using a magnetic separator and antibody- or Streptavidin-conjugated microbeads (Miltenyi Biotec GmbH, Bergisch Gladbach, Germany). Isolates of hepatic stellate cells (HSCs) were obtained as described elsewhere[62]. Mouse hepatoduodenal and celiac lymph nodes[42] were excised, and digested in 2 mg/ml Collagenase D for 30 min at 37 °C. Live/dead exclusion in FACS was performed using Zombie-NIR fixable viability dye (Biolegend, San Diego, USA) in combination with antibodies/reagents (Tables S4–6) to sort the cell populations at equal ratios using a FACS Aria III cell sorter. Information regarding the composition of individual cell populations in each dataset can be found below. GEM generation and library preparation

was performed according to the manufacturer's protocol (10x Genomics, Pleasanton, USA). Quality control of libraries was performed using Agilent Bioanalyser High Sensitivity DNA Chips. Single and dual index libraries were sequenced on a NovaSeq S1 or NextSeq 2000 with a read-depth of 40,000 reads/cell. FASTQ files were mapped to a reference using kallisto[63] (DDC atlas data). CellRanger was used for mapping of transcriptome, antibody-derived tags and human data. Combined scRNA- and scATAC-seq (10x Multiome) was performed as described in the demonstrated protocol for nuclei isolation for single cell Multiome ATAC + Gene Expression Sequencing (CG000365.Rev C). Digitonin treatment was performed for 210 s, nuclei suspensions washed, quantified and loaded at a concentration between 3000–8000 nuclei/μl to target a recovery of 10,000 nuclei. Before proceeding to GEM generation, a transposition reaction was performed on the isolated nuclei, where transposase enzymes fragmented accessible chromatin regions and inserted sequencing adapters. Nuclei were then loaded into the 10x Genomics Chromium Controller on a Chip J, where they were partitioned into Gel Bead-In Emulsions (GEMs), enabling the simultaneous capture of gene expression (GEX) and chromatin accessibility data from the same nuclei. Within the GEMs, reverse transcription converted mRNA into barcoded cDNA, unique to each nucleus. Following GEM breaking, the cDNA was recovered and amplified, alongside the tagmented DNA fragments from the ATAC reaction. Both the GEX and ATAC libraries were then purified, amplified, and quantified separately. Finally, these libraries were sequenced on separate flow cells, as the GEX and ATAC libraries require different sequencing parameters. Transposition and Chip loading for Multiome experiments were conducted at the Single Cell Center of the University of Würzburg.

Cell enrichment strategies for individual scRNA-seq datasets:
- Liver steady-state and DDC-dataset: to enrich all relevant cell types at equal ratios, two independent rounds of cell isolation and sequencing were performed at day 0 (ctrl), D3, D5, D9, D18 or D19, D25. For STOP and D5-DT data, only protocol B was used. D18 and D19 timepoints were summarized as D19 to facilitate timepoint-specific interpretation within the atlas. The following cell populations were sorted: (1) alive singlet, Cd45⁺, Lyve-1⁺, Pdpn⁺, Epcam⁺, Cd31⁺ Lyve-1⁻, retinoid-autofluorescent cells (405 nm excitation, 450/40 emission)[62]. (2) alive cells from HSC prep, Cd45⁻ Cd11c⁺ Cd64⁻, Epcam⁺, Cd3e⁺ Lyve-1⁻.
- Mouse CITE-seq LN dataset (ctrl and DDC): the following cell types were enriched from pooled hepatoduodenal and celiac LNs at D0, D6, and D19 and mixed at equal ratios: alive singlet, Cd45⁺ Cd11c⁻, Cd45⁺ Cd11c⁺, Pdpn⁺, Cd45⁻ Pdpn⁻.
- γδ T cell datasets: cells from liver and LN were labeled using hashtag antibodies and alive Cd45⁺ Cd3e⁺ TCRβ⁻ TCRγδ⁺ cells were sorted at indicated disease timepoints. The relative yield of γδ T cells at every timepoint represents the fraction of sorted cell numbers per mouse and timepoint.

- Human liver dataset: non-cholestatic disease controls and far distant human liver tissue was used for cell isolation from 6 patients (4 female, 2 male, age range 45–76 years, receiving a lobectomy or hemihepatectomy for oncologic resection) and the following cell populations were FACS enriched at equal distribution: (1) alive, (2) CD45+ CD3E−, (3) CD45+ CD3E+, (4) CD45+ CD11C+, (5) CLEC4G+, (6) CLEC4G− CD31+, (7) EPCAM+. This gating strategy was applied to enrich for immune cells of the immunobiliary niche, while the specific localization of the respective cells was not further assessed. Cell types of the human atlas were clustered at low resolution, so that LAM did not cluster separately, but were part of the FCGR3+ cluster. Lymphatic endothelial cells and portal fibroblasts, as important cell types of the portal/immunobiliary niche, were not actively enriched for in our sorting strategy and are thus not part of the dataset.

## Bioinformatic analysis of scRNA-seq data

For the initial temporal DDC dataset (Fig. 1), a threshold of 1500 transcripts for mouse and 1000 transcripts for human datasets was used. VarID2[28] was run with the following parameters: mintotal = 1500, minexpr = 5, minnumber = 5, ccor = 0.4. FGenes was used to remove genes without a gene symbol, mitochondrial and ribosomal genes. The pruneKnn function was run with the parameters: large=TRUE, regNB=TRUE, knn=25, and Leiden clustering performed. Cluster identity subsets used for reclustering individual cell types are listed in the results section. For the quantification of DC subsets overall (Fig. 3A), a data subset of the clusters 5, 15, 17, 18, 19, 24 was reclustered to quantify DC subsets across timepoints.

Differential gene expression analysis was conducted using the RaceID3 algorithm. Based on a background model of transcript count distribution, a negative binomial distribution across clusters was inferred. Multiple testing corrected adjusted $p$ values (Benjamini–Hochberg) were then estimated across clusters. For the analysis of Xenium data, we used publicly available human liver control data (Table S7). Xenium data were processed within Seurat and principal components, UMAP, community detection and clustering information obtained. For the inference of human suspension-based liver scRNA-seq data, we used robust cell type decomposition (RCTD) as part of the spacexr package[18]. For the identification of common cellular neighborhoods, we used Seurat's "BuildNicheAssay", which identifies similarities based on k-means clustering of neighborhood data (neighbors.k = 15). Transition probabilities within cDC2 subsets were calculated using the plotTrProbs function of VarID2[28]. Pseudotemporal ordering of self-organizing maps (SOMs) was performed using FateID[29]. Signac[64] was used for the analysis of merged D0, DDC D5 Multiome data. RNA and ATAC data were tested for quality metrics and clustered based on information within each modality. Predicted gene activities levels were compared to RNA expression within transcriptome-based dimensionality-reduced space.

Seurat's ability was used to integrate multimodal data, including antibody-derived tag (ADT) data[65]. ADT data was transformed using centered log ratio transformation (CLR) and ADT data displayed within the gene expression space. SCENIC[36] was used to infer transcription factor regulon activity. Regulon activity per cell type was visualized in a heatmap and regulon specificity scores (rss) calculated using the inbuild calcRSS function. Pathway enrichment analysis was performed using ReactomePA[66]. cDC2A/B gene signatures were derived by curation from ref. 13. Residency and circulatory gene signatures[40,41] were derived from[48]. Vγ4 andVγ6 gene signatures were calculated by including the top 50 differentially expressed genes within a Vγ4 cluster (1) and Vγ6 cluster (9) of a TCR repertoire information containing reference[48]. For Niche Covariation (NiCo) analysis[24] of the human liver Xenium dataset, we utilized NiCo's annotation module with spatial guide clustering resolution 0.5. For the calculation of niche interaction maps, cell type enrichment ratios within cellular neighborhoods of a central cell were calculated, and a logistic regression to predict central cell type identities from these ratios was performed. The regression coefficients were then used to reconstruct the interaction map. Inferred latent factors from expression data were used to identify a small number of gene programs explaining cell state variability of each cell type. Covariation analysis of these latent factors across co-localized cell types may thus capture the effect of cell-cell interactions on gene program activity[24].

More detailed information about the bioinformatic methods and access to the deposited scRNA-seq data can be found in the supplementary information (Tables S7 and 8).

## Generation of DDC spatial transcriptomics data and analysis

For the generation of the scRNA-seq data reference for NiCo analysis, the DDC complete data (Fig. 1J) were merged with exclusion of cluster 11 (low QC) with the hepatic γδ T cell data containing both γδ T 1 and γδ T17 subsets (Fig. 4A). Merged data were clustered and annotated using Seurat (resolution 0.8) and doublet clusters (cl. 22, 31, 35, 36), clusters of poor quality and <200 cells removed (cl. 39, 40). This resulted in a scRNA-seq reference containing 81,168 cells across 26 cell types and states.

For imaging-based spatial transcriptomics, the Xenium platform (10x Genomics) was used together with the Xenium Mouse Atlassing panel and a customized 100-gene add-on panel (Table S1) to further resolve liver and immune cell subsets. The experimental protocol followed the manufacturer's recommendations (CG000580, CG000749, and CG000584). The Cell Segmentation Staining Reagents (PN-1000661) were used to generate segmentation data based on a combination of cell boundary and intracellular markers. Step-wise image data acquisition was conducted after the DDC tissue region of interest was marked using the in-built Xenium Analyzer. Low-quality transcripts were filtered out using the framework of the Xenium analysis software, and transcript identities were annotated based on the codebook information. $Q$ scores representing transcript identity confidence and quality were calculated, and a $Q$ score cutoff was chosen as 0.2.

A more detailed experimental and bioinformatic description of the ST workflow can be found here[67].

## Histochemical and immunohistochemical analysis of human and mouse liver tissues

Three μm thick sections of a previously characterized human tissue microarray (TMA) with a core size of 1.5 mm that contained 78 non-cholestatic disease control liver tissues[19,20] was immunohistochemically stained for CK7, CK19, CD34, PDPN, CD1A, CD3, CD21, CD23, TRDC and CD207 (Table S6). Non-cholestatic liver disease controls stained negative for CD1A, CD21, and CD23. Antigen retrieval was performed using Ventana CC1 solution and primary antibodies were incubated 24–40 min. DAB solution was added after linker- and multi-incubation and sections counterstained with haematoxylin. All used antibodies can be found in the Supplementary Table 6. Antibodies were tested prior to use according to the manufacturer's recommendations. For proteogenomics, 1 μg of labeled antibodies was used per 2 million cells. TMA slides were scanned using a Pannoramic Scan II scanning device (3DHISTECH, Budapest, Hungary) with a 40x magnification. Positive cell detection of liver tissues was performed within the in-built TMA dearrayer of QuPath after setting a single intensity threshold[5]. Correlation analysis of TMA cores was conducted using the corrplot package in R for stainings containing at least 20 datapoints. Human PSC liver tissues ($n = 18$) were stained against TRDC and CD207. RNAscope (Bio-Techne, Minneapolis, USA) duplex in situ hybridization of human PSC tissues was performed using commercially available probes against TRDC, CLEC10A, and CD207 according to the manufacturer's recommendations. Tissues were provided in accordance with the regulations of the Tissue Bank of the National

Center for Tumor Diseases (NCT) Heidelberg and the ethics committee of Heidelberg University (S-230/20, S-206, 207/05). The use of the fresh human resection material was approved under the ethics vote 2012-293N-MA and 2021-2320-1, including written prior patient consent. For immunohistochemical detection of mouse Mgl1/2, liver tissue antigen retrieval was performed at pH 6 in a steamer for 8 min, and Mgl1/2 antibody (#AF4297, R&D Systems, Minneapolis, USA) was incubated for 1 h at room temperature (1:75 dilution). Horse anti-goat IgG coupled alkaline phosphatase-based detection was performed for 7 min. For the detection of Pdgfrb in mouse liver tissues, sections were incubated 1 h with the primary antibody after heat-induced antigen retrieval at pH 9. An anti-rabbit secondary antibody conjugated to AP was applied (Polyview Plus AP reagent, ENZO Life Sciences GmbH, Lörrach, Germany) and the signal was visualized using Permanent AP Red (Zytovision GmbH, Bremerhaven, Germany).

•Dendritic and T cell isolation, FACS, cultivation, and semi-quantitative PCR

Cells were isolated from 6–12-week-old C57Bl/6J mice. After tissue digestion and gradient centrifugation, cells were enriched using antibody-coated microbeads and LS columns (Miltenyi Biotec, Bergisch Gladbach, Germany). A published DC gating strategy was used to detect and quantify cDC1 and cDC2[26] For in vitro cytokine re-stimulation, T cells were stimulated for 3 hrs with 50 ng/ml PMA (Merck, Darmstadt, Germany), 1 μg/ml ionomycin (Merck) and 1 μg/ml Brefeldin A (Merck). After re-stimulation, cells were surface-stained, fixed and permeabilized using FOXP3/Transcription Factor Staining Buffer Set (eBioscience, San Diego, USA) and stained against Il17a and data acquired using a FACSCelesta (BD, New Jersey, USA). For liver DC−splenic T cell coculture, hepatic DCs were isolated and magnetically enriched using CD11c microbeads and LS columns according to the manufacturer's recommendations. Simultaneously to hepatic DC isolation, splenic T cells were isolated in a two-step isolation procedure to enrich the fraction of γδ T cells. First, a negative selection against CD4+ and CD8+ conventional T cells was performed using CD4/CD8 (TIL) microbeads and LD columns (Miltenyi Biotec). The negative fraction was then enriched using CD3e microbeads and LS columns. Liver DCs and splenic T cells were seeded in a 1:1 ratio in RPMI 160 (Gibco) supplemented with 10% FCS (Corning, New York, USA) and penicillin/ streptomycin (Thermo Scientific, Waltham, USA) in addition with mouse recombinant Il23 (10 ng/μl, Thermo Scientific). For TCR stimulation, 24 wells were coated overnight with anti-CD3e antibody (5 μg/ml, Biolegend), washed 3 times with PBS, and enriched splenic T cells cultured for 20 h with DDC D5 bile, Lgals9, Il18, Il23a, Cxcl16, anti-mouse Icos-activating antibody or DC-conditioned medium derived from ctrl or DDC D5-derived liver DCs (mixed with RPMI containing 10% FCS and Pen/Strep in a 1:1 ratio). Mouse bile was isolated from gallbladders from DDC D5 mice, and bile was transferred into 1.5 ml tubes. A centrifugation step was performed to remove insoluble particles, and supernatants were used for stimulation assays (1:500 dilution).

Cells were harvested for FACS analysis 20 h after culture. For the analysis of circulating γδT cells, mouse blood was withdrawn by intracardiac puncture, and erythrocytes were lysed using ACK lysis buffer (Thermo Scientific) prior to FACS staining and data acquired using a Cytek Aurora (Cytek Biosciences, Fremont, USA).

For bulk RNA isolation from cells and tissues, the Arcturus PicoPure RNA isolation kit was used (Thermo Scientific, Waltham, USA). After cDNA synthesis using RevertAid First Strand cDNA Synthesis kit (Thermo Scientific), semiquantitative PCR was set up using SYBR Green (Bio-Rad, Hercules, USA) using 40 amplification cycles[61]. Melting curves were generated to test for product amplification specificity, and samples were measured in technical triplicates. Gene expression was normalized using the normaqPCR package and the measurement of four independent housekeepers (Actin, Gapdh, Hprt, Ppia)[7]. A list of intron-spanning primers used

for qPCR (GC content 40–60%) can be found in Supplementary Table 9.

## Sample preparation, fluorescence microscopy, and quantification

Mouse blood was withdrawn by intracardiac puncture using a heparinized 1 ml syringe and plasma obtained by centrifugation (30 min $2000 \times g$)[68]. Blood plasma-derived parameters were detected using a VE tube 30 device and Liver and Kidney Profile Disks (Mindray, Shenzhen, China). Snap-frozen liver tissues were cut into 5 μm thick tissue sections using a Leica CM 3050 S cryostat and fixed in ice-cold methanol. To block unspecific binding, tissue sections were incubated with goat-serum prior to incubation with primary and secondary antibodies. Sections were washed and mounted into Fluoromount G (Southern Biotech, Birmingham, USA) with or without DAPI. Six color immunofluorescence microscopy images were obtained using a Zeiss LSM780 laser scanning confocal microscope. Detectors were adjusted to detect 6 wavelength windows using 4 excitation laser wavelengths (405, 488, 561, 639 nm). The far-red laser was used for simultaneous excitation of AF639 and AF700 fluorescent dyes, which could be detected using wavelength-specific detection windows. The 405 nm laser was used for a combined excitation of BV421 and BV605. Image tiles were obtained using an oil immersion 40x objective and stitched based on a default overlap region of 10%.

Four-color IF images were obtained using an Olympus IX83 microscope containing a Yokogawa spinning disc unit.

To optimize image display, raw images were background subtracted (rolling ball, default settings), and an adjustment of the display window was performed using FIJI[8]. All images within one experiment were adjusted identically. Colocalization analysis of image channels and interaction areas was processed using the "math" function in FIJI. Average cell colocalization areas per group were calculated and displayed as a heatmap.

## Quantification and statistical analysis

For quantification and statistical analysis, R Studio was used, and data are presented as mean ± SD. Graphs were plotted using R Studio or Excel, and panels were organized using Adobe Photoshop and Adobe Illustrator. For information regarding independent repeats of experiments, please refer to the respective figure legends. Statistical tests are specified in the figure legends. Shapiro−Wilk test was used to test for normality. The Wilcoxon rank-sum test was used for non-normally distributed data. $T$-test was used for normally distributed data. ANOVA was used in combination with Tukey's test (post hoc). Boxplots display the median and the interquartile range (IQR). Whiskers extend to datapoints within 1.5 IQR. The number of animals per group in each experiment is indicated in the figure legends.

## Reporting summary

Further information on research design is available in the Nature Portfolio Reporting Summary linked to this article.

# Data availability

All generated single-cell sequencing data are available at GEO as raw and processed data under the following accession numbers: human liver data, GSE280852; mouse DDC atlas, GSE280985; Multiome data, GSE281196; CITE-seq/Hash-multiplexed data, GSE281197. DDC D5 spatial transcriptomics data is accessible at GSE311681. The human tissue microarray data cannot be made accessible due to institute-specific guidelines and human ethics regulations. Source data are provided with this paper.

# Code availability

No major novel code was generated for the data analysis in this manuscript. Please access Zenodo for workspaces/.rds files [https://

doi.org/10.5281/zenodo.16365892][69] and GitHub [https://doi.org/10.5281/zenodo.18667318][70] for code to reproduce/re-analyze data.

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

## Acknowledgements

The authors thank Konrad Schuldes, Sebastian Hobitz, Rémi Doucet Ladevèze, Wiebke Brethauer, Tim Rau, and Carolin Kerber for excellent technical assistance. We would like to thank Akiko Iwasaki, Kristin Hogquist, Wolfgang Kastenmüller, and Martin Väth for providing Mgl2-DTR, Il17a/f-KO, Tcrd-KO, and Tcrd-GDL mice. We would like to thank the Core Unit for FACS of the IZKF Würzburg and Christian Linden for supporting this study. We would like to thank the Center of Model Systems and Comparative Pathology (CMCP) and Christine Schmitt, Heike Conrad, Diana Lutz, and Karin Rebholz for their support. We would like to thank Sabine Roth from the Institute of Pathology in Würzburg. We would like to thank Haristi Gaitantzi, Vanessa Hartwig (Department of Surgery, Medical Faculty Mannheim), Emrullah Birgin (Department of Surgery, University Hospital Ulm), and Chang-Feng Chu (Leibniz Institute for Natural Product Research and Infection Biology, Jena) for their support with fresh human liver tissue. We thank the Single-cell Center Würzburg and Emmanuel Saliba for technical support in performing Xenium. We thank the Core Unit Systems Medicine and Panagiota Arampatzi, Mugdha Srivastava, Pierre Khoueiry, and Tom Graefenhahn. We thank the Deep Sequencing Facility at the Max Planck Institute for Immunobiology and Epigenetics (Chiara Bella and Ulrike Böhnisch) and Helmholtz Centre for Infection Research Braunschweig (Michael Jarek, Maren Scharfe, and Doris Järke). We thank Heike Wagner and Sabine Kranz, Stefan Brandt, Gallina Miller, Kim Patricia Rohr, and Denice Schmitt (Animal Facility of the Würzburg Institute of Systems Immunology). We want to thank Martin Väth, Christin Friedrich, and Milas Ugur for helpful discussions. Schemes within manuscript figures were generated with BioRender. This work was supported by the German Research Foundation (DFG) (SFB1583/1 Project #492620490, SPP1937 GA 2129/2-2, INST 93/1072-1 Project #471222118, INST 93/1164-1), by the European Research Council (ERC) (818846—ImmuNiche—ERC-2018-COG), by the Chan-Zuckerberg-Initiative (CZI) Seed Networks for the Human Cell Atlas, and by the Bundesministerium für Bildung und Forschung (BMBF) (TissueNet —031L0311A and CureFib—01EJ2201C), all to D.G. S.T. received a Single-Cell Seed Grant as part of the Single-Cell Seed Grant Initiative supported by the Bavarian State Ministry of Economic Affairs, Regional Development and Energy at the Helmholtz-Institute for RNA-based Infection Research implemented in the Single-Cell Center Würzburg.

## Author contributions

D.G. and S.T. conceived and coordinated the project. S.T. designed and performed experiments and computational analysis. H.H. and S.B. supported FACS analysis. H.H. performed D.C. in vitro experiments. A.A. supported bioinformatic analysis. J.S. provided support with mouse strains. N.V. and T.K. performed the spatial transcriptomics experiments. S. provided support for the design and analysis of T cell experiments. F.I. provided support with sequencing. T.P. contributed to immunohistochemistry experiments. M.T. and A.R. provided the PSC histological

assessment. K.B.H. and C.E.Z. supported human liver tissue analysis. T.P. and J.K. provided support with human scRNA-seq data. N.R. supervised K.B.H. D.G. supervised the project. S.T. and D.G. wrote the manuscript. All authors read and edited the manuscript.

## Funding

## Competing interests

D.G. serves on the scientific advisory board of Gordian Biotechnology. The remaining authors declare no competing interests.

## Additional information

[1]Würzburg Institute of Systems Immunology, Julius-Maximilians-Universität Würzburg, Würzburg, Germany. [2]Helmholtz Institute for RNA-based Infection Research (HIRI), Helmholtz-Center for Infection Research (HZI), Würzburg, Germany. [3]University Hospital Freiburg, Department of Gastroenterology, Hepatology, Endocrinology and Infectious Diseases, Freiburg, Germany. [4]Center for Model System and Comparative Pathology, Institute of Pathology, University Hospital Heidelberg, Heidelberg, Germany. [5]University Hospital Heidelberg, Institute of Pathology, Heidelberg, Germany. [6]Leibniz Institute for Natural Product Research and Infection Biology, Jena, Germany. [7]Friedrich Schiller University Jena, Jena, Germany. [8]Department of Pathology, University of Cambridge, Cambridge, UK. [9]I. Department of Medicine, University Medical Center Hamburg-Eppendorf, Hamburg, Germany. [10]Institute of Pathology, University of Würzburg, Würzburg, Germany. [11]Department of Surgery, Medical Faculty Mannheim, Heidelberg University, Mannheim, Germany. [12]Department of Surgery, University Hospital Ulm, Ulm, Germany. [13]CAIDAS—Center for Artificial Intelligence and Data Science, Würzburg, Germany. [14]Present address: Cancer Immunology Program, Peter MacCallum Cancer Centre, Melbourne, VIC, Australia. [15]Present address: Sir Peter MacCallum Department of Oncology, The University of Melbourne, Melbourne, VIC, Australia. [16]Present address: Center for Immunology, University of Minnesota, Minneapolis, MN, USA. ✉e-mail: dominic.gruen@uni-wuerzburg.de

