## [Transparent Peer Review file · Nature Communications]

An immunobiliary single-cell atlas resolves crosstalk of type 2 cDCs and gdT cells in cholangitis

Corresponding Author: Professor Dominic Gruen

Version 0:

Reviewer comments:

Reviewer #1

(Remarks to the Author)

Overall: Impressive detail and breadth of experiments that convincingly demonstrate a role for cDC-GD T cell axis in murine cholestatic-induced liver fibrosis. Methodologies are cutting-edge and sound. Unclear if unsupervised analysis however would pinpoint cDC-GD T cell interactome as main/important driver of biliary inflammation. Figures are generally excellent, informative and well organized but could be simplified to focus main messages. Supplementary is extensive and substantial text is devoted to describing supplementary results that should be reduced/consolidated to improve clarity and conciseness. Clinical features of patient material needs greater detail and number of patient material analyzed to validate cDC-GD T cell translational relevance is limited (2 PSC and no disease controls).

1. Abstract – slightly disjointed and lacking clear focus but could be dramatically improved with minor revisions. Authors state their goal is to elucidate unconventional T cells but then jump to mapping cell state trajectory of preDCs to cDC2B before returning to GD T cells and IL-17. Brief but better description/contextualization of abbreviated terms (e.g. preDCs, cDC2B, GD T cells, IL-17) would significantly improve readability and understanding for a broader readership.
2. Public data analyzed for Fig 1 should be cited in main text of manuscript and patients/pseudonormal material used to generate scRNA-seq reference should be briefly defined in main text with general clinical features of patients detailed in a table.
3. Relative and absolute number of GD T cells appears limited in biliary niche by scRNA-seq and IHC of human liver tissue. What is the relative proportion of GD T cells as a percentage of total T cells measured by scRNA-seq and IF staining for TRDC? Is the HE image for data in Fig 1a, Fig 1c available? Would be informative to include as small image overlaid on Fig 1a or in supplementary.
4. How many of the TMA samples were analyzed to confirm spatial localization of indicated markers in Fig 1d? What is the disease diagnosis of these patients? Is it surprising to observe indicated level of T cell infiltration (CD3) in 'healthy' liver? Same comment regarding periductal fibrosis in TRDC HE staining – why is it there if these samples are representative of 'pseudonormal' biliary microenvironment? Were any GD T cell markers (or T cell markers besides CD45) included in IF staining of steady state mouse liver to confirm presence of T cells/GD T cells around bile ducts?
5. Why were different protocols used to isolate cells for scRNA-seq? Assumption that FACS could not separate all subsets simultaneously but unclear as written.
6. Transitional text between scRNA-seq DDC atlas and subsequent subsetting of BEC cluster lacks clear rationale given that authors state they performed scRNA-seq to investigate unbiased dynamics of gene expression across the biliary niche. Difficult to discern what cell types and cell states change most dramatically during disease progression – did the unsupervised analysis point toward BEC or was this the subsetting done to answer defined hypothesis? Were there clear enrichment of cDC and GD T cell signals at any timepoint(s)?
7. Definition of latent factors and relevance of BEC and DC gene programs unclear as written and should be further explained for non-bioinformaticians. Difficult to interpret data shown in Fig 2F and Fig 2G.

8. Umap in Fig 3C be simplified (also split) to delineate cDC2A and cDC2B clusters? Similarly, could timepoint umap overlays in Fig 3D be consolidated into one umap with trajectory analysis shown by arrows to better indicate dynamics of cDC2 states at early to late to recovery timepoints?

9. How closely does cDC2 state at 'STOP' compare to cDC2 markers at steady-state (i.e. D0)? Including numbers of mice/cells analyzed by scRNA-seq (Fig 3C and 3G) would aid intra and inter comparisons of cluster sizes. Similarly, dot overlays representing number of analyzed mice should be included for Fig 3B and 3F bar graph.

10. From data in S5, appears Il17a, Rorc and Rora transcriptomic signatures are relatively weak (expression less than 1 LF above comparator). What is comparator here, D0? As stated by the authors, abundance of GD T cells are very low - are the receptor-ligand interactions known to facilitate recruitment and activation with cDC2 exclusive to GD T cells?

11. What are the absolute numbers of conventional Il17a+ T cells relative to Il17a+ GD T cells? Although a greater proportion of GD T cells appear Il17a, their overall scarcity (Figure S7A) questions their importance in cholestatic fibrosis.

12. How did DT depletion of cDC2B impact other immune subsets, particularly other T cell populations irrespective of Il17a expression?

13. Authors repeatedly attempt to translate mouse findings to PSC however few PSC samples were analyzed (n=2) and appropriate disease controls for biliary inflammation (e.g. PBC) and general end-stage liver fibrosis (e.g. ALD, MASH) are lacking.

14. DDC and BDL models of cholestatic liver injury were used to study GD T cells but alternative models of non-cholestatic liver injury used to study cDC1/2 dynamics (e.g. CDE diet) were not included as controls, which would help determine if relevance of GD T cells are truly specific to cholestatic disease or general liver disease feature. Use of GD T cell knockouts would clearly identify their role in cholestatic fibrosis as well as education of cDC into inflammatory or resolving cell states.

Reviewer #2

(Remarks to the Author)

This is an interesting study by a well established group in the field presenting a multimodal single-cell atlas of the immunobiliary niche in mouse cholangitis, complemented by human data. The authors highlight dynamic cDC2B maturation, $\gamma\delta$ T17 induction, and functional relevance of the ICOSL-ICOS axis. The dataset is rich, and the study addresses an important and relatively unexplored area. However, in its current form, several claims appear overstated or insufficiently supported by the presented data. The study is more descriptive than mechanistic, with limited causality in validations. Human translations are preliminary, and several interpretive issues need clarification. Please find in the following my points.

Spatial analyses (Fig. 1)

It is difficult to appreciate whether CD3E+ cells are truly enriched around ducts/large vessels. Quantification of their localization would strengthen this point. The density map for CD207 is unclear; why was a distinct visualization chosen for this marker only? If biliary epithelial cells (BECs) and immune cells were sorted separately, why are BECs so underrepresented relative to immune cells? What input ratios were used when loading the cells? Many proteins are inconsistently formatted (e.g., some are capitalized, others not). Please standardize (e.g., ITGAX+ cells). Figure legends indicate n=3, yet there are no quantitative analyses. Including barplots would help. Annotation strategy needs clearer explanation. A dotplot of marker genes validating cluster identities would be welcome. Timepoints in legends could be reordered chronologically for clarity. The rationale for using t-SNE (rather than UMAP throughout) is not clear.

cDC2 trajectory and cluster interpretation (Fig. 2-3)

The claim that cluster 8 expresses chemotaxis/adhesion genes is not entirely convincing. Cluster 16 seems to express more of these genes except Il23a. Exclusion of clusters 3, 12, 15 due to residual cDC1 signatures requires more transparency — which genes drove this decision, and are they uniquely cDC1-specific? There is some confusion regarding Mgl2 expression: overall Mgl2+ cells increase, yet the proportion of Mgl2+ cDC2B decreases. This apparent contradiction should be clarified. There seems to be a discrepancy in the interpretation of DC recruitment. The authors claim that Mgl2- preDCs are recruited along disease progression, yet their own data (e.g., Fig. 3E) show an increase in Mgl2+ cells, which they themselves describe as the more immature DCs. This would rather suggest that Mgl2+ precursors are recruited into the niche. The authors should clarify this trajectory: are Mgl2+ cells the recruited precursors that then lose Mgl2 as they mature, or are truly Mgl2- preDCs entering the tissue?

Human validation

The presence of IL17+ $\gamma\delta$ T cells in human PSC is implied but not convincingly shown. Given ongoing debate about human $\gamma\delta$ T17, this claim should be toned down or better validated. The human validation of $\gamma\delta$ T17 identity remains unclear in their data. Do human $\gamma\delta$ T cells in PSC express canonical Th17-defining transcription factors (e.g., RORC, MAF, IRF4) and cytokines (IL17A, IL17F)? The current data (Suppl. Fig. S7H-J) show no IL17A transcript levels, and no clear evidence for a broader Th17 signature in human $\gamma\delta$ T cells. Since the existence of human $\gamma\delta$ T17 cells remains controversial, stronger

evidence or a more cautious interpretation is needed. At minimum, the authors should explicitly show whether RORC, MAF, IL17A/F are expressed in their human $\gamma\delta$ T cell subset. Cross-species conservation of interactions is intriguing but may be overstated.

Reviewer #3

(Remarks to the Author)

In this manuscript, the authors create a scientific narrative demonstrating the importance of type 2 cDCs and gamma delta T cells. The majority of which is described in a mouse model of biliary injury using a range of high level scientific techniques.

Obviously, the role of type 2 cDCs reported here is not a new one, and it was first described in a range of mouse models by Müller AL et al (J Hepatol. 2022 Dec;77(6):1532-154) (which is acknowledged in the references. Therefore, the novelty of this study is in the potential cross talk with the gamma delta cell population in the biliary niche. In doing so it provides mechanistic insight to the role of dendritic cells in biliary injury and potential ligands that could be targeted for drug discovery.

Overall, the authors have submitted a large amount of data to provide robustness to the proposed narrative. However I would say that within the large volume of experimental data – which skips between human and murine models, there is perhaps a distraction for the reader particularly for the reasons I highlight below.

My major comments for this manuscript include

1) I think the closeness of this model (ie mirroring to PSC) requires firstly some careful clarification by the authors. PSC is an autoimmune liver disease of the liver and colon. Its genetic risk is HLA driven, and the importance of CD4+/CD8+ and T regs is well described. The inflammatory response is therefore the driver of biliary injury in PSC. This is in strong contrast to the mouse model studied here, where the induced epithelial/cholangiocyte damage secondarily induces inflammation – therefore they are not one in the same. The DCC model is therefore not a mouse model of PSC. Perhaps the authors should therefore highlight that what these have in common, and the point of their study – ie how biliary injury may amplify the local immune response and perturbate epithelial resultant injury.

2) In the opening introduction they say PSC ultimately leads to transplant – this is not true – 50% are alive and well at 20 years. I would not say that the disease is characterised by dysfunctional myeloid responses. Therefore, to compare PSC to a toxin mediated liver injury, needs careful discussion by the authors and questions why they wish to try and reproduce some of their findings in PSC.

3) the authors show that CD1C and CD3E transcripts are present near large vessels and biliary tracts (Fig1A). I must admit – I think here spatial has been used for spatial sake. The figure presented loses the clarity needed that would be gained by a simple HE that has been stained for these markers at low and high power, which would then show a broad range of zones 1-3 of the liver, portal tracts and central veins – to really understand the normal distribution of these cells in a normal liver. Secondly, I do not really understand how you can claim that the pseudo normal liver dataset is the biliary niche. The immune sets that have been fax sorted could come from anywhere. Why do you claim they are totally representative of the biliary niche?. However, I expect that the BuildNiche assay suggests a biliary and lobular niche from the generated resultant dataset.

4) The authors then describe once again that have generated a single cell RNA seq from DDC fed mice at days 3,5,9 and 25. Once again there is a repeated claim that all these cells are from the immunobiliary niche – once again they could come from any compartment of the liver (Portal, Z1,2,3 and central) – I think this is slightly misleading for non-mouse hepatic experts.

5) The authors then demonstrate that the small ducts in portal fields during DCC injury release a range of inflammatory genes highlighting these are chemoattractant to T and DC cells – as they are for many cells, so this in itself is not proof of a direct interaction. Therefore, the specific importance of these observations can only be generalised. The authors then highlight similar findings in the BDL model but not the CCL4 model – which is to be expected – given the latter is a hepatocyte damage model. This is then followed by speculation that BEC TNF production activates DC2's and is followed by an experiment where ex-vivo the DCs are treated with LPS – a TL4 agonist. The relevance of which was not clear to me. Why were they not treated with TNF in the ex-vivo analysis. However, I fully expect that the co- ligand analysis was supportive of TNF interaction between DC2's and the biliary compartment.

6) The authors then show that there is an expansion of DC2s in their murine model and confirmed this by FaxAnalysis. The authors then focus on the DC2B lineage – and show an accumulation of mature cells and pre-cells repopulating the niche with the expression of genes that would favour local IL-17 production. Finally, they show that this population lose their ability to express Mgl2 (and propose this as a local maturity marker).

7) The authors then provide evidence for 2 populations of Gamma delta T cells – based on Scart1 and 2 expression. These 2 populations show differences in IL-17 expression within different TCR clones. This was shown to be due to differences in key transcription factors. In addition, competition for signals between alpha beta and gamma delta is provided for this niche. However, note is made of this low frequency population within the experiment and the pooling was required to conduct the experimental design. So, the importance of this low frequency cellular niche has to be questioned. Furthermore I am not sure that the demonstration of IL-17a knock out reversing fibrosis is direct proof that gamma delta T cells drive fibrosis concretely.

This is because there are many cells that produce IL-17 in these models.

8) The authors then show that depletion of DC2B cells show a reduction in BEC inflammatory gene expression and also show IL-17 down regulation in gamma delta T cells - suggesting a functional relationship between these 2 cellular compartments

9) Finally, the authors show a clear evidence of Lymph Node communication between DC2 and gamma delta T cells and provide interactive ligand relationships between these 2 compartments.

Suggestive improvements/extra work could include:

1: I personally think that the PSC data is not needed for this paper – it's not representative of this model and therefore its relevance to the paper has to be called into question

2: Would it not be more beneficial to develop biliary organoids from humans and induce injury/ and see if they attract/modulate DC2 migration in migration assays

3: Use human DC2s and co-culture with gamma delta T cells to understand interactions

4) Why has a gamma/delta T cell knock out not be studied in this model - would that not be possible to show their importance in fibrosis in this model if this cellular source was of vital importance

Version 1:

Reviewer comments:

Reviewer #1

(Remarks to the Author)

I thank the Reviewers for addressing the original criticisms raised with their initial submission.

It would be greatly appreciated if page numbers for all manuscript changes that are discussed in the PBP response are included and clearly stated in the PBP instead of writing they have been changed in the manuscript without page numbers or including the change in the PBP. Sometimes the text changes have been included in the PBP whereas in other locations, the text changes are only stated and not included in PBP and referenced to sections of the manuscript which makes it difficult to know what has been changed and how so without a significant re-review of the entire manuscript. PBP figures should be labeled with figure numbers (e.g. PBP fig.1) to aid review.

1. Abstract greatly improved. Clear and concise.

2. Table is adequate.

3. Proportion of GD T cells of total T cells has not been provided. Please include GD T cell proportions alongside proportions of IL17+ GDT cell numbers/total IL17+ T cells (stacked bar plots). It is important to highlight the rarity of these cells as showing the proportion of IL17+ cells does to convey the scarcity of IL17+ GD T cells in general. Statement regarding density of GD T cells is not particularly informative (i.e. Overall TRDC+ cell concentration was $29.28 \pm 27.61/\text{mm}^2$, while local concentration within peribiliary regions could reach much higher densities (Fig. S7B).", see results section). This statement does not clearly illustrate relative increase of GD T cells in peribiliary areas vs non-bile duct regions. Please subset the two regions and provide relative fold-increase of GD T cells in peribiliary regions vs other areas.

4. As the pseudonormal tissue shows consistent evidence of liver damage and immune infiltration (acknowledged that TRM may be 'normal' or may be related to liver damage), the term pseudonormal should be replaced with 'non-cholestatic disease controls' to clarify that these samples are not from healthy livers but individuals that can be considered useful controls compared to cholestatic liver indications. Quantification of TCRD+ cells at D5 of DDC treatment should be provided in addition to representative IF image.

5. Please include explanatory statement in main text of manuscript (i.e. 'To enrich all cell types at equal rations, two independent rounds of cell isolation and sequencing were performed at all time points').

6. Appropriate citations of prior work supporting BEC-centric DDC damage are needed for introductory statement.

7. Additional text is useful but does not appear in main manuscript. Please clearly note page number and section the new text regarding latent factor analysis has been added.

8. Good, thank you for clarifying.

9. Please include number of mice analyzed in the figure panels for Fig.3C and Fig.3G (i.e. n=2). Please include changes to new figures in PBP.

10. Ok thanks for clarification.

11. Agreed that this point is similar to (3) and questions there apply also to original comments in (11).

12. Interpretation of DT depletion is inconclusive as it appears multiple subsets are affected, not only GD T cells. Clusters are not annotated in umap(A) so difficult to determine which are Th1, Th2 or Th17 AB T cells or GD T cells. Dot plots in (C) are confusing – does this represent all cells are only GD T cell cluster as appears to be all cells but should be shown for GD T cell cluster vs others that appear impacted with DT treatment. Explanation/findings from GD KO mice are generally unclear and need to be rephrased.

13. Explanation regarding removal of PSC samples is perplexing as removal eliminates major translation to human setting. Data of in situ staining is discussed but now shown.

Reviewer #2

(Remarks to the Author)

The authors have carefully addressed my concerns.

Reviewer #3

(Remarks to the Author)

Thankyou for the resubmission of the manuscript. First of all I would like to thank the authors for the resubmission of their manuscript and for their detailed rebuttal response to the all the points I previously raised.

In doing so, the manuscript is now far more comprehensive and generalisable to the field of hepatology, and provides a detailed understanding of the role of cDC2B and gamma delta cells in cholestatic liver disease.

In particular the correlation analysis and extra spatial xenium analysis (Fig 5) add to the strength of the mechanistic data provided. Also the additional GD KO experiments provide support for the importance of this cellular interaction. Additionally I am grateful to the authors for sharing additional data on the D18 model that they are currently working on to add strength to their arguments.

Finally the additional bile exposure experiments and TNF data provided take the biological model provided to its natural limits. The abstract and conclusions are also now far more balanced.

In conclusion I am therefore satisfied all the points I previously raised have been addressed.

Response to the reviewer comments

The reviewer comments are shown in black and our responses are highlighted in blue. Changes in the manuscript are highlighted in blue.

Reviewer #1:

Overall: Impressive detail and breadth of experiments that convincingly demonstrate a role for cDC-GD T cell axis in murine cholestatic-induced liver fibrosis. Methodologies are cutting-edge and sound. Unclear if unsupervised analysis however would pinpoint cDC-GD T cell interactome as main/important driver of biliary inflammation.

We thank the reviewer for this critical feedback. We conducted our cDC2-gdT cell communication analysis (e.g. Fig. S6) in a hypothesis-driven way, based on our observation that the main IL17 producing cell population were gdT cells. Based on this, we decided to further explore gdT cell communication in a supervised fashion. In that regard, this analysis was not conducted in the whole dataset in an explorative manner but with a clear hypothesis and focused scientific question. To support this claim of the importance of gdT, we quantified absolute numbers of abT17 and gdT17 cells in FACS data and point out that the trend is even stronger in scRNA-seq data (see R1 Q11).

Figures are generally excellent, informative and well organized but could be simplified to focus main messages. Supplementary is extensive and substantial text is devoted to describing supplementary results that should be reduced/consolidated to improve clarity and conciseness.

To streamline the manuscript, we reduced some supplemental findings and changed the text wherever applicable. However, we also want to point out the necessity to generate additional data to address all concerns raised by the reviewers, so that also novel supplementary data was generated as part of this manuscript revision.

Clinical features of patient material needs greater detail and number of patient material analyzed to validate cDC-GD T cell translational relevance is limited (2 PSC and no disease controls).

We acknowledge this important concern and agree that the scRNA-seq data of human PSC and its interpretability is limited (2 patients). To circumvent this point, we also analysed published gdT data from Poch et al. [3] (11 patients). Additionally, we would like to emphasize that the tissue cohorts for immunohistochemistry (n=18) and in situ hybridization (n=5/6) were larger. PSC scRNA-seq data currently remains scarcely available, so that we see the importance of making these data accessible, but we fully agree that we cannot draw robust conclusion from such data. As a consequence, we removed the PSC scRNA-seq data in the revised version of the manuscript.

To increase the clinical information of the pseudonormal liver and PSC scRNA-seq data (n=6), we included a supplemental table summarizing additional patient information (**novel Figure S1A**).

sample ID	sex	age	surgical indication	scRNA-seq technology	derivation process	dissociation reagent
MA	m	52	liver metastasis (liposarcoma)	10x 3' v3.1 (Dual Index)	mechanical and, enzymatic dissociation, FACS	Collagenase D
MA1	f	52	liver metastasis (neuroendocrine tumor)	10x 3' v3.1 (Dual Index)	mechanical and, enzymatic dissociation, FACS	Collagenase D
MA2	f	71	in situ split timepoint colorectal liver metastasis	10x 3' v3.1 (Dual Index)	mechanical and, enzymatic dissociation, FACS	Collagenase D
MA3	f	57	hemi hepatectomy	10x 3' v3.1 (Dual Index)	mechanical and, enzymatic dissociation, FACS	Collagenase D
MA4	m	44	colorectal liver metastasis	10x 3' v3.1 (Dual Index)	mechanical and, enzymatic dissociation, FACS	Collagenase D
MA5	f	76	pancreatic cancer, double duct sign	10x 3' v3.1 (Dual Index)	mechanical and, enzymatic dissociation, FACS	Collagenase D
PSC1	f	53	PSC liver explantation	10x 3' v3.1 (Dual Index)	mechanical dissociation, enzymatic dissociation, FACS	Collagenase D
PSC2	m	21	PSC liver explantation	10x 3' v3.1 (Dual Index)	mechanical and, enzymatic dissociation, FACS	Collagenase D

Novel Fig. S1A: Summary of the human pseudonormal liver tissues used for scRNA-seq.

1. Abstract – slightly disjointed and lacking clear focus but could be dramatically improved with minor revisions. Authors state their goal is to elucidate unconventional T cells but then jump to mapping cell state trajectory of preDCs to cDC2B before returning to GD T cells and IL-17. Brief but better description/contextualization of abbreviated terms (e.g. preDCs, cDC2B, GD T cells, IL-17) would significantly improve readability and understanding for a broader readership.

Thank you for this important feedback on the abstract of our manuscript. Based on the feedback, we rewrote the abstract and improved contextualization of abbreviations and tried to re-order the findings on cDC2 and gdT cells to make them more comprehensible. Please find the adjusted abstract below:

Improved Abstract (197/200 words):

The liver biliary niche serves as a reservoir of tissue-resident immune cells and supports tissue fibrosis upon damage, yet the role of peribiliary immune cells during cholangitis remains poorly understood. Mirroring human biliary diseases with bile acid retention, we induced cholestatic liver injury in mice to establish a spatial and multimodal single-cell sequencing atlas for liver and liver-draining lymph nodes (LN). We characterized a hepatic disease state trajectory from dendritic cell precursors (preDCs) to a mature subset of pro-inflammatory Mgl2⁺ type 2 conventional dendritic cells (cDC2B) and observed dynamic crosstalk with $\gamma\delta$ T cells inducing an IL17-response. Dissection of the cDC2B- $\gamma\delta$ T cell communication node identified the Icosl-Icos axis as a critical cell contact-dependent interaction that was validated *in vitro*. *In vivo*, cDC2B depletion attenuated $\gamma\delta$ T17 responses in cholestatic liver injury and liver fibrosis was reduced in a model of inducible $\gamma\delta$ T cell depletion and in an IL17-deficient background. Conservation of spatial niche architecture and molecular interactions was confirmed in human cholangitis. Our work demonstrates dynamic turnover of cDC2 within the biliary niche during cholestasis, and a profibrogenic function of $\gamma\delta$ T cells contingent on the induction by peribiliary cDC2B, highlighting relevant disease determinants within the immunobiliary and liver-draining LN niche.

2. Public data analyzed for Fig 1 should be cited in main text of manuscript and patients/pseudonormal material used to generate scRNA-seq reference should be briefly defined in main text with general clinical features of patients detailed in a table.

Thank you for these suggestions, that we incorporated in the main manuscript in the following manner:

- We included the data source in the Methods Table file. In particular, information on the public data of **Fig. 1A** was added (derived from 10x genomics datasets, source is provided in the supplements). This data was published as part of the 10x genomics website, without a manuscript reference. Clinical features are not applicable for this sample.
- scRNA-seq data (6 patients): we generated a novel table (Fig. S1A) that we refer to in the main text of the article indicating clinical information of the resected human liver specimens.

3. Relative and absolute number of GD T cells appears limited in biliary niche by scRNA-seq and IHC of human liver tissue. What is the relative proportion of GD T cells as a percentage of total T cells measured by scRNA-seq and IF staining for TRDC? Is the HE image for data in Fig 1a, Fig 1c available? Would be informative to include as small image overlaid on Fig 1a or in supplementary.

Thank you for these valuable suggestions that aim to point out the importance of gdT cells compared to abT cells.

We included information on the absolute numbers of gdT and T cells using our FACS data derived from Fig. S7B. For this quantification, we summed up Il17⁺ abT and gdT cells to visualize the percentage of Il17⁺ cells across abT and gdT cell compartments (total numbers per sample analysed, see stacked barplot below). In this analysis we could see that the gdT cells contributed the majority of Il17⁺ cells (D5/6: 58,23% and D10: 50,76%). This analysis supports our claim regarding the importance of these cells.

Furthermore, we plotted log-normalized Il17a expression in our T cell subset of the DDC atlas data (compare Fig S5A-C; see below). This depiction visualizes the high specificity of Il17a expression to gdT cells, even more specific as to what is seen in the FACS data. This may be explained due to unspecific stimulation of abT cells in the process of cytokine restimulation in vitro using PMA/Ionomycin/Brefeldin A. The specificity of Il17a expression in the scRNA-seq data makes us believe, that the true estimation of Il17a⁺ cells in the abT and gdT cell compartments may be better represented by the scRNA-seq data instead of the FACS data (due to unspecific stimulation of all cells). With both data modalities showing a high specificity for the gdT cell compartment, we demonstrate strong support for the relevance of gdT cells as Il17 producers.

FACS and scRNA-seq quantification of IL17a producing T cell subsets: Stacked barplots displaying stratification of IL17a⁺ cells per sample into abT⁺ and gdT⁺ subsets as quantified by FACS (left). Even though gdT cells are rare, they become an important cell population making up to 58,23% (DDC D5/6) and 50,76% (DDC D10) of IL17a⁺ cells. UMAP depicting the T cell subset (middle) and log-normalized IL17a expression (right column). In this unperturbed data, the specificity of IL17a transcripts to the gdT cluster (cluster 15) is even more prominent to detect.

Finally, we quantified TRDC IHC data in human PSC within the larger cohort of the two cohort being analyzed (quantification of n=10 samples). We included the statement of gdT concentration in PSC samples (“**Overall TRDC⁺ cell concentration was 29.28±27.61/mm², while local concentration within peribiliary regions could reach much higher densities (Fig. S7B).**”, see results section). This statement helps the reader to categorize the cell concentration in general.

We were also thankful for the suggestion to include the **HE image from the ROI shown in Fig. 1A**. Unfortunately, the data to load and allocate the file to a novel slot into the Seurat object using “Read10X_Image()” is not available in the open repository (datafile “tissue_lowres_image.png”).

This is why we could not include the HE of the ROI into the supplemental files as suggested.

4. How many of the TMA samples were analyzed to confirm spatial localization of indicated markers in Fig 1d? What is the disease diagnosis of these patients? **(a)** Is it surprising to observe indicated level of T cell infiltration (CD3) in ‘healthy’ liver? Same comment regarding periductal fibrosis in TRDC HE staining – why is it there if these samples are representative of ‘pseudonormal’ biliary microenvironment? **(b)** Were any GD T cell markers (or T cell markers besides CD45) included in IF staining of steady state mouse liver to confirm presence of T cells/GD T cells around bile ducts? **(c)**

Thank you for these questions and the possibility to clarify the data on our tissue cohorts.

(a) The TMA initially contained 78 pseudonormal tissue cores from 78 patients, these cores were derived from far distant regions of the resected hepatic hemangioma specimen as being characterized in [4, 5]. Because this TMA was heavily used (approx. 40 IHC stainings), the actual number of stained cores is different on specific stainings, since cores were used up during consecutive stainings. We included the n-numbers for each IHC into the novel correlation plot (**novel Fig. 1E**, see also R2 comment 1 “spatial analyses”).

Indeed, resection specimens from far distant liver tissue of hepatic hemangiomas are often considered a very good source for control liver tissue, yet based on the age of these patients and their lifestyle – these tissues may already have been exposed to hepatotoxins during the lifetime. This explains the variability of the tissues we declare in the paper as “pseudonormal”. We agree, that these tissues thereby reflect a range of different tissue states, with some tissues already exhibiting some peribiliary fibrosis.

- (a) With regard to existing T cell subsets that have a tissue residency phenotype (T_{RM} cells), we would also expect to see T cells in the “pseudonormal” liver tissue. TRDC⁺ cells are indeed quite rare in this dataset, however, if they are to be detected, they also have this preferential localization to the peribiliary/portal niche. Regarding the TRDC immunohistochemical staining in Fig. 1D, we chose this image to exemplify the location of the gdT cells, while we agree with the statement of peribiliary fibrosis. We see these changes in accordance with our statement of “pseudonormal” liver – a term that we decided to use to contextualize that liver tissues available in archives in Institutes of Pathology often come with some inherent abnormality as being the reason of surgical resection and the advanced age of the patients that undergo such surgical treatment.
- (b) For steady state mouse liver tissue, we analysed T cells using Cd3e IF (previously not shown and now included as novel part for **Fig. S5E**).

While the presence of these cells could be clearly imaged at DDC D5 using standard imaging techniques (section thickness 5 μ m), these cells were difficult to detect using same techniques in steady state. The same trend was true for Tcrd-GDL reporter mice and their localization around the peribiliary tree: we could convincingly detect areas with a clear EGFP signal in DDC D5 but not control Tcrd-GDL liver tissues. This may not exclude their presence at low concentrations and the possibility that these cells could be detected when more extensive efforts for the imaging of these cell types would be undertaken (thick sectioning etc.). This holds true especially for the Scart1⁺ gdT fraction, which is characterized by the expression of a tissue residency gene signature. Based on the fact, that we could convincingly image these cells at DDC D5 and the knowledge that these cells become more abundant in DDC-induced cholestasis, we did not include these points in the manuscript to avoid unnecessary complication of the storyline. In the human “pseudonormal” data, these cells can be detected, as also pointed out by the reviewer. Whether these are tissue resident cells in non-inflamed liver or whether these cells have been actively recruited because of inflammatory signal cues cannot be fully elucidated using these samples, the marker panels being used and the clinical information being available.

5. Why were different protocols used to isolate cells for scRNA-seq? Assumption that FACS could not separate all subsets simultaneously but unclear as written.

Thank you for the possibility to clarify our experimental conduct. We included an explanatory paragraph, that explains the use of two different scRNA-seq protocols to isolate cell populations of interest (see Methods section subsection “Cell enrichment strategies for individual scRNA-seq datasets”).

6. Transitional text between scRNA-seq DDC atlas and subsequent subsetting of BEC cluster lacks clear rationale given that authors state they performed scRNA-seq to investigate unbiased dynamics of gene expression across the biliary niche (a). Difficult to discern what cell types and cell states change most dramatically during disease progression – did the unsupervised analysis point toward BEC or was this the subsetting done to answer defined hypothesis? (b)

Were there clear enrichment of cDC and gdT cell signals at any timepoint(s)? (c)

Thank you for raising these points.

(a) After the observation of this specific immunological niche around the biliary tree, we chose the xenobiotic DDC model that would specifically affect this immunological niche. We included an explanatory sentence detailing our rationale to first analyse the BEC compartment in this data. Please refer to results, chapter “DDC diet induces small duct inflammatory disease”, first sentence.

(b) Based on other complementary analyses such as the IF-based morphologic characterization of this model, which clearly showed the BEC to be strongly affected, we decided to focus on these cells first. To clarify the rationale, we included a short explanatory sentence into the manuscript (see response to (a))

We would like to point out here, that a general cell abundance analysis of cell types within the whole atlas may have only limited interpretability, since the aim during scRNA-seq atlas generation was also to balance out drastic changes in cell abundances using our cell enrichment and mixing strategy prior to GEM generation and scRNA-seq.

(c) The later focus on cDC was again based on the morphological relationship between BEC and cDC, which we visualized in human (Fig. 1D) and mouse (Fig. 1F-I). However, indeed these cDC2 and gdT cells show a temporal distribution in the atlas across timepoints indicating importance in early disease dynamics (see Fig. A,B below). For cDC2 a strong increase can be observed at DDC D5, making it the most abundant DC subset at this timepoint. gdT cells also show increased abundances, and the abundance curve is shifted to later timepoints, which may indicate a temporal dependency of gdT influx and cDC2 maturation.

To not overly complicate the already dense manuscript, we decided to not include these results, but display them here as evidence of such temporal regulation and as support of our further focus on these cells. Additionally, previous literature (Müller et al.) strongly pointed towards relevant dynamics within the cDC2 compartment [6].

We believe these additional data support our rationale to investigate cDC2 and gdT cells.

Protocol-specific total number of DC and gdT cells captured in scRNA-seq data per timepoint: (A) Total number of DC subsets derived from protocol A, which only enriched for Cd45+ and macrophages specifically. The abundance of DC subsets from protocol A can thus be quantitatively interpreted, while protocol B aimed to isolate DC at similar ratios per timepoint. (B) gdT cell numbers derived from protocol B. T cells had to be enriched specifically, since T cell subsets were not well captured in protocol A.

7. Definition of latent factors and relevance of BEC and DC gene programs unclear as written and should be further explained for non-bioinformaticians. Difficult to interpret data shown in Fig 2F and Fig 2G.

Thank you for this important feedback. We extended the methods section with a description for improving the understanding of NiCo's [7] use of latent factors and the calculation of niche interaction maps. Please find below the excerpt of introduced explanation:

“For the calculation of niche interaction maps, cell type enrichment ratios within cellular neighborhoods of a central cell were calculated and a logistic regression to predict central cell type identities from these ratios was performed. The regression coefficients were then used to reconstruct the interaction map. Inferred latent factors from expression data were used to identify a small number of gene programs explaining cell state variability of each cell type. Covariation analysis of these latent factors across co-localized cell types may thus capture the effect cell-cell interactions on gene program activity [7].”

8. Umap in Fig 3C be simplified (also split) to delineate cDC2A and cDC2B clusters? **(a)**
Similarly, could timepoint umap overlays in Fig 3D be consolidated into one umap with trajectory analysis shown by arrows to better indicate dynamics of cDC2 states at early to late to recovery timepoints? **(b)**

Thank you for this feedback.

(a) We believe that the grouped visualization of the cDC2 timepoints captures the cDC2 cell states in an interpretable manner. In particular, differences between steady state and diseased state are captured by the y-coordinates while variability within diseased states is captured along the x-axis.

To improve the understanding of the cDC2 trajectories, we annotated the cDC2-specific map of clusters (**adjusted Fig. 3C**) to exemplify the directions of the timepoint-specific disease trajectories. The subsets of **Clec10a⁺ cDC2B**, **preDC**, **Ccr7⁺ mat DC** and **non-diseased cDC2** will give orientation within this gene expression space.

(b) The summarized trajectory as proposed by the reviewer is already part of the Supplemental Figure S4D visualizing transition probabilities calculated by VarID2 [8] across clusters and timepoints.

9. How closely does cDC2 state at 'STOP' compare to cDC2 markers at steady-state (i.e. D0)?
(a) Including numbers of mice/cells analyzed by scRNA-seq (Fig 3C and 3G) would aid intra and inter comparisons of cluster sizes **(b)**. Similarly, dot overlays representing number of analyzed mice should be included for Fig 3B and 3F bar graph. **(c)**

Thank you for your valuable suggestions.

(a) We compared the expression of the cDC2A/B gene signatures using the dotplot representation of VarID2 and included D0 into our analysis together with the timepoints D19 and D19-STOP. Here, we found no trend towards the expression of cDC2A or cDC2B genes. Thus, we believe these data are difficult to interpret and existing dotplots in Fig. 3H already show the essence of the information. We decided to show the DotPlots here for the reviewer's information but not to include this information into the main manuscript.

Dotplots displaying sample-specific cDC2A/B gene expression in DDC D0, D18/19 and STOP conditions. Log-normalized gene expression is shown and fraction of cells expressing the gene is encoded by dot size.

(b) This is a helpful suggestion, and we decided to include **cell numbers** into the Figures to make the data more interpretable. For the DC comparison overall, we included the total DC numbers into the bar plot of Fig. 3A and for cDC2 into Fig. 3D and Fig. 3G. We also included an explanatory sentence in the methods section, how cDC1, cDC2, pDC subsets were quantified based on which data subsets.

For the statement of the mice being used per experiment, we believe, this information is not informative for scRNA-seq, since for many populations not the absolute number of cells within one sample was sorted but only sufficient numbers to be able to conduct GEM generation as quickly as possible. This is in accordance with scRNA-seq protocols to avoid long processing time. For all sequencing experiments: Two mice of the same timepoint were pooled and cell isolation conducted as stated in the methods section, so that this information is already accessible in the manuscript.

(c) We included absolute numbers in the dot overlays into Fig. 3A, Fig. 3F and Fig. S4B and decided to switch the display to boxplots. Data points of individual samples are now depicted.

10. From data in S5, appears Il17a, Rorc and Rora transcriptomic signatures are relatively weak (expression less than 1 LF above comparator). What is comparator here, D0? **(a)** As stated by the authors, abundance of GD T cells are very low - are the receptor-ligand interactions known to facilitate recruitment and activation with cDC2 exclusive to GD T cells? **(b)**

Thank you for the possibility to clarify our methodology.

- (a) With regard to the dotplot information shown in Fig. S5B, we would like to mention that this plot shows log₂-normalized expression values without any comparator. This means that overall expression values were compared. Expression values of selected genes such as Il17a, Rora or Rorc are encoded by dot color and these genes are expressed at low levels. Dot size encodes the fraction of cells expressing the gene of interest.
- (b) Abundance of gdT cells is indeed low. The importance of the gdT- cDC2 interactions that we identified could be justified by two reasons:
- First, we identified gdT cell-exclusive interactions that can selectively activate gdT cells, as implied by the reviewer. However, this is not the case for many of the interactions that we identified.
 - Second, the importance of the interactions was demonstrated by the expression of marker molecules in T cell populations of interest and the fact that these cells show an effector cell phenotype. Indeed, only gdT cells express high levels of Il17a, indicating a selective T17 response in gdT but not abT cells. Although the interaction of cDC2 with gdT cells may not be entirely specific, cDC2 serve as an important cellular interaction partner to induce gdT17 responses in this model. We also confirmed this interaction *in vitro* (coculture) and *in vivo* (Mgl2-DTR depletion) in our manuscript.

11. What are the absolute numbers of conventional Il17a+ T cells relative to Il17a+ GD T cells? Although a greater proportion of GD T cells appear Il17a, their overall scarcity (Figure S7A) questions their importance in cholestatic fibrosis.

Thank you for this important suggestion.

We conducted the proposed quantification by FACS after cytokine restimulation and point out that with $58.23 \pm 19.3\%$ (DDC D5/6) and $50.76 \pm 7.9\%$ (DDC D10) a significant proportion of Il17a⁺ cells has a gdT cell identity (see also comment of R1 Q3). As pointed out in **R1Q3**, without cytokine restimulation and by the assessment of Il17a transcript levels in the T cell subset of the scRNA-seq data, this difference is even more prominent, indicating that gdT cells are the main source of Il17a.

Additionally, even if there is a minority of non-gdT cells expressing Il17a, we want to point out the different effector differentiation dynamics of unconventional (UTC, among them gdT cells) and conventional T cells (abT cells) [9]. UTC have a more rapid and quick effector response compared to conventional T cells, thereby essentially contributing to the initiation phase to orchestrate a profibrotic response.

Together, we believe these points convincingly highlight the importance of the gdT cell compartment as a source of Il17a in the DDC diet model.

12. How did DT depletion of cDC2B impact other immune subsets, particularly other T cell populations irrespective of Il17a expression?

We thank the reviewer for raising this important question.

To include information on the dependence of T cell responses and cDC2 depletion, we included a dotplot displaying the expression of Th1, Th2, Th17 marker genes and other markers of T cell activation (transcription factors (TF) and general markers of activation). For this, we generated a data subset containing NK and T cells (clusters 2, 3, 10, 12, 19, **Fig. A**). This subset contained 4364 cells, with a slight predominance of T cells in the DDC D5-DT condition (DDC D5: 1727 cells; DDC D5-DT: 2637 cells). The most striking difference between the T cell compartments was the fact that the T cells in the non-depleted condition were primarily naïve ($Sell^+$, $Ccr7^+$, $Cd44^-$, **Fig. B**). For $gdT17$ related genes, this analysis confirmed an attenuated Il17a response in the DT condition (**Fig. C**). These observations could be in line with the known shortcomings of DTR models, including immunogenicity of Diphtheria toxin as described in the literature [10].

Another important aspect of adverse effects that reduce specificity in DTR models is off-target cell-killing, e.g., of myeloid cells. To assess the specificity of the depletion strategy, we analysed our scRNA-seq data and analysed the cell composition across samples within mature tissue resident dendritic cells (cDC1, cDC2, pDC) and of LAM/KC as another major myeloid compartment in the liver (**Fig. D**). This strategy exemplifies the specificity of the depletion, while excluding DC precursors that will ultimately repopulate the niche.

Indeed, Mgl2-DTR based depletion was highly specific to cDC2B (cluster 1) at DDC D5-DT (11.47-fold reduction). cDC1 were also reduced but to a much lower extent (1.53-fold reduction, **Fig. D**). pDC and KC/LAM were increased in number after DT depletion (2.26- and 1.29-fold increase). These changes are in line with previous descriptions about specific caveats of DTR models, while simultaneously showing convincingly, that the cell type of interest is most heavily affected by the depletion strategy.

We decided to include Figure Panel D of this information into the manuscript (**novel Fig. S9B**). We also include a statement in the discussion section to address the shortcomings of DTR models (“... and exclude potential DT- mediated off-target effects affecting our analysis.”).

T cell activation and myeloid cell distribution across DDC D5 and DDC D5-DT conditions:

(A) UMAP representation displaying clusters (left) and samples (right) of the T cell subset of merged DDC D5 and DDC D5-DT data.

(B) UMAP representation of the T cell subset of the combined DDC D5/DDC D5-DT dataset displaying log-normalized gene expression of *Ccr7*, *Sell*, and *Cd44*.

(C) Dotplot displaying expression of T cell marker genes (left) and gdT cell marker genes (right). Log-normalized expression is color coded and fraction of cells expressing the gene is encoded by dot size. **(D)** Stacked barplot displaying sample distribution of mature DC subsets and Mφ within DDC D5/DDC D5-DT conditions.

Furthermore, we conducted qPCR of tissue derived transcripts to catch global changes over time, analysing two timepoints (DDC D5 and D10) with and without DT-based depletion. For these experiments, the depletion was induced as previously on day DDC D-1 and DDC D2 (compare scheme in Fig. 6A), so that a certain degree of repopulation of mature cDC2B would be expected at D10.

We confirmed altered fibrosis initiation on D5-DT (significant reduced tissue levels of *Pdgfrb*, *Des* transcripts), cDC2B depletion (*Mgl2* transcript levels) and aspects of gdT cell biology (*Tcrd* transcript levels), all of which were partially attenuated at D10, reflecting the repopulation of the cDC2 niche by incoming preDC. We believe that these data support our study and included this qPCR analysis into the supplemental Figure files (**novel Fig 9H**).

Novel Fig 9H: Barplot displaying tissue-derived transcript levels in DDC D5, DDC D5-DT, DDC D10, DDC D10-DT conditions. Asterisks indicate level of significance. Data normalized to control liver tissue transcript levels. One sided T test (n= 4-6 mice/group), *p<0.05.

13. Authors repeatedly attempt to translate mouse findings to PSC however few PSC samples were analyzed (n=2) and appropriate disease controls for biliary inflammation (e.g. PBC) and general end-stage liver fibrosis (e.g. ALD, MASH) are lacking.

Thank you for this feedback. Indeed, for the PSC scRNA-seq adequate controls are lacking. This makes a comparative analysis difficult. Data of cell populations such as $\gamma\delta$ T cell and cDC2 are currently still scarcely available and existing datasets in PSC do not enrich for these cell populations. We thus decided to remove the scRNA-seq data of the two PSC patients.

Additionally, we reframed the statements of the transferability of the findings of the DDC model towards human PSC. Unfortunately, we do not have access to a high sample number of the requested non-PSC liver controls (ALD/MASH/PBC), so that such a comparison cannot be delivered by us. In fact, we believe that $\gamma\delta$ T cells also play a role in other disease context that lead to fibrosis. To make our results more comparable, we included a quantification of 10 PSC tissues that were digitalized and TRDC⁺ cells were quantified. This information (“Overall TRDC⁺ cell concentration was $29.28 \pm 27.61/\text{mm}^2$, while local concentration within peribiliary regions could reach much higher densities (Fig. S5H).”) has been now included into the main manuscript.

Finally, we also conducted another round of duplex in situ hybridizations of TRDC and IL17A in human PSC liver (n=5). While the majority of TRDC⁺ cells was IL17A negative, thereby reflecting scRNAseq data and the ongoing discussion of the presence of IL17A⁺ $\gamma\delta$ T in human disease, we could also find regions with double positive cells. Yet, even detectable, the fraction of cells being IL17A⁺ is much lower in the human samples. Based on these results and to not further complicate the storyline of the manuscript, we decided to not include these results and increase our focus on the mouse data.

14. DDC and BDL models of cholestatic liver injury were used to study GD T cells but alternative models of non-cholestatic liver injury used to study cDC1/2 dynamics (e.g. CDE diet) were not included as controls, which would help determine if relevance of GD T cells are truly specific to cholestatic disease or general liver disease feature. **(a)** Use of GD T cell knockouts would clearly identify their role in cholestatic fibrosis **(b)** as well as education of cDC into inflammatory or resolving cell states **(c)**.

Thank you for bringing up these important points.

(a) To address this concern directly, we measured gdT17 cells in CDE D5 and CDE D10 providing additional information to clarify the specificity of gdT17 responses in the context of cholestasis. Indeed, gdT17 responses were reduced at CDE D5 ($p < 0.001$) and a trend towards increased gdT-mediated and delayed Il17a responses at CDE D10/11 ($p = 0.058$) was detectable.

These results exemplify the early onset and prominent gdT17 responses in cholestasis. The results are now part of the novel Fig. S7B.

Part of novel. Fig. S7B. Boxplot displaying frequency of Il17a⁺ cells within Cd3⁺ abTCR⁺ (light grey) and Cd3⁺ gdTCR⁺ (dark grey) compartments in the CDE diet model after cytokine re-stimulation. Two-sided t-test (5 independent experiments, n=5-9 mice/group). *** $p < 0.001$, n.s., not significant.

(b) Furthermore, we used two gdT cell knockout models (Tcrd-KO and Tcrd-GDL models enabling constitutive and inducible KO) to study gdT cells in the context of cholestasis and biliary fibrosis. While constitutive gdT knockout showed a compensation by abT cells in the production of Il17a (**compare Fig S7C**), we conducted additional analyses to further study these cells using the DT-based, conditional depletion model. These experiments are explained and listed in full detail in the response of **reviewer 3 point 7**.

In brief, we assessed the degree of cholestasis using blood plasma profiling, conducted global analysis using qPCR of tissue derived RNAs and isolated the global DC compartment by magnetic enrichment.

(c) The question of the re-education of DCs by gdT cells is interesting, as it would demonstrate a bi-directionality in cellular crosstalk within the cDC2-gdT cell signalling hub.

We conducted a DDC diet experiments using Tcrd-KO (constitutive knockout), Tcrd-GDL (inducible knockout) and B16 controls analysing two different timepoints of DDC diet feeding. For both experiments and the conditional depletion in Tcrd-GDL mice, DT was given on D -1 before and D2 after starting the DDC diet (equivalent design, as compared to Mgl2-DTR experiments). We analysed transcript abundance by qPCR on D5 of magnetically enriched

Cd11c⁺ cells with the aim to resolve expression of DC maturation markers in dependency of gdT cell presence in cholestasis.

At an early timepoint of cholestasis (**DDC D5**), we observed an increased abundance of transcripts defining mature DC states (*Ccr7*, *Cd80*, *Ccl17*, *Ccl22*, see panel A below).

These data indeed show differences between the constitutive and inducible gdT knockout in cDC2 education, however these data may further complicate the storyline of the manuscript. In fact, the manuscript mainly focuses on cDC2-induced effects on gdT cells, while re-education is not further analysed. Therefore, we believe these results are beyond the scope of the manuscript and we decided to not include these results into the main manuscript.

Barplots depicting normalized transcript levels in CD11c⁺ enriched cells after DDC D5 treatment. Data normalized to transcript levels in C57BL/6J mice receiving DDC for 5 days (equals 1). Group size as following: Bl6 (n=6), Tcrd-KO (n=4), Tcrd-GDL (n=5). Statistics (two sided t-test shown for Tcrd-KO (TCRD-tm_KO) and Tcrd-KO (ITC_GDL, both compared to DDC D5 control).

Reviewer #2:

This is an interesting study by a well established group in the field presenting a multimodal single-cell atlas of the immunobiliary niche in mouse cholangitis, complemented by human data. The authors highlight dynamic cDC2B maturation, $\gamma\delta$ T 17 induction, and functional relevance of the ICOSL–ICOS axis. The dataset is rich, and the study addresses an important and relatively unexplored area. However, in its current form, several claims appear overstated or insufficiently supported by the presented data. The study is more descriptive than mechanistic, with limited causality in validations. Human translations are preliminary, and several interpretive issues need clarification.

Please find in the following my points.

Thank you for this positive feedback.

Spatial analyses (Fig. 1)

It is difficult to appreciate whether CD3E+ cells are truly enriched around ducts/large vessels. Quantification of their localization would strengthen this point **(a)**. The density map for CD207 is unclear; why was a distinct visualization chosen for this marker only? **(b)** If biliary epithelial cells (BECs) and immune cells were sorted separately, why are BECs so underrepresented relative to immune cells? **(c)** What input ratios were used when loading the cells? **(d)** Many proteins are inconsistently formatted (e.g., some are capitalized, others not). Please standardize (e.g., ITGAX+ cells). Figure legends indicate n=3, yet there are no quantitative analyses. Including barplots would help. **(e)** Annotation strategy needs clearer explanation. A dotplot of marker genes validating cluster identities would be welcome. **(f)** Timepoints in legends could be reordered chronologically for clarity **(g)**. The rationale for using t-SNE (rather than UMAP throughout) is not clear **(h)**.

Thank you for pointing out various inconsistencies in our manuscript. We would like to reply to the requests in a point-by-point fashion:

(a) Thank you for pointing out that a quantitative metric would help to assess enrichment of different cell types in the immunobiliary niche. To do so, we conducted a spearman correlation analysis of marker-positive cells detected in our tissue microarray dataset. As a figure output we generated a correlation map (novel **Fig. 1E**) for the TMA cores, which were analysed using all markers throughout several studies [4, 5]. This enabled correlation of marker positive cell numbers that should be enriched in portal fields only, e.g. CK7⁺ BEC with various immune markers that are enriched in the immunobiliary niche. While the statistical power may vary depending on the number cores being analysed per staining (19 stainings were quantitatively analysed), we see a strong correlation for CD3/CD8 and CK7, while CD207 and CK7 show a correlative trend, however, not reaching the level of significance. The results of this analysis can give a first hint regarding the localization of the cells, while the number of cores being analysed in the TMA determine the statistical power, which varied among stainings. Markers such as CK19 and TRDC with a core number was below 20, were excluded from the analysis.

We believe the requested correlation analysis improves our manuscript and supports the hypothesis of an immunobiliary niche that contains T cells and DCs, which is later on confirmed using independent techniques.

E
- (b)** We decided to remove the display of the cell concentration in the TMA core overview for CD207. This was indeed only shown for this marker, and we decided to show the overviews of the TMA cores for all markers instead. We included several other markers to give an overview that multiple immune cell types show an immunobiliary enriched localization.
- (c)** We would like to clarify our FACS sorting strategy: The number of FACS-sorted cells and the input ratio was defined, immune cells may seem overrepresented by the fact that in protocol A Cd45⁺ cells were used as one population of interest to capture abundant immune cells, while for rare immune cells additional populations were sorted and pooled. BEC may be slightly underrepresented based on the digestion protocol causing incomplete digestion of cholangiocyte fragments, which may be detected as “cells” in FACS, however, do not pass QC in scRNA-seq. In comparison to other published disease atlas data, we think that the BEC compartment is well covered in our data. Expecting a low yield in the BEC compartment, we decided to enrich for this population in both protocols.
- (d)** The input ratios are explained in the methods section – yet we appreciate the question, since the number of cells being input and being recovered may vary, as also exemplified for BEC in **(c)**.
- (e)** Thank you for pointing out inconsistencies with regard to the statement of the number of analysed samples and the naming of proteins (e.g. ITGAX+). We corrected such errors throughout the Figures (Fig. 1F, 1G, 6E, S4A, S5F). The manuscript text was also corrected in this regard.
- (f)** A heatmap/dotplot of marker genes is already part of the manuscript for the human steady state and the DDC atlas (Fig S1B, S2H).
- (g)** Thank you for this suggestion. We changed the legends depicting the timepoints and samples in a chronological order in all respective figures (Fig. 1K, 2A, S2G, S3A, S5A).
- (h)** We would like to point out that the tSNE embedding was solely used for visualization of our complete atlas. We do not derive any further analysis from this complete atlas data. We would also like to note that there is no clear rationale for choosing either

UMAP and tSNE as they can capture the same amount of information. This is rather a function of the input parameters used. Since our tSNE map visualizes the data structure well, we would like to keep it.

cDC2 trajectory and cluster interpretation (Fig. 2–3)

The claim that cluster 8 expresses chemotaxis/adhesion genes is not entirely convincing. Cluster 16 seems to express more of these genes except *Il23a* (a). Exclusion of clusters 3, 12, 15 due to residual cDC1 signatures requires more transparency — which genes drove this decision, and are they uniquely cDC1-specific? (b) There is some confusion regarding *Mgl2* expression: overall *Mgl2*⁺ cells increase, yet the proportion of *Mgl2*⁺ cDC2B decreases. This apparent contradiction should be clarified. (c) There seems to be a discrepancy in the interpretation of DC recruitment. The authors claim that *Mgl2*[−] preDCs are recruited along disease progression, yet their own data (e.g., Fig. 3E) show an increase in *Mgl2*⁺ cells, which they themselves describe as the more immature DCs. This would rather suggest that *Mgl2*⁺ precursors are recruited into the niche. The authors should clarify this trajectory: are *Mgl2*⁺ cells the recruited precursors that then lose *Mgl2* as they mature, or are truly *Mgl2*[−] preDCs entering the tissue? (d)

Thank you for this feedback. We would like to respond to the individual questions point by point:

- (a) Thank you for this feedback, we rewrote the manuscript and include information that clusters 4,8 and 16 express chemotaxis/adhesion genes in the BEC subset of the data.
- (b) We visualize the log-normalized expression of cDC1 genes *Xcr1*, *Cadm1*, *Gcsam* to demonstrate residual cDC1 signatures in clusters 3, 12, 15 (see Figure panel B below). These clusters are *Mgl2* and *Clec10a* negative (see main Fig. 3D). We believe, this is evidence of the different cell identity of these clusters and we now list the genes that prompted us to remove these clusters from our analysis in the results chapter.

UMAP representations of cDC2 subset (A) and log-normalized expression of *Xcr1*, *Cadm1*, *Gcsam*, which depict the cDC1 identity in clusters 3,12, 15 (B).

- (c) This is an important point. Since we already anticipated a potential caveat we had included an explanatory sentence into the results chapter of these data (“...***Mgl1* and *Mgl2* were co-detected using immunohistochemistry, which marks immature cDCs and macrophages [29]**”). To provide further clarification, we added an a more detailed statement to this section: “**As such, combined *Mgl1* and *Mgl2* detection does not enable quantification of mature *Mgl2*⁺ cDC2B but detects macrophages and cDC2 independent of their maturation state instead.**”

For clarification, it is important to note that the immunohistochemistry stains *Mgl1* and *Mgl2* (Fig. 3E) and these data cannot be compared with FACS and scRNA-seq data, where *Mgl2* protein or transcript is detected specifically. Yet, because of the very clear morphology and adjacency relationship of chromogenic *Mgl1*/*Mgl2*⁺ cells, we would like to keep these data in the main manuscript. We hope we could clarify the presentation and the biological meaning of these data.

(d) This is in line with the response for (c) and the fact that Fig. 3E does not depict a specific staining for Mgl2. This should clarify, that the increased cell numbers to be detected at the immunobiliary niche include most likely macrophages, cDC2 and DC precursors, which can be stained by combined staining against Mgl1/Mgl2. In that regard, the staining itself may be more comparative to a more broad cDC2B marker such as Clec10a (compare UMAP in Fig. 3D). I hope these explanations resolve the confusion and increase the clarity of the made statements.

With regard to the trajectory from preDC to mature Ccr7⁺ cDC2, we would like to make the following explanations: Ly6c2⁺ Cd7⁺ Siglech⁺ preDCs are Mgl2-negative, these cells then upregulate Mgl2 expression as part of their tissue adaptation. In the process off antigen-uptake and DC maturation, these cells undergo gene expression changes (maturation on and off switch, respectively), which then leads to the downregulation of Mgl2-expression, while Mgl2 protein levels remain expressed (compare LN scRNA-seq data, Fig. 7C). We agree, that these expression changes are complex to grasp, mainly because of the maturation-off switch during DC maturation. To make this more clear, we modified the results section and included an explanatory sentence into the main manuscript:

“Cells derived from later timepoints (from D5 onwards) included **Mgl2^{negative}** DC precursors (preDCs) expressing Cd7, Ly6c2, Tcf4, Runx2, Siglech, and Csf1r (Fig. S4C). The majority of cDC2 derived from D9 onwards were Mgl2-negative, suggesting that Mgl2 expression may be a proxy for the cDC2B **tissue residency and** maturation state. **However, Mgl2 expression in cDC2B is downregulated as part of the MAT-OFF switch as seen in tissue egressing Ccr7⁺ DCs (cluster 11).**”

Human validation

The presence of IL17+ $\gamma\delta$ T cells in human PSC is implied but not convincingly shown. Given ongoing debate about human $\gamma\delta$ T17, this claim should be toned down or better validated (a). The human validation of $\gamma\delta$ T17 identity remains unclear in their data. Do human $\gamma\delta$ T cells in PSC express canonical Th17-defining transcription factors (e.g., RORC, MAF, IRF4) and cytokines (IL17A, IL17F)? (b) The current data (Suppl. Fig. S7H–J) show no IL17A transcript levels, and no clear evidence for a broader Th17 signature in human $\gamma\delta$ T cells. Since the existence of human $\gamma\delta$ T17 cells remains controversial, stronger evidence or a more cautious interpretation is needed (c). At minimum, the authors should explicitly show whether RORC, MAF, IL17A/F are expressed in their human $\gamma\delta$ T cell subset (d). Cross-species conservation of interactions is intriguing but may be overstated (e).

Thank you for this feedback regarding the human validation. Please see our responses below:

(a) We toned down our statements (see revised manuscript) and also removed human PSC scRNA-seq data including the ligand-receptor interaction analysis. Additionally, we conducted TRDC/IL17A in situ hybridization, where double positive cells could be detected, however the majority of TRDC⁺ cells were IL17A negative. These results reflect the controversy of the existence of such cells in the human setting. As a consequence for our additional analyses, we removed the cell-cell communication analyses and also the gdT analyses from the published Poch et al. data (compare previous S7).

- (b) We plotted the expression of RORC, MAF, IRF4 in the $\gamma\delta$ T subset of the Poch et al. data (previously done for the UTC subset in previous Fig. S7, see below). Here, we decided to not integrate the data to see patient-specific differences in the $\gamma\delta$ T compartment. While IL17a expression is low throughout the samples, we see sample-specific differences for the expression of T17 identity forming TF, with e.g. a convincing expression of RORC, MAF in “sample 14” (non-IBD PSC). This expression heterogeneity of RORC, MAF, IRF4 is also recapitulated in our dataset comprising two patients, while again IL17A expression is only detectable in few cells. We conclude, that the $\gamma\delta$ T cell subset in human PSC is heterogeneous, where $\gamma\delta$ T cells expressing a T17-identity forming TF signature may exist. These cells, however, do not express IL17A. Whether the expression of IL17A may be rapidly triggered upon cholestatic insults needs to be further elucidated. The data may also point out, that $\gamma\delta$ T cells in PSC may have other functions beyond the secretion of IL17A. This data is also in line with further experimental data generated in (c, see below).
- (c) To generate additional data *in situ*, we conducted TRDC/IL17A duplex *in situ* hybridization in 5 human PSC tissues. Indeed, the majority of TRDC⁺ cells were IL17A negative, however throughout the tissues double positive cells could be detected at low frequency (see image below). While this data cannot fully resolve the debate about of human $\gamma\delta$ T17 cells in PSC, it supports the idea of $\gamma\delta$ T heterogeneity in those samples throughout certain disease stages and spatially defined tissue domains.
- (d) MAF, RORC, IL17A/F were also plotted in the PSC data, see Figure panel B below.
- (e) We rewrote the manuscript and excluded these results on human $\gamma\delta$ T cells, since the data is far less clear compared to our data in mouse. Additionally, the small number of samples, especially in our own scRNA-seq data makes interpretation difficult. Based on this revision, we fully agree with the suggestion to remove such data (compare also R3). These changes should also address our previous potential overstatements made in the initial submission.

A**gdT subset Poch et al. Data (JHEP 2021)****C****B****PSC data (own data, n=2)
gdT and cDC2 subset
(used for CellChat in Fig. S7)**
gdT cells in human PSC. (A) UMAP representations depicting gdT cell subset of the Poch et al. data [3]. Magenta ellipses highlight gdT cells from sample 14, which show a TF profile matching T17 polarization, while being negative for IL17A. (B) cDC2 and gdT cell subset from own scRNA-seq data used for cell-cell communication analysis (compare previous Fig. S7). Ellipses indicates gdT cells matching a T17 TF expression phenotype, while IL17A expression is absent. (C) Duplex in situ hybridization analysis on human PSC tissues. Assessment of novel TRDC/IL17A ISH data confirms that the majority of TRDC⁺ cells are IL17A negative, while few double positive cells can also be detected (see higher magnification region of interest).

Reviewer #3:

In this manuscript, the authors create a scientific narrative demonstrating the importance of type 2 cDCs and gamma delta T cells. The majority of which is described in a mouse model of biliary injury using a range of high level scientific techniques.

Obviously, the role of type 2 cDCs reported here is not a new one, and it was first described in a range of mouse models by Müller AL et al (J Hepatol. 2022 Dec;77(6):1532-154) (which is acknowledged in the references). Therefore, the novelty of this study is in the potential cross talk with the gamma delta cell population in the biliary niche. In doing so it provides mechanistic insight to the role of dendritic cells in biliary injury and potential ligands that could be targeted for drug discovery.

Overall, the authors have submitted a large amount of data to provide robustness to the proposed narrative. However I would say that within the large volume of experimental data – which skips between human and murine models, there is perhaps a distraction for the reader particularly for the reasons I highlight below.

Thank you for the constructive feedback and the possibility to clarify and improve the main statements made in our manuscript.

My major comments for this manuscript include

1) I think the closeness of this model (ie mirroring to PSC) requires firstly some careful clarification by the authors. PSC is an autoimmune liver disease of the liver and colon. Its genetic risk is HLA driven, and the importance of CD4+/CD8+ and T regs is well described. The inflammatory response is therefore the driver of biliary injury in PSC. This is in strong contrast to the mouse model studied here, where the induced epithelial/cholangiocyte damage secondarily induces inflammation – therefore they are not one in the same. The DCC model is therefore not a mouse model of PSC. Perhaps the authors should therefore highlight that what these have in common, and the point of their study – ie how biliary injury may amplify the local immune response and perturbate epithelial resultant injury.

Thank you for the feedback and the possibility to improve our manuscript, we adjusted the introduction section accordingly (Introduction, first paragraph). We agree, that the pathophysiology is not identical in DDC and PSC and discuss common features of both immune-mediated (PSC pathophysiology) as well as epithelial damage induced models (DDC/cholestasis pathophysiology) in the discussion part (Discussion, first paragraph). The implemented changes are as following (changes in bold):

Introduction:

Primary sclerosing cholangitis (PSC) is a multifactorial autoimmune-related disease with rising occurrence in the US and EU [1, 2], and the disease course **may lead to cirrhosis and the requirement of liver transplantation**. While there are different subtypes of PSC [3], and PSC is frequently linked to other chronic inflammatory diseases, the common histological features include periductal fibrosis, proliferation of reactive bile ducts known as ductular reaction (DR), bile duct stenosis, dilatation and rarefication leading to biliary cirrhosis, **cholestasis and altered myeloid responses** [4, 5]. Clinical presentation of PSC and its chronic disease activity may include intermittent episodes of acute cholangitis and cholestasis [6] and involve dysfunctional cDC2 and T cell responses contributing to disease severity [7, 8], yet the role of

unconventional T cells (UTC), their potential crosstalk with cDC2 **and their role in amplifying biliary injury responses** remains to be elucidated.

Discussion (changes in bold):

While a number of immunological susceptibility genes have been identified in PSC [43], the disease-aggravating and self-perpetuating biliary niche- **and injury**-specific immunological circuits that maintain and drive disease pathology remain incompletely understood. **The DDC diet model used in this study, may mimic certain aspects of cholestasis and biliary injury in human PSC and how these local cues may amplify local tissue injury responses.**

2) In the opening introduction they say PSC ultimately leads to transplant – this is not true – 50% are alive and well at 20 years. I would not say that the disease is characterised by dysfunctional myeloid responses. Therefore, to compare PSC to a toxin mediated liver injury, needs careful discussion by the authors and questions why they wish to try and reproduce some of their findings in PSC.

Thank you for this correction. We changed the introduction section accordingly with regard to the necessity of receiving a transplant (see first sentence of introduction) and the existence of dysfunctional myeloid responses (“**altered** myeloid responses” instead). We also clarify the differences of HLA T-cell mediated autoimmunity and the damage-induced model in this study. We point out that we want to understand local damage-induced cell signaling circuits that may lead to disease aggravation/amplification, which are mediated by e.g. local cholestasis (see discussion, first paragraph).

3) the authors show that CD1C and CD3E transcripts are present near large vessels and biliary tracts (Fig1A). I must admit – I think here spatial has been used for spatial sake. The figure presented loses the clarity needed that would be gained by a simple HE that has been stained for these markers at low and high power, which would then show a broad range of zones 1-3 of the liver, portal tracts and central veins – to really understand the normal distribution of these cells in a normal liver. (a) Secondly, I do not really understand how you can claim that the pseudo normal liver dataset is the biliary niche. The immune sets that have been fax sorted could come from anywhere. Why do you claim they are totally representative of the biliary niche?. (b) However, I accept that the BuildNiche assay suggests a biliary and lobular niche from the generated resultant dataset.

Thank you for this feedback, please find below our responses to the points raised:

(a) The spatial data in Fig. 1A was used as a screening approach. Subsequent IHC-based analyses show indeed the proposed tissue localization (compare Fig. 1D). As such, we believe the use of the technique is justified in this setting. To exemplify this, we extended the visualization of the TMA cores for additional immune cell markers, that show a particular enrichment at the portal and not lobular niche (e.g. CD117).

We also made a corrplot analysis of respective markers supporting our claims (**new Fig. 1E, compare response to R2**).

In the novel Fig. 1E, a spearman correlation analysis of marker-positive cells detected in our tissue microarray dataset is depicted. This analysis enabled the correlation of marker positive cell numbers that should be enriched in portal fields only, e.g. with

CK7⁺ BEC. For example, we see a strong correlation for CD3/CD8 and CK7, while CD207 and CK7 show a correlative trend, however, not reaching the level of significance, which may be due to different statistical power based on the number of cores being analysed.

We believe the requested correlation analysis improves our manuscript and supports the hypothesis of an immunobiliary niche that contains T cells and DCs, which is later on confirmed using independent techniques.

Correlation Plot (novel Fig. 1E) depicting Spearman correlation coefficients among marker-positive cells (%). The n-number of cores being analyzed containing pseudonormal human liver tissue are listed in the updated Fig. 1E. Levels of significance displayed by asterisks.

- (b) We see the importance for clarification and reframe the statements we made in the respective sections. We would like to further explain the data and changes in the following:

The reviewer points out correctly, that the human scRNA-seq data is derived from small and unprocessed tissue pieces derived from human surgery. Based on this, no manipulation such as perfusion was conducted, so that the immune cell location of the suspension scRNA-seq is not specific to its histological location. We corrected this statement in the manuscript and now state, that we aimed for an enrichment of immune cells residing at the immunobiliary niche (Results, first paragraph). This aim was followed by first identifying cells that specifically reside at the biliary niche, and then selective enrichment of such cells in our scRNA-seq data by FACS sorting of respective cells (cDC2, gdT). We agree that this does not enable specific spatial enrichment, since contaminating sources may give rise to our populations of interest. For the human scRNA-seq data and the aim to enrich for immune cells, we modified our statements to avoid any overstatement:

“For cell type annotation of this high-resolution ST reference, we generated a human pseudonormal liver scRNA-seq reference dataset with the aim to enrich for immune cells.”

Explicitly, we removed the statement, that these cells may be spatially enriched from the immunobiliary niche.

With regard to mouse, we did a lot of morphology-based analyses (colocalization, IF, IHC) in combination with the findings of the human TMA and the use of bioinformatic tools (e.g. build niche assay). We then enriched respective cell populations in an informed setting, so that we expect these cells to be derived from the immunobiliary niche – at least to a certain extent.

To consolidate this finding and our key message in mouse, we conducted spatial transcriptomics on the DDC D5 timepoint using the Xenium Platform with a pre-build cell type gene panel containing 379 genes (“mouse tissue atlasing panel”) in combination with a self-designed add-on panel of 100 genes that enables specific localization analysis of cDC2 (e.g. *Mgl2*, *Clec10a*) and gdT cells (*Trdv4*, *Scart1*, *Scart2*, *Il23r*) among other cells (novel main Figure 5).

This assay confirmed the respective findings in our manuscript, including cDC2-gdT cells in close proximity to each other within the peribiliary niche, expression of transcripts of T17-polarization related transcription factors by gdT cells (e.g. *Irf4*) and respective niche interactions as outlined by NiCo analysis. In our point of view, this additional analysis strengthens our claims about the immunobiliary niche made in the manuscript. To include these results, we generated a novel paragraph in the results chapter (“**Spatial niche deconvolution resolves cDC2B and gdT cell containing cholangitis-associated tissue domains**”). Please see the novel Fig. 5 below including the Figure legends. For the main results text please refer to the revised manuscript.

Figure 5

Figure 5. Spatial niche deconvolution identifies cDC2B and gdT cell containing cholangitis-associated tissue domains. (A) Portal field overview of cell segmentation stain (top) and decoded transcripts (bottom) of a mouse DDC D5 liver tissue. Image overviews of whole sample are found on the top left. Scale bar 2 mm and 200 μm. **(B)** Density maps displaying DDC D5 liver tissue overview with binned expression of *Epcam*, *Mgl2*, *Cd3e* and *Ifr4*. Binsize 80 μm. **(C)** Spatial UMAP (top) and spatial location map (bottom) highlighting NiCo cell type annotations of DDC D5 liver tissue. Right column indicates cluster number, annotated cell types and predicted cell numbers in spatial section. NM indicates not matched cells [7]. **(D)** Spatial maps of NiCo predicted BEC, cDC2 and I17a⁺ gdT cells. **(E)** Top: Spatial cell type interaction map derived by NiCo using a radius of 20. Normalized logistic regression coefficient cutoff $c = 0.05$. **(F)** Spatial neighbourhoods (normalized logistic regression coefficients) for sd BEC, cDC2, I17a⁺ gdT cells. Error bars indicate s. e. of the coefficient estimates. **(G)** High resolution images containing segmentation stain information (left) or transcript abundance (right) of gdT cell- (*Cd163l1*, *5830411N06Rik*, *Il23r*, *Trdv4*), T cell- (*Cd3e*), T17 polarization (*Ifr4*), cDC2B- (*Mgl2*, *Clec10a*) and BEC-related (*Epcam*) transcripts in a DDC D5 liver portal field of interest. **(H)** Left: TRDC and CD207 immunohistochemical staining on human PSC liver tissue ($n=18$). Peribiliary immune infiltrates are enriched for TRDC+ and CD207+ cells. Asterisk indicates bile duct. Scale bar, 40 μm. Middle: High- and low-resolution visualization of color-coded TRDC+ and CD207+ detection counts in human PSC tissue. Analysis indicates varying cell concentration within different tissue compartments. Scale bar, 250 μm. Right: Duplex TRDC/CLEC10A, TRDC/CD207 in-situ hybridization of human PSC liver tissue ($n=5$). Cellular colocalization and/or close proximity relationships between cells can be detected. Scale bar 10 μm.

4) The authors then describe once again that have generated a single cell RNA seq from DDC fed mice at days 3,5,9 and 25. Once again there is a repeated claim that all these cells are from the immunobiliary niche – once again they could come from any compartment of the liver (Portal, Z1,2,3 and central) – I think this is slightly misleading for non-mouse hepatic

experts.

Thank you for your feedback.

This comment aligns with the concerns raised in 3). We included a clarifying statement to avoid presenting any misleading information and to better clarify our results in context. Please also refer to our response to point 3) and, in particular, to the supporting novel spatial transcriptomics data in new Figure 5.

Additionally, we would like to make the following points:

The experimental conduct of our tissue atlas contains an *in situ* perfusion step via the portal vein. This leads to a strong reduction of intravascular circulating immune cells, so that we believe that the later enriched cells are tissue resident. Together with our new spatial transcriptomics data, this explains, why we believe the cDC2 and peribiliary T cell compartment may be captured very specifically. However, we fully agree that other cells captured in our atlas such as LSEC, KC etc. are indeed not part of the immunobiliary niche.

To clarify this, we made the following changes:

- We included low resolution images of the Mgl1/2 staining of Fig. 3E into the Supplemental Figure 4H to highlight its very specified localization (novel supplemental Fig. 4H). The specific localization of these cells at the portal/immunobiliary niche can be appreciated. However, we agree that the localization of gdT cells can be both portal and lobular in steady state liver.
- We included a novel spatial transcriptomics dataset of DDC D5 to demonstrate existing immune cell gradients in a ST dataset. Again, we can see a close proximity of DC2, and gdT17 in the disease context (compare novel Fig. 5D), while other immune populations may show a more balanced distribution throughout the tissue.

Additionally, we also generated DDC D18 ST Xenium data, however, this data is part of a different project and can thus not be included into this study. However, to exemplify the reproducibility of the identified niches between DDC D5 and D18 tissues, we are including these data into this response letter.

Based on the same NiCo analysis, the gdT17/cDC2/Ccr7+ mat DC/sd BEC niche is also detected at D18, supporting our claims of the relevance of a gdT cell/cDC2 communication node.

NiCo interaction analysis of DDC D18 tissue using same scRNA-seq reference and same add-on pool. (A) Niche interaction map captures the Il17a⁺ gdT cell communication node with interactions to cDC2. (B) Spatial map and spatial UMAPs with predicted location of cDC2 and gdT. Cells colocalize in close proximity to large vessels (portal fields). Unpublished data.

We hope the novel data in Fig. 2 and 5 (ST data including DDC D5), the extended information about the Mgl1/2 immunohistochemistry data and the changes within the manuscript address previous concerns of the reviewer.

5) The authors then demonstrate that the small ducts in portal fields during DCC injury release a range of inflammatory genes highlighting these are chemoattractant to T and DC cells – as they are for many cells, so this in itself is not proof of a direct interaction. Therefore, the specific Importance of these observations can only be generalised. (a)

The authors then highlight similar findings in the BDL model but not the CCL4 model – which is to be expected – given the latter is a hepatocyte damage model. (b) This is then followed by speculation that BEC TNF production activates DC2's and is followed by an experiment where ex-vivo the DCs are treated with LPS – a TL4 agonist. The relevance of which was not clear to me. Why were they not treated with TNF in the ex-vivo analysis. (c) However, I fully accept that the co- ligand analysis was supportive of TNF interaction between DC2's and the biliary compartment.

This comment includes several points that we would like to address separately:

(a) We changed the sentence about the identified chemoattractant molecules and point out that these have general effects on multiple immune cell populations. We now point out that these findings have to be seen in the general context (see edited sentence in the results chapter “...which are known to regulate cDC2 and T cells **among other immune cell populations.**”).

(b) This is a valid and important point, yet we also want to point out that hepatocyte damage-driven models may simultaneously cause changes in the biliary compartment (e.g. ductular reaction). We included novel data of an acute hepatocyte damage and steatohepatitis model (CDE diet), that shows a similar but delayed trend towards gdT17 responses (**novel Fig. S7B, compare comment R1 Q14**). Furthermore, cDC2B localization in the CDE model remains detectable at the portal and peribiliary site (**novel Fig. S4I depicting immunohistochemical staining of Mgl1/2 on CDE**) even though this model induces hepatocyte damage. So, indeed cell-specific (hepatocyte/cholangiocyte) damage responses targeting different cell types can be seen. By including this immunohistochemistry, we also strengthen the aspects of the Mgl1/2 cell recruitment observed in DDC (compare Fig. 3E). In our point of view, in the CDE setting, the absent colocalization of cDC2B and the site of hepatocyte injury could even serve as a potential explanation of the delayed gdT17 responses compared to the DDC model.

Another way to differentiate between models would be the categorization into acute and chronic damage models. We believe the acute damage response may be a stronger inducer of respective changes in the gdT cell compartment. This may explain, why the novel CDE data, as an acute damage model, may also result in a delayed gdT17 response, while such differences may not be observed in chronic liver damage.

Novel Figure S4H, I: (H) Mgl1/2 immunohistochemistry depicting the specific localization of Mgl1/2-positive cells in control and DDC D19 liver. **(I)** Mgl1/2 immunohistochemistry depicting positive cells at portal fields in steady state and a rapid-onset experimental model of steatohepatitis (CDE D8, D14)

(c) Thank you for pointing this out. The rationale would indeed be much more clear, if we stimulated the cells with **Tnf** instead of a Nfkb inducer such as LPS.

We repeated the experiment using recombinant Tnf and measured multiple maturation genes by qPCR after Tnf treatment. Results were strongly comparable to the previously generated results with LPS. **We replaced the graph in Fig. 2D (see below).**

Novel Figure 2D depicting relative RNA levels of DC maturation markers in control cultivated DCs (light grey) and in Tnf treatment (dark grey).

6) The authors then show that there is an expansion of DC2s in their murine model and confirmed this by FaxAnalysis. The authors then focus on the DC2B lineage – and show an accumulation of mature cells and pre-cells repopulating the niche with the expression of genes that would favour local IL-17 production. Finally, they show that this population lose their ability to express Mgl2 (and propose this as a local maturity marker).

Thank you for summarising the rationale of the manuscript in this point. We agree with the summary and point out that we clarified the concerns by the other reviewers in these text sections (e.g. clarification of the Mgl1/2 staining and the abundance of conventional and unconventional T cells producing Il17a).

7) The authors then provide evidence for 2 populations of Gamma delta T cells – based on Scart1 and 2 expression. These 2 populations show differences in IL-17 expression within different TCR clones. This was shown to be due to differences in key transcription factors. In addition, competition for signals between alpha beta and gamma delta is provided for this niche. However, note is made of this low frequency population within the experiment and the pooling was required to conduct the experimental design **(a)**. So, the importance of this

low frequency cellular niche has to be questioned **(b)**. Furthermore I am not sure that the demonstration of IL-17a knock out reversing fibrosis is direct proof that gamma delta T cells drive fibrosis concretely **(c)**. This is because there are many cells that produce IL-17 in these models.

Thank you for your feedback. Please see our responses to your questions below:

(a) We quantify the absolute number of gdT17 and abT17 using FACS to demonstrate, that gdT17 cells are of importance in the process of fibrosis. This crucial point has also been raised by Reviewer 1 and the quantification, which shows that gdT17 are the major IL17-producing cell type is described in our response to **R1 Q3**. Furthermore, the unperturbed (non-restimulated) T cell subset data even hint towards a higher specificity of IL17a expression in our data (compare Fig. S5B). We note that the capacity of scRNA-seq versus FACS data to reflect *in vivo* T cell states have to be discerned, with scRNA-seq data likely reflecting the *in situ* more faithfully as laid out in our response to **R1 Q3**.

Furthermore, with the new ST data included, we can also visualize gdT cells in DDC disease, which further supports an important pathophysiological role of gdT17 cells in this model (novel Fig. 5). With regard to the pooling of the gdT cell multiplexed experiment depicted in Fig 4A, we would like to respond, that we conducted such pooling to save costs. Indeed, in control liver tissue, few gdT cells can be isolated, however, this drastically changes upon DDC treatment.

(b) In summary, we demonstrate that gdT cells are an exclusive source of IL17 according to our scRNAseq data, and a major source in restimulated cells *in vitro*. Together with their peribiliary localization detected in our novel spatial transcriptomics data (Fig. 5), these observations strongly nominate gdT cells as a key cell type governing the response to DDC diet. Beyond these data, the importance of gdT cells in the initiation of a profibrotic response is supported by the known differences between gdT and abT cells such as a faster gdT cell response [9].

(c) Thank you for pointing out, that IL17a may also be derived from sources other than gdT cells. To strengthen the significance of gdT cells as a source of IL17a and a driver of a profibrotic response, we now added experimental data from the Tcrd-GDL model for the depletion of gdT cells using diphtheria toxin [2] while administering DDC.

A combined DT-based gdT depletion on Day -1 and 2 and a DDC diet until D5 was conducted to read out the following parameters:

- a. We measured blood plasma cholestasis parameters (total bilirubin and alkaline phosphatase), which were significantly reduced in the context of conditional gdT cell depletion 5 days after initiation of the diet (see **novel Fig. 4I**).
- b. We measured transcript abundance of typical fibrosis markers after tissue extraction of RNA and cDNA synthesis and found a uniform trend towards an increase of abundance of fibrosis-associated transcripts reaching levels of statistical significance (**novel Fig.4J**).
- c. We enriched DCs by magnetic purification of CD11c⁺ cells and analysed the transcript levels of markers associated with DC maturation. These results could give some additional insight if gdT cell depletion is associated with altered DC education/maturation effects (as was also suggested in **R1 Q14**). Indeed maturation-related transcripts of DCs showed increased abundance of *Ccr7*, *Ccl17*, *Ccl22*, *Cd80* and *Arpin*. This could point towards two directions: first, the

influx of immature preDC could be altered and potentially reduced in the context of gdT depletion, or second, homing of fully mature DC may be altered. Altogether, we hope that these new data address the reviewers' comments and identify gdT cells as fibrosis-promoting cells in liver cholestasis.

Novel Figure 4I,J depicting comparative analysis at DDC D5 condition using C57BL/6J, TCRD_KO, TCRD_GDL mice. Blood plasma parameters of cholestasis are reduced at DDC D5 in an inducible model of gdT cell depletion (total bilirubin and alkaline phosphatase, left). Tissue-derived transcript levels of fibrosis genes in C57BL/6J, Tcrd-KO, Tcrd-GDL mice reveal differences between control and gdT ko models (right). These figures have been included as novel panels into Fig. 4 (novel Fig. I,J).

8) The authors then show that depletion of DC2B cells show a reduction in BEC inflammatory gene expression and also show IL-17 down regulation in gamma delta T cells - suggesting a functional relationship between these 2 cellular compartments

9) Finally, the authors show a clear evidence of Lymph Node communication between DC2 and gamma delta T cells and provide interactive ligand relationships between these 2 compartments.

8+9: we agree with the summary of the reviewer regarding our statements and do not see any points raised to address in these comments.

Suggestive improvements/extra work could include:

1: I personally think that the PSC data is not needed for this paper – it's not representative of this model and therefore its relevance to the paper has to be called into question

We put the data into context and changed our statements according to the comments of R1 and R2. Regarding disease amplification via local biliary injury, we believe that certain aspects of pathophysiology of PSC may be reflected by this model, while we set the PSC disease evolution into clear context. Newly generated data (TRDC/IL17A ISH) etc. may imply a certain translational relevance, so we decided to keep PSC data to some extent, while we agree that not all the data related to PSC has to be included into this manuscript. We followed the suggestion and excluded the PSC scRNA-seq data (previous S7E-H).

2: Would it not be more beneficial to develop biliary organoids from humans and induce injury/ and see if they attract/modulate DC2 migration in migration assays

Thank you for this suggestion. Unfortunately, our laboratory does not have frequent access to fresh human liver tissue, so that human biliary organoids cannot be grown. We believe establishing the technique of cultivation of human liver organoids, which are currently not established in our lab would be beyond the scope of the current manuscript.

However, we are thankful for the idea of the reviewer to test for DC migration and we conducted the following experiment: We isolated mouse bile from DDC D5 mouse gall bladder to experimentally test if bile may have chemo-attractant properties for DCs. We used Matrigel-coated transwell inserts (pore size of 8 μm) and cultivated isolated DC into the upper part of the insert to investigate if these cells would migrate towards a concentration gradient of bile.

Compared to control conditions, there was no clear trend towards increased migration using this (potentially too simple) setup. These data are in line with another cultivation experiment we conducted, where we treated liver DCs with DDC D5 bile and measured transcript abundance of DC maturation markers. Here, the results showed a significant reduction of *Cd80*, *Ccl17*, *Ccl22* transcript levels (n=3 per group, one sided t-test) supporting the idea, that bile acid treatment may have some DC-toxicity related effects. We believe that both experiments (transwell migration, in vitro cultivation) do not add much value to the current manuscript and we decided to not include these results, but want to emphasize the experimental efforts we made as part of our manuscript revision.

Finally, we also decided to treat enriched gdT cells with DDC D5 bile to see if Il17a production may be induced, independent of the presence of DCs. Such data could further substantiate if exogenous environmental factors alone suffice to induce gdT17 states in cholestasis.

We mimicked the cholestatic environment in gdT by administration of DDC D5 derived mouse bile (1:500 dilution) and cultivated these cells for 24 h. Il17a production in gdT cells among untreated and bile acid treated groups was similar and not significantly different. This indicates that other cell-derived factors, as further exemplified in our study are likely to be the main inducers of a gdT17 cell state. We believe this result is of importance and we included these novel results as new part of Fig. S7D (see novel plot below).

Novel Supplemental Figure S7D depicting gdT17 cell states in untreated and DDC D5 bile treated gdT cells. Two independent experiments (n=5).

3: Use human DC2s and co-culture with gamma delta T cells to understand interactions

Thank you for this suggestion. We have thought about the potential design of such an experiment and the gain of information it could yield. For example, we could isolate and stimulate DC and gdT cells from human blood and later co-culture these cells to investigate interaction between these cells.

Based on the major modifications conducted as part of the revision of this manuscript, where we generated additional functional readouts of mouse gdT cells and the fact that we toned

down the transferability of the identified mechanisms to human PSC, we believe that these experiments are beyond the current scope.

4) Why has a gamma/delta T cell knock out not be studied in this model - would that not be possible to show their importance in fibrosis in this model if this cellular source was of vital importance

Thank you for this feedback. We are thankful for this feedback, which prompted us to conduct additional in vivo experiment using Tcrd-KO (constitutive KO) and Tcrd-GDL (inducible KO) mice to elucidate the role of the gdT cells more specifically, their niche and the expression of Il17a in the context of liver fibrosis. We believe these data strongly improved our revised manuscript. Please refer to **R1 Q14** and **R3 Q7** for the additional data generated and implemented into the revised manuscript.

References

1. Itohara, S., et al., *T cell receptor delta gene mutant mice: independent generation of alpha beta T cells and programmed rearrangements of gamma delta TCR genes*. Cell, 1993. **72**(3): p. 337-48.
2. Sandroock, I., et al., *Genetic models reveal origin, persistence and non-redundant functions of IL-17-producing gammadelta T cells*. J Exp Med, 2018. **215**(12): p. 3006-3018.
3. Poch, T., et al., *Single-cell atlas of hepatic T cells reveals expansion of liver-resident naive-like CD4(+) T cells in primary sclerosing cholangitis*. J Hepatol, 2021. **75**(2): p. 414-423.
4. Thomann, S., et al., *Immunologic landscape of human hepatic hemangiomas and epithelioid hemangioendotheliomas*. Hepatol Commun, 2024. **8**(1).
5. Thomann, S., et al., *Digital Staging of Hepatic Hemangiomas Reveals Spatial Heterogeneity in Endothelial Cell Composition and Vascular Senescence*. J Histochem Cytochem, 2022. **70**(7): p. 531-541.
6. Muller, A.L., et al., *Inflammatory type 2 conventional dendritic cells contribute to murine and human cholangitis*. J Hepatol, 2022. **77**(6): p. 1532-1544.
7. Agrawal, A., et al., *NiCo identifies extrinsic drivers of cell state modulation by niche covariation analysis*. Nat Commun, 2024. **15**(1): p. 10628.
8. Rosales-Alvarez, R.E., et al., *VarID2 quantifies gene expression noise dynamics and unveils functional heterogeneity of ageing hematopoietic stem cells*. Genome Biol, 2023. **24**(1): p. 148.
9. Godfrey, D.I., et al., *The burgeoning family of unconventional T cells*. Nat Immunol, 2015. **16**(11): p. 1114-23.
10. Ruedl, C. and S. Jung, *DTR-mediated conditional cell ablation-Progress and challenges*. Eur J Immunol, 2018. **48**(7): p. 1114-1119.

Response to the reviewer comments

The reviewer comments are shown in black and our responses are highlighted in blue. Changes in the manuscript are highlighted red as in track-changes mode.

REVIEWERS' COMMENTS

Reviewer #1 (Remarks to the Author):

I thank the Reviewers for addressing the original criticisms raised with their initial submission. It would be greatly appreciated if page numbers for all manuscript changes that are discussed in the PBP response are included and clearly stated in the PBP instead of writing they have been changed in the manuscript without page numbers or including the change in the PBP. Sometimes the text changes have been included in the PBP whereas in other locations, the text changes are only stated and not included in PBP and referenced to sections of the manuscript which makes it difficult to know what has been changed and how so without a significant re-review of the entire manuscript. PBP figures should be labeled with figure numbers (e.g. PBP fig.1) to aid review.

We thank the reviewer for this feedback and understand the point, that it would have been more precise to state the manuscript changes more specifically at all parts of the manuscript being changed including page numbers and line numbers. We would like to apologize for this point and aim to make our statements more specific and better comprehensible for the current response letter.

For this second response letter, we now also include a clear numbering of figures within the PBP response letter or the Figures within the Manuscript and make clear reference, where the main manuscript was changed.

Important: Our page and line references refer to the track changes mode, when all changed tracks are displayed. Track changes and page and line references are different, when changes are minimized.

1. Abstract greatly improved. Clear and concise.
2. Table is adequate.
3. Proportion of GD T cells of total T cells has not been provided. Please include GD T cell proportions alongside proportions of IL17+ GDT cell numbers/total IL17+ T cells (stacked bar plots). It is important to highlight the rarity of these cells as showing the proportion of IL17+ cells does to convey the scarcity of IL17+ GD T cells in general. Statement regarding density of GD T cells is not particularly informative (i.e. Overall TRDC+ cell concentration was $29.28 \pm 27.61/\text{mm}^2$, while local concentration within peribiliary regions could reach much higher densities (Fig. S7B).", see results section). This statement does not clearly illustrate relative increase of GD T cells in peribiliary areas vs non-bile duct regions. Please subset the two regions and provide relative fold-increase of GD T cells in peribiliary regions vs other areas.

Thank you for this constructive feedback. We included the comparison of the of $\gamma\delta$ T cells vs all other T cells derived from our flow cytometric analysis to illustrate the rare abundance of

these cells in general (PBP Fig. 1A). Even though these cells are very scarce in the liver, they significantly contribute to the pool of Il17a⁺ cells (PBP Fig. 1B). This observation in the FACS quantifications can also be detected in the scRNA-seq data, where at later timepoints $\gamma\delta$ T17 cells comprise approx. 4% of all T cells (D9: 3.9%, D19: 2.6%, D25: 3.7%).

Furthermore, we implemented the changes regarding the $\gamma\delta$ T cell quantification in Fig. S7B. For this analysis, we quantified regions of interests containing bile ducts/portal regions and lobular regions. Quantification of the TRDC⁺ cells in these regions was then conducted. 4 region of interests (ROI) were analysed per case (n=10 PSC). While biliary/portal region TRDC⁺ cell infiltration was heterogeneous, we quantified regions with high disease activity, typically at the portolobular interface. The average TRDC⁺ cell concentration was approx. 3.34 fold higher in portobiliary-inflamed regions (66.33/mm² +/- 54.07) vs lobular concentration (19.87 +/-25.38, two-sided T test, p = 8.16*10⁻⁶). We changed the respective sentence containing the overall previous information in the main manuscript and copy the changed sentence into the PBP.

New excerpt covering TRDC⁺ cell concentration in PSC (**page 11 line 305-309 in main manuscript, changes in RED**):

While PSC pathophysiology is more complex and not identical with the acute injury-driven immune response in a xenobiotic model of cholangitis, we could still detect TRDC⁺ and CD207⁺ cells, which showed a localization-dependent concentration **towards inflamed biliary foci compared to a lobular localization** (average TRDC⁺ cell concentration **in portobiliary and lobular regions was 66.33 ±54.07/mm² and 19.87±25.38/mm²**; Fig. 5H).

PBP Fig. 1

PBP Figure 1. (A) Stacked barplot displaying flow cytometric relative frequencies of $\gamma\delta$ T and non- $\gamma\delta$ T cells in mouse liver at D0, DDC D5/6 and DDC D10 timepoints. (B) Stacked barplots displaying stratification of Il17a⁺ cells per sample into $\alpha\beta$ T and $\gamma\delta$ T subsets as quantified by FACS. (C) TRDC⁺ cell concentration in human PSC tissues within portobiliary and lobular niches. FACS analysis (A,B): ctrl: n=9 mice, DDC D5: n=5 mice, DDC D10: n=4 mice. PSC tissues (n=10, 40 images per ROI).

4. As the pseudonormal tissue shows consistent evidence of liver damage and immune infiltration (acknowledged that TRM may be 'normal' or may be related to liver damage), the term pseudonormal should be replaced with 'non-cholestatic disease controls' to clarify that these samples are not from healthy livers but individuals that can be considered useful controls compared to cholestatic liver indications. Quantification of TCRD⁺ cells at D5 of DDC treatment should be provided in addition to representative IF image.

Thank you for these valuable suggestion to improve the manuscript.

We changed the terminology of previously stated "pseudonormal" liver tissue into "non-cholestatic disease control" liver tissue throughout the manuscript as suggested.

We believe, the use of the ITC-GDL TRDC⁺ cells are not informative to quantify for the following reasons:

- TRDC⁺ cells were very scarce and only to be detected using high magnification (100x) objectives and in distinct areas of the biliary tree. Thus, the approach to record images was based on a screening approach to specifically look for portal field regions with high magnification, while many regions were also devoid of TRDC⁺ cells. These observations are in line with our NiCo analysis (Main Fig. 5C/D, 162 Il17a⁺ γδ T cells on one whole section of a DDC D5 liver) and our previous PBP response for R3 Q4 displaying DDC D18 results of our NiCo algorithm. All the different modalities (microscopy, flow cytometry, spatial transcriptomics) indicate a rare presence of these cells, so that other means of quantifications, that are already delivered in the manuscript are more accurate and informative for the reader.

5. Please include explanatory statement in main text of manuscript (i.e. ‘To enrich all cell types at equal ratios, two independent rounds of cell isolation and sequencing were performed at all time points’).

Thank you for the suggestion. We included such explanation into the main body of the manuscript as suggested (last sentence of first results paragraph). The respective sentence can be found at **page 5, line 105-106 (changes in RED)**:

To enrich all cell types at equal ratios, two independent rounds of cell isolation and sequencing were performed at all timepoints (see methods) and we were able to capture abundant cells of the liver immunobiliary niche, as well as rare liver immune cells such as cDCs (Fig. 1K, S2F-H, **Supplementary Data S1, Methods**).

6. Appropriate citations of prior work supporting BEC-centric DDC damage are needed for introductory statement.

Thank you for the feedback. We included one reference using spatial and single cell profiling of the DDC model to exemplify its use to study BEC-responses (**page 4, line 133- page 5 line 147**):

Wu, B., Shentu, X., Nan, H. et al. A spatiotemporal atlas of cholestatic injury and repair in mice. Nat Genet 56, 938–952 (2024). <https://doi.org/10.1038/s41588-024-01687-w>

7. Additional text is useful but does not appear in main manuscript. Please clearly note page number and section the new text regarding latent factor analysis has been added.

Thank you for this feedback. This was likely a mistake in the final editing of the revised manuscript and we would like to apologize for the inconsistency between response letter and main manuscript.

We included the paragraph below at methods section “Generation of DDC spatial transcriptomics data and analysis” (**page 19, line 545-550**):

“For the calculation of niche interaction maps, cell type enrichment ratios within cellular neighborhoods of a central cell were calculated and a logistic regression to predict central

cell type identities from these ratios was performed. The regression coefficients were then used to reconstruct the interaction map. Inferred latent factors from expression data were used to identify a small number of gene programs explaining cell state variability of each cell type. Covariation analysis of these latent factors across co-localized cell types may thus capture the effect cell-cell interactions on gene program activity [7].”

8. Good, thank you for clarifying.

9. Please include number of mice analyzed in the figure panels for Fig. 3C and Fig. 3G (i.e. n=2). Please include changes to new figures in PBP.

Thank you for this suggestion. We included the use of mouse numbers in Fig. 3C and Fig. 3G. The changed main Figure 3 can be found below:

Changed Main Figure 3, now containing information regarding the use of mice in the panels C and G.

10. Ok thanks for clarification.

11. Agreed that this point is similar to (3) and questions there apply also to original comments in (11).

12. Interpretation of DT depletion is inconclusive as it appears multiple subsets are affected, not only GD T cells. Clusters are not annotated in umap(A) so difficult to determine which are Th1, Th2 or Th17 AB T cells or GD T cells. Dot plots in (C) are confusing – does this represent all cells are only GD T cell cluster as appears to be all cells but should be shown for GD T cell cluster vs others that appear impacted with DT treatment. Explanation/findings from GD KO mice are generally unclear and need to be rephrased.

Thank you for your feedback and the possibility to clarify our results.

Indeed, multiple subsets are affected by this cellular depletion, and the DTR system also induces off-target effects and the immunogenicity of Diphtheria toxin, as also mentioned in the previous PBP and literature (compare R1 Q12) [10] and objectified by the extent of cells that express, e.g., *Ifng* across conditions (**PBP Fig. 2A,B**). However, a comparative analysis of NK/NKT/T cell subsets is difficult to assess, since some of the cell types have almost exclusively been captured only within one of the two conditions (e.g. NK/NKT cell compartment). Further quantitative analysis, such as FACS, has not been conducted for these cell types, making a conclusion if certain cell types prevail after DT injection impossible.

$\gamma\delta$ T cells were part of this dataset within cluster 15 in both conditions. $\gamma\delta$ T cells were expressing *Trdv4*, *Cd1631* (*Scart1*) and exclusively *Il17a*. The increased expression of *Il17a* in the non-depleted condition is already visualized in **Fig. 6C,D** of the main manuscript. To address the shortcomings of the DTR-model, we already included a text change in the main manuscript in the previously submitted manuscript (compare previous **PBP R1 Q12**).

Based on your feedback, we changed and rephrased the passage of TRDC KO to better elucidate the shared microanatomical niche aspect of $\gamma\delta$ T and $\alpha\beta$ T cells and the disease promoting role of $\gamma\delta$ T cells (**page 9 line 272 following**, changes in **RED**):

To interrogate, if $\gamma\delta$ T and $\alpha\beta$ T cells compete for signal cues **within the peribiliary niche**, *Tcrd*-knockout (KO) mice were treated with DDC. Indeed, compensation of the $\gamma\delta$ T17 function was reflected by a higher percentage of *Il17a*⁺ $\alpha\beta$ T cells in *Tcrd*-KO versus control mice at DDC D5, suggesting potential competition for $\gamma\delta$ T17/Th17 polarization cues in liver cholestasis **within a shared microanatomical niche** (Fig. S7C).

In vitro treatment of $\gamma\delta$ T cells with DDC D5 bile did not induce *Il17a* expression (Fig. S7D). Conditioned medium of DDC D5 derived DCs showed a trend towards inducing increased *Il17a*⁺ $\gamma\delta$ T cell frequencies ($p=0.06$), however, compared to direct co-culture, the fraction of *Il17a*⁺ cells was low (Fig. S7E). Direct co-culture of DDC D5- or control liver-derived DCs with UTCs from control spleen led to increased *Il17a* production in $\gamma\delta$ T cells but only weak induction in $\alpha\beta$ T cells, indicating a DC-derived contribution to the induction of a $\gamma\delta$ T17 cell state (Fig. 4G).

Based on this observation, we used TCR stimulation *in vitro* to further elucidate the impact of contact-based/soluble mediators in inducing a $\gamma\delta$ T17 cell state. Specific stimulation with recombinant *Il18*, *Cxcl16*, *Lgals9* and activating *Icosl*-antibodies revealed a moderate $\gamma\delta$ T17 cell state induction for *Icosl* activation, while soluble mediators did not induce significant differences (Fig. 4H, S7F). This prompted us to test a **disease-contributing**

role of $\gamma\delta T$ cells in DDC-induced damage response and cholestasis using a constitutive (Tcrd-KO) and conditional depletion model of $\gamma\delta T$ cells (Tcrd-GDL [17]). Conditional $\gamma\delta T$ cell depletion was associated with reduced circulating total bilirubin and alkaline phosphatase levels indicating reduced cholestasis and disease activity (Fig. 4I). Furthermore, tissue-derived transcript levels of *Pdgfrb*, *Col1a1* were reduced in Tcrd-KO and/or Tcrd-GDL liver tissues implying a functional contingency of liver fibrosis on $\gamma\delta T$ cell-derived Il17a (Fig. 4J). To directly test the effect of Il17a on fibrosis, we treated wildtype (wt) and Il17a/f KO mice with a long term DDC diet (16 days) and quantified the number of *Pdgfrb*⁺ myofibroblasts and collagen (Sirius Red) in tissue sections. The number *Pdgfrb*⁺ cells and collagen area was significantly reduced in Il17a/f KO versus wt mice, mirroring a profibrogenic role of Rorc-induced Il17-responses [39] (Fig. 4K,L). Similarly, tissue-derived transcript levels for *Acta2*, *Des*, *Col1a1* were reduced in DDC-treated Il17a/f KO vs controls (Fig. S7G).

Together, these results suggest Il17a-producing $\gamma\delta T$ cells as an early-onset fibrosis-supporting communication node in murine cholangitis.

To visualize the respective T cell population in the DDC D5/DDC D5-DT conditions, we have included **PBP Fig. 2** below.

We hope these explanations and changes address the main points being raised.

PBP Fig. 2

A

B

PBP Fig. 2. T cell activation and myeloid cell distribution across DDC D5 and DDC D5-DT conditions:

(A) UMAP representation displaying clusters (left) and samples (right) of the T cell subset of merged DDC D5 and DDC D5-DT data.

(B) UMAP representation of the T cell subset of the combined DDC D5/DDC D5-DT dataset displaying log-normalized gene expression of marker genes. Log-normalized expression is color coded and fraction of cells expressing the gene is encoded by dot size (right).

13. Explanation regarding removal of PSC samples is perplexing as removal eliminates major translation to human setting. Data of in situ staining is discussed but now shown.

Thank you for the possibility to further clarify our modifications in presenting parts of the human PSC data. We decided to remove data, where patient sample numbers were too low and adequate controls were missing. This was the case for the scRNA-seq of the two PSC patients that we analyzed.

However, we still included major results of human PSC tissues, where we conducted immunohistochemical analysis (n=18 PSC tissues) and in-situ hybridization (n=5 PSC tissues). Based on these findings, we believe that major translation is still given, and which is why we do not want to include the word “murine” into our manuscript abstract (see comment on revised manuscript file).

Regarding the **In Situ Hybridization (ISH) on human PSC tissues**, we would like to make the following clarification:

In the manuscript, we highlight the colocalization of TRDC⁺ with CLEC10A⁺/CD207⁺ cells, which fit the narrative being generated in mouse. However, we did not include the results of IL17A⁺ TRDC⁺ ISH, which is more difficult to interpret, since only a minority of TRDC⁺ cells coexpressed IL17A⁺. This partial exclusion of ISH results may have led to a misunderstanding, that all ISH results have been removed, which is not the case (see Fig. 5H).

We hope these explanations clarified the rationale to include or exclude results, while in general we aimed for a mouse-centric point of view, as also suggested by the reviewers and the editor.

Reviewer #2 (Remarks to the Author):

The authors have carefully addressed my concerns.

We would like to thank Reviewer #2 for this positive feedback and your previous helpful suggestions.

Reviewer #3 (Remarks to the Author):

Thankyou for the resubmission of the manuscript. First of all I would like to thank the authors for the resubmission of their manuscript and for their detailed rebuttal response to the all the points I previously raised.

In doing so, the manuscript is now far more comprehensive and generalisable to the field of hepatology, and provides a detailed understanding of the role of cDC2B and gamma delta cells in cholestatic liver disease.

In particular the correlation analysis and extra spatial xenium analysis (Fig 5) add to the strength of the mechanistic data provided. Also the additional GD KO experiments provide support for the importance of this cellular interaction. Additionally I am grateful to the authors for sharing additional data on the D18 model that they are currently working on to add strength to their arguments.

Finally the additional bile exposure experiments and TNF data provided take the biological model provided to its natural limits. The abstract and conclusions are also now far more balanced.

In conclusion I am therefore satisfied all the points I previously raised have been addressed.

We would like to thank Reviewer #3 for this positive feedback and your previous helpful suggestions.

Finally, we would like to state, that we changed the display of **Main Fig. 1E**. In the previous version, the spearman correlations within the corrplot function were calculated, while displaying the statistics of a different correlation metric, making the statistical display incorrect.

We corrected this mistake as part of the final review of all statistics and now conducted an two-sided Spearman's rank correlation test and display the respective p-values in the corrplot. This change also demonstrates the statistical significance between CD207⁺ and KRT7⁺ cells. Additionally, in the previous version of the **source data**, we changed the labeling of the groups in **S7B** for DDC. When double checking the source data for re-calculating the statistics, we also found that previous **source data for S9H** was showing qPCR data of a different experiment. We corrected this and now include the correct source file for S9H.